# QuPeD: Quantized Personalization via Distillation with Applications to Federated Learning

**Kaan Ozkara**
University of California, Los Angeles
kaan@ucla.edu

**Navjot Singh**
University of California, Los Angeles
navjotsingh@ucla.edu

**Deepesh Data**
University of California, Los Angeles
deepesh.data@gmail.com

**Suhas Diggavi**
University of California, Los Angeles
suhas@ee.ucla.edu

## Abstract

Traditionally, federated learning (FL) aims to train a single global model while collaboratively using multiple clients and a server. Two natural challenges that FL algorithms face are heterogeneity in data across clients and collaboration of clients with *diverse resources*. In this work, we introduce a *quantized* and *personalized* FL algorithm QuPeD that facilitates collective (personalized model compression) training via *knowledge distillation* (KD) among clients who have access to heterogeneous data and resources. For personalization, we allow clients to learn *compressed personalized models* with different quantization parameters and model dimensions/structures. Towards this, first we propose an algorithm for learning quantized models through a relaxed optimization problem, where quantization values are also optimized over. When each client participating in the (federated) learning process has different requirements for the compressed model (both in model dimension and precision), we formulate a compressed personalization framework by introducing knowledge distillation loss for local client objectives collaborating through a global model. We develop an alternating proximal gradient update for solving this compressed personalization problem, and analyze its convergence properties. Numerically, we validate that QuPeD outperforms competing personalized FL methods, FedAvg, and local training of clients in various heterogeneous settings.

## 1 Introduction

Federated Learning (FL) is a learning procedure where the aim is to utilize vast amount of data residing in numerous (in millions) edge devices (clients) to train machine learning models without collecting clients' data [26]. Formally, if there are $n$ clients and $f_i$ denotes the local loss function at client $i$, then traditional FL learns a single global model by minimizing

$$\arg\min_{\mathbf{w} \in \mathbb{R}^d} \left( f(\mathbf{w}) := \frac{1}{n} \sum_{i=1}^{n} f_i(\mathbf{w}) \right). \tag{1}$$

It has been realized lately that a *single* model may not provide good performance to all the clients in settings where data is distributed *heterogeneously*. This leads to the need for personalized learning, where each client wants to learn its own model [7, 8]. Since a locally learned client model may not generalize well due to insufficient data, in personalized FL process, clients maintain personalized models locally and utilize other clients' data via a global model. *Resource diversity* among clients,

35th Conference on Neural Information Processing Systems (NeurIPS 2021).

which is inherent to FL as the participating edge devices may vary widely in terms of resources, is often overlooked in personalized FL literature. This resource diversity may necessitate clients to learn personalized models with *different precision* as well as *different dimension/architecture*. Systematically studying both these resource heterogeneity together with data heterogeneity in personalized FL is the primary objective of this paper.

In this work, we propose a model compression framework[1] for personalized FL via knowledge distillation (KD) [14] that addresses both data and resource heterogeneity in a unified manner. Our framework allows collaboration among clients with different resource requirements both in terms of *precision* as well as *model dimension/structure*, for learning personalized quantized models (PQMs). Motivated by FL, where edge devices are resource constrained when actively used (*e.g.* when several applications are actively running on a battery powered smartphone) and available for training when not in use (*e.g.*, while charging and on wi-fi), we do training in full precision for learning compressed models to be deployed for *inference time*. For efficient model compression, we learn the quantization parameters for each client by including quantization levels in the optimization problem itself. First, we investigate our approach in a *centralized* setup, by formulating a relaxed optimization problem and minimizing it through alternating proximal gradient steps, inspired by [3]. To extend this to FL for learning PQMs with *different* dimensions/architectures, we employ our centralized algorithm locally at clients and introduce KD loss for collaboration of personalized and global models. Although there exist empirical works where KD is used in personalized FL [20], we formalize it as an optimization problem, solve it using alternating proximal updates, and analyze its convergence.

**Contributions.** Our contributions can be summarized as follows:

- In the centralized case, we propose a novel relaxed optimization problem that enables optimization over quantization values (centers) as well as model parameters. We use alternating proximal updates to minimize the objective and analyze its convergence properties.

- More importantly, our work is the first to formulate a personalized FL optimization problem where clients may have different model dimensions and precision requirements for their personalized models. Our proposed scheme combines alternating proximal updates with knowledge distillation.

- For optimizing a non-convex objective, in the centralized setup, we recover the standard convergence rate of $\mathcal{O}(1/T)$ (despite optimizing over quantization centers), and for federated setting, we recover the standard convergence rate of $\mathcal{O}(1/\sqrt{T})$ (despite learning PQMs with different precisions/dimensions). In the federated setting, our convergence bound has an error term that depends on multiplication of two terms averaged over clients: one characterizing client's local model smoothness and the other data heterogeneity with respect to overall data distribution.[2]

- We perform image and text classification experiments on multiple datasets in various resource and data heterogeneity settings, and compare performance of QuPeD against Per-FedAvg [8], pFedMe [7], FedAvg [26], and local training of clients. We observe that QuPeD in full precision outperforms all these methods on all the datasets that we considered for our experiments. Further, our results show that even with quantization, QuPeD outperforms the other methods in *full precision* on multiple settings and datasets, demonstrating the effectiveness of our scheme for FL settings.

Our work should not be confused with works in distributed/federated learning, where models/gradients are compressed for *communication efficiency* [2, 17]. We also achieve communication efficiency through local iterations, but the main goal of our work is personalized quantization for inference.

**Related work.** To the best of our knowledge, this is the first work in personalized federated learning where the aim is to learn quantized and personalized models potentially having different dimensions/structures for inference. Our work can be seen in the intersection of *personalized federated learning* and *learning quantized models*; we also employ *knowledge distillation* for collaboration.

*Personalized federated learning:* Recent works adopted different approaches for learning personalized models: (i) Combine global and local models throughout the training [6, 12, 25]; (ii) first learn a global model and then personalize it locally [8]; (iii) consider multiple global models to collaborate among only those clients that share similar personalized models [9, 25, 31, 34]; (iv) augment the

---

[1]Model compression (MC) allows inference time deployment of a compressed model. Though MC is a generic term comprising different methods, we will focus on its quantization (number of bits per model parameter) aspect.

[2]An error term depending on data heterogeneity is commonly observed in personalized FL algorithms [7, 8].

traditional FL objective via a penalty term that enables collaboration between global and personalized models [7, 11, 12].

*Learning quantized models:* There are two kinds of approaches for training quantized networks that are of our interest. The first one approximates the hard quantization function by using a soft surrogate [5, 10, 24, 32], while the other one iteratively projects the model parameters onto the fixed set of centers [1, 15, 19, 33]. Each approach has its own limitation; see Section 2.1 for a discussion. While the initial focus in learning quantized networks was on achieving good empirical performance, there are some works that analyzed convergence properties [1, 21, 33], but only in the centralized case. Among these, [1] analyzed convergence for a relaxed/regularized loss function using proximal updates.

*Knowledge distillation (KD):* KD [14] is a framework for transfer learning that is generally used to train a small student network using the soft labels generated by a deep teacher network. It can also be used to train two or more networks mutually by switching teacher and students in each iteration [35]. KD has been employed in FL settings as an alternative to simple aggregation which is not feasible when clients have models with different dimensions [23]. [20] used KD in personalized FL by assuming existence of a public dataset. [27] used KD in combination with quantization in a centralized case for model compression; in contrast, we do not use KD for model compression but for collaboration between personalized and global model. Unlike the above works which are empirical, our paper is the first to formalize an optimization problem for personalized FL training with KD and analyze its convergence properties. Our proposed scheme yields personalized client models with different precision/dimension through local alternating proximal updates; see Section 2.2 for details.

**Paper organization:** In Section 2, we formulate the optimization problem to be minimized. In Sections 3 and 4, we describe our algorithms along-with the main convergence results for the centralized and personalized settings, respectively. Section 5 provides extensive numerical results. Omitted proofs/details and additional experiments are provided in supplementary material.

## 2  Problem Formulation

Our goal in this paper is for clients to collaboratively learn personalized quantized models (with potentially different precision and model sizes/types). To this end, below, we first state our final objective function that we will end up optimizing in this paper for learning personalized quantized models, and then in the rest of this section we will describe the genesis of this objective.

Recall from (1), in the traditional FL setting, the local loss function at client $i$ is denoted by $f_i$. For personalized compressed model training, we define the following augmented loss function at client $i$:

$$
F_i(\mathbf{x}_i, \mathbf{c}_i, \mathbf{w}) := (1 - \lambda_p)\left(f_i(\mathbf{x}_i) + f_i(\widetilde{Q}_{\mathbf{c}_i}(\mathbf{x}_i))\right) + \lambda R(\mathbf{x}_i, \mathbf{c}_i) \\
+ \lambda_p \left(f_i^{KD}(\mathbf{x}_i, \mathbf{w}) + f_i^{KD}(\widetilde{Q}_{\mathbf{c}_i}(\mathbf{x}_i), \mathbf{w})\right). \tag{2}
$$

Here, $\mathbf{w} \in \mathbb{R}^d$ denotes the global model, $\mathbf{x}_i \in \mathbb{R}^{d_i}$ denotes the personalized model of dimension $d_i$ at client $i$, $\mathbf{c}_i \in \mathbb{R}^{m_i}$ denotes the model quantization centers (where $m_i$ is the number of centers), $\widetilde{Q}_{\mathbf{c}_i}$ denotes the soft-quantization function with respect to (w.r.t.) the set of centers $\mathbf{c}_i$, $R(\mathbf{x}_i, \mathbf{c}_i)$ denotes the distance function, $f_i^{KD}$ denotes the knowledge distillation (KD) loss [14] between the two input models on client $i$'s dataset, $\lambda$ is a design parameter for enforcing quantization (large $\lambda$ forces weights to be close to respective centers), and $\lambda_p$ controls the weighted average of regular loss and KD loss functions (higher $\lambda_p$ can be used when client data is limited). We will formally define the undefined quantities later in this section. Consequently, our main objective becomes:

$$
\min_{\mathbf{w} \in \mathbb{R}^d, \{\mathbf{x}_i \in \mathbb{R}^{d_i}, \mathbf{c}_i \in \mathbb{R}^{m_i} : i=1,\ldots,n\}} \left(F(\mathbf{w}, \{\mathbf{x}_i\}, \{\mathbf{c}_i\}) := \frac{1}{n}\sum_{i=1}^{n} F_i(\mathbf{x}_i, \mathbf{c}_i, \mathbf{w})\right). \tag{3}
$$

Thus, our formulation allows different clients to have personalized models $\mathbf{x}_1, \ldots, \mathbf{x}_n$ with different dimensions $d_1, \ldots, d_n$ and architectures, different number of quantization levels $m_1, \ldots, m_n$ (larger the $m_i$, higher the precision), and different quantization values in those quantization levels. Note that there are two layers of personalization, first is due to data heterogeneity, which is reflected in clients learning different models $\mathbf{x}_1, \ldots, \mathbf{x}_n$, and second is due to resource diversity, which is reflected in clients learning models with different sizes, both in terms in dimension as well as precision.

In Section 2.1, we motivate how we came up with the first three terms in (2), which are in fact about a centralized setting because the function $f_i$ and the parameters involved, i.e., $\mathbf{x}_i, \mathbf{c}_i$, are local to client $i$; and then, in Section 2.2, we motivate the use of the last two terms containing $f_i^{KD}$ in (2).

## 2.1 Model Compression in the Centralized Setup

Consider a setting where an objective function $f : \mathbb{R}^{d+m} \to \mathbb{R}$ (which could be a neural network loss function) is optimized over both the quantization centers $\mathbf{c} \in \mathbb{R}^m$ and the assignment of model parameters (or weights) $\mathbf{x} \in \mathbb{R}^d$ to those centers. There are two ways to approach this problem, and we describe these approaches, their limitations, and the possible resolutions below.

**Approach 1.** A natural approach is to explicitly put a constraint that weights belong to the set of centers, which suggests solving the following problem: $\min_{\mathbf{x},\mathbf{c}} f(\mathbf{x}) + \delta_{\mathbf{c}}(\mathbf{x})$, where $\delta_{\mathbf{c}}$ denotes the indicator function for $\mathbf{c} \in \mathbb{R}^m$, and for any $\mathbf{c} \in \mathbb{R}^m, \mathbf{x} \in \mathbb{R}^d$, define $\delta_{\mathbf{c}}(\mathbf{x}) := 0$ if $\forall\, j \in [d]$, $x_j = c$ for some $c \in \{c_1, \ldots, c_m\}$, otherwise, define $\delta_{\mathbf{c}}(\mathbf{x}) := \infty$. However, the discontinuity of $\delta_{\mathbf{c}}(\mathbf{x})$ makes minimize this objective challenging. To mitigate this, like recent works [1, 33], we can approximate $\delta_{\mathbf{c}}(\mathbf{x})$ using a distance function $R(\mathbf{x}, \mathbf{c})$ that is continuous everywhere (*e.g.*, the $\ell_1$-distance, $R(\mathbf{x}, \mathbf{c}) := \min\{\frac{1}{2}\|\mathbf{z} - \mathbf{x}\|_1 : z_i \in \{c_1, \cdots, c_m\}, \forall i\}$).[3] This suggests solving:

$$\min_{\mathbf{x},\mathbf{c}} f(\mathbf{x}) + \lambda R(\mathbf{x}, \mathbf{c}). \tag{4}$$

The centers are optimized to be close to the mean or median (depending on $R$) of the weights; however, there is no guarantee that this will help minimizing objective $f$. We believe that modeling the direct effect that centers have on the loss is crucial for a complete quantized training (see Appendix E for empirical verification of this fact), and our second approach is based on this idea.

**Approach 2.** We can embed the quantization function into the loss function itself, thus solving the problem: $\min_{\mathbf{x},\mathbf{c}} (h(\mathbf{x}, \mathbf{c}) := f(Q_{\mathbf{c}}(\mathbf{x})))$, where for every $\mathbf{x} \in \mathbb{R}^d, \mathbf{c} \in \mathbb{R}^m$, the (hard) quantization function is defined as $Q_{\mathbf{c}}(\mathbf{x})_i := c_k$, where $k = \arg\min_{j \in [m]}\{|x_i - c_j|\}$, which maps individual weights to the closest centers. Note that $Q_{\mathbf{c}}(\mathbf{x})$ is actually a staircase function for which the derivative w.r.t. $\mathbf{x}$ is 0 almost everywhere, which discourages the use of gradient-based methods to optimize the above objective. To overcome this, similar to [10, 32], we can use a *soft* quantization function $\widetilde{Q}_{\mathbf{c}}(\mathbf{x})$ that is differentiable everywhere with derivative not necessarily 0. For e.g., element-wise *sigmoid* or *tanh* functions, used by [32] and [10], respectively.[4] This suggests the following relaxation:

$$\min_{\mathbf{x},\mathbf{c}} (h(\mathbf{x}, \mathbf{c}) := f(\widetilde{Q}_{\mathbf{c}}(\mathbf{x}))). \tag{5}$$

Though we can observe the effect of centers on neural network loss in (5); however, the gradient w.r.t. $\mathbf{x}$ is heavily dependent on the choice of $\widetilde{Q}_{\mathbf{c}}$ and optimizing over $\mathbf{x}$ might deviate too much from optimizing the neural network loss function. For instance, in the limiting case when $\widetilde{Q}_{\mathbf{c}}(\mathbf{x}) \to Q_{\mathbf{c}}(\mathbf{x})$, gradient w.r.t. $\mathbf{x}$ is 0 almost everywhere; hence, every point becomes a first order stationary point.

**Our proposed objective for model quantization.** Our aim is to come up with an objective function that would not diminish the significance of both $\mathbf{x}$ and $\mathbf{c}$ in the overall procedure. To leverage the benefits of both, we combine both optimization problems (4) and (5) into one problem:

$$\min_{\mathbf{x},\mathbf{c}} (F_\lambda(\mathbf{x}, \mathbf{c}) := f(\mathbf{x}) + f(\widetilde{Q}_{\mathbf{c}}(\mathbf{x})) + \lambda R(\mathbf{x}, \mathbf{c})). \tag{6}$$

Here, the first term preserves the connection of $\mathbf{x}$ to neural network loss function, and the second term enables the optimization of centers w.r.t. the neural network training loss itself. As a result, we obtain an objective function that is continuous everywhere, and for which we can use Lipschitz tools in the convergence analysis – which previous works did not exploit. In fact, we compute the Lipschitz parameters for a specific soft quantization function $\widetilde{Q}_{\mathbf{c}}(\mathbf{x})$ based on *sigmoid* in Appendix A.

**Remark 1.** *It is important to note that with this new objective function, we are able to optimize not only over weights but also over centers. This allows us to theoretically analyze how the movements of the centers affect the convergence. As far as we know, this has not been the case in the literature of quantized neural network training. Moreover, we observe numerically that optimizing over centers improves performance of the network; see Appendix E.*

---

[3] [1] and [33] proposed to approximate the indicator function $\delta_{\mathbf{c}}(\mathbf{x})$ using a distance function $R_{\mathbf{c}}(\mathbf{x})$, where $\mathbf{c}$ is fixed, and unlike ours, it is not a variable that the loss function is optimized over.

[4] In their setup, the quantization centers are fixed. In contrast, we are also optimizing over these centers.

## 2.2 Towards Personalized Quantized Federated Learning: Knowledge Distillation

Note that the objective function defined in (6) can be used for learning a quantized model locally at any client. There are multiple ways to extend that objective for learning personalized quantized models (PQMs) via collaboration. For example, when all clients want to learn personalized models with the *same* dimension (but with different quantization levels), then one natural approach is to add an $\ell_2$ penalty term in the objective that would prevent local models from drifting away from the global model and from simply fitting to local data. This approach, in fact, has been adopted in previous works [7, 12, 22] for learning personalized models in FL, though not quantized ones.[5]

In this paper, since we allow clients to learn PQMs with potentially *different* dimensions, the above approach of adding a $\ell_2$ penalty term in the objective is not feasible. Observe that, the purpose of incorporating a $\ell_2$ penalty in the objective is to ensure that the personalized models do not have significantly different output class scores compared to the global model which is trained using the data generated at all clients; this does not, however, require the global model to have the same dimension as that of local models and can be satisfied by augmenting the local objective (6) with a certain *knowledge distillation* (KD) loss. In our setting, since clients' goal is to learn personalized models with different dimensions that may also have *different* quantization levels, we augment the local objective (6) with two separate KD losses: $f_i^{KD}(\mathbf{x}_i, \mathbf{w})$ and $f_i^{KD}(\widetilde{Q}_{\mathbf{c}_i}(\mathbf{x}_i), \mathbf{w})$, where the first one ensures that the behavior of $\mathbf{x}_i \in \mathbb{R}^{d_i}$ is not very different from that of $\mathbf{w} \in \mathbb{R}^d$, and the second one ensures the same for the quantized version of $\mathbf{x}_i$ and $\mathbf{w}$. Formally, we define them using KL divergence as follows: $f_i^{KD}(\mathbf{x}_i, \mathbf{w}) := D_{KL}(s_i^w(\mathbf{w}) \| s_i(\mathbf{x}_i))$ and $f_i^{KD}(\widetilde{Q}_{\mathbf{c}_i}(\mathbf{x}_i), \mathbf{w}) := D_{KL}(s_i^w(\mathbf{w}) \| s_i(\widetilde{Q}_{\mathbf{c}_i}(\mathbf{x}_i)))$, where $s_i^w$ and $s_i$ denote functions whose inputs are global and personalized models, respectively – and data samples implicitly – and outputs are the softmax classification probabilities of the network.

We need to train $\mathbf{x}_i$ and $\mathbf{w}$ mutually. Identifying the limitations of existing approaches for theoretical analysis (as mentioned in related work in Section 1), we use reverse KL updates (*i.e.*, taking gradient steps w.r.t. the first parameter in $D_{KL}(\cdot, \cdot)$) to train the teacher network $\mathbf{w}$ from the student network $\mathbf{x}_i$. This type of update can be shown to converge and also empirically outperforms [29] (see Section 5). We want to emphasize that though there are works [20, 23] that have used KD in FL and studied its performance (only empirically), ours is the first work that carefully formalizes it as an optimization problem (that also incorporate quantization) which is necessary to analyze convergence properties.

## 3 Centralized Model Quantization Training

In this section, we propose a centralized training scheme (Algorithm 1) for minimizing (6) by optimizing over $\mathbf{x} \in \mathbb{R}^d$ (the model parameters) and $\mathbf{c} \in \mathbb{R}^m$ (quantization values/centers). During training, we keep $\mathbf{x}$ full precision and learn the optimal quantization parameters $\mathbf{c}$. The learned quantization values are then used to hard-quantize the personalized models to get quantized models for deployment in a memory-constrained setting.

**Description of the algorithm.** We optimize (6) through alternating proximal gradient descent. The model parameters and the quantization vector are initialized to random vectors $\mathbf{x}^0$ and $\mathbf{c}^0$. The objective in (6) is composed of two parts: the loss function $f(\mathbf{x}) + f(\widetilde{Q}_{\mathbf{c}}(\mathbf{x}))$ and a quan-

---

**Algorithm 1** Centralized Model Quantization

**Input:** Regularization parameter $\lambda$; initialize the full precision model $\mathbf{x}^0$ and quantization centers $\mathbf{c}^0$; a penalty function enforcing quantization $R(\mathbf{x}, \mathbf{c})$; a soft quantizer $\widetilde{Q}_{\mathbf{c}}(\mathbf{x})$; and learning rates $\eta_1, \eta_2$.

1: **for** $t = 0$ **to** $T - 1$ **do**
2:     Compute $\mathbf{g}^t = \nabla_{\mathbf{x}^t} f(\mathbf{x}^t) + \nabla_{\mathbf{x}^t} f(\widetilde{Q}_{\mathbf{c}^t}(\mathbf{x}^t))$

3:     $\mathbf{x}^{t+1} = \text{prox}_{\eta_1 \lambda R_{\mathbf{c}^t}}(\mathbf{x}^t - \eta_1 \mathbf{g}^t)$

4:     Compute $\mathbf{h}^t = \nabla_{\mathbf{c}^t} f(\widetilde{Q}_{\mathbf{c}^t}(\mathbf{x}^{t+1}))$
5:     $\mathbf{c}^{t+1} = \text{prox}_{\eta_2 \lambda R_{\mathbf{x}^{t+1}}}(\mathbf{c}^t - \eta_2 \mathbf{h}^t)$

6: **end for**

**Output:** Quantized model $\hat{\mathbf{x}}^T = Q_{\mathbf{c}^T}(\mathbf{x}^T)$

---

tization inducing term $R(\mathbf{x}, \mathbf{c})$, which we control by a regularization coefficient $\lambda$. At each $t$, we compute gradient $\mathbf{g}^t$ of the loss function w.r.t. $\mathbf{x}^t$ (line 2), and then take the gradient step followed by $\text{prox}$ step for updating $\mathbf{x}^t$ to $\mathbf{x}^{t+1}$ (line 3). For the centers, we similarly take a gradient step and follow it by a $\text{prox}$ step for updating $\mathbf{c}^t$ to $\mathbf{c}^{t+1}$ (line 4-5). These update steps ensure that we

---

[5]We also analyze this approach in our setting (for learning PQMs but having the *same* dimension), and we call this version QuPeL; see Appendix D for problem setup and convergence results of QuPeL. In Section 5, we demonstrate that QuPeD (for the same task but using KD as opposed to the $\ell_2$ penalty) outperforms QuPeL.

learn the model parameters and quantization vector tied together through proximal[6] mapping of the regularization function $R$. Finally, we quantize the full-precision model $\mathbf{x}^T$ using $Q_{\mathbf{c}^T}$ (line 7).

**Assumptions.** We make the following assumptions on $f$:

**A.1** *(Finite lower bound of $f$):* $f(\mathbf{x}) > -\infty, \forall \mathbf{x} \in \mathbb{R}^d$, which implies $F_\lambda(\mathbf{x}, \mathbf{c}) > -\infty$ for any $\mathbf{x} \in \mathbb{R}^d, \mathbf{c} \in \mathbb{R}^m, \lambda \in \mathbb{R}$.

**A.2** *(Smoothness of $f$):* $f$ is $L$-smooth, i.e., for all $\mathbf{x}, \mathbf{y} \in \mathbb{R}^d$, we have $f(\mathbf{y}) \leq f(\mathbf{x}) + \langle \nabla f(\mathbf{x}), \mathbf{y} - \mathbf{x} \rangle + \frac{L}{2}\|\mathbf{x} - \mathbf{y}\|^2$.

**A.3** *(Bounded gradients of $f$):* $\|\nabla f(\mathbf{x})\|_2 \leq G < \infty, \forall \mathbf{x} \in \mathbb{R}^d$.

**A.1** and **A.2** are standard assumptions for convergence analysis of smooth objectives; and **A.3** is commonly used for non-convex optimization, *e.g.*, for personalized FL in [8]. To make the composite function $f(\widetilde{Q}_{\mathbf{c}}(\mathbf{x}))$ smooth, we need additional assumptions on the soft quantization function $\widetilde{Q}_{\mathbf{c}}(\mathbf{x})$. Our choice of $\widetilde{Q}_{\mathbf{c}}(\mathbf{x})$ (see Appendix A) naturally satisfies these assumptions.

**A.4** *(Smoothness of the soft quantizer):* We assume that $\widetilde{Q}_{\mathbf{c}}(\mathbf{x})$ is $l_{Q_1}$-Lipschitz and $L_{Q_1}$-smooth w.r.t. $\mathbf{x}$, i.e., for $\mathbf{c} \in \mathbb{R}^m$: $\forall \mathbf{x}, \mathbf{y} \in \mathbb{R}^d$: $\|\widetilde{Q}_{\mathbf{c}}(\mathbf{x}) - \widetilde{Q}_{\mathbf{c}}(\mathbf{y})\| \leq l_{Q_1}\|\mathbf{x} - \mathbf{y}\|$ and $\|\nabla_{\mathbf{x}}\widetilde{Q}_{\mathbf{c}}(\mathbf{x}) - \nabla_{\mathbf{y}}\widetilde{Q}_{\mathbf{c}}(\mathbf{y})\| \leq L_{Q_1}\|\mathbf{x} - \mathbf{y}\|$. We also assume $\widetilde{Q}_{\mathbf{c}}(\mathbf{x})$ is $l_{Q_2}$-Lipschitz and $L_{Q_2}$-smooth w.r.t. $\mathbf{c}$, i.e., for $\mathbf{x} \in \mathbb{R}^d$: $\forall \mathbf{c}, \mathbf{d} \in \mathbb{R}^m$: $\|\widetilde{Q}_{\mathbf{c}}(\mathbf{x}) - \widetilde{Q}_{\mathbf{d}}(\mathbf{x})\| \leq l_{Q_2}\|\mathbf{c} - \mathbf{d}\|$ and $\|\nabla_{\mathbf{c}}\widetilde{Q}_{\mathbf{c}}(\mathbf{x}) - \nabla_{\mathbf{d}}\widetilde{Q}_{\mathbf{d}}(\mathbf{x})\| \leq L_{Q_2}\|\mathbf{c} - \mathbf{d}\|$.

**A.5** *(Bound on partial gradients of soft quantizer):* There exists constants $G_{Q_1}, G_{Q_2} < \infty$ such that: $\|\nabla_{\mathbf{x}}\widetilde{Q}_{\mathbf{c}}(\mathbf{x})\|_F = \|\nabla\widetilde{Q}_{\mathbf{c}}(\mathbf{x})_{1:d,:}\|_F \leq G_{Q_1}$ and $\|\nabla_{\mathbf{c}}\widetilde{Q}_{\mathbf{c}}(\mathbf{x})\|_F = \|\nabla\widetilde{Q}_{\mathbf{c}}(\mathbf{x})_{d+1:d+m,:}\|_F \leq G_{Q_2}$, where $\mathbf{X}_{p:q,:}$ denotes sub-matrix of $\mathbf{X}$ with rows between $p$ and $q$, and $\|\cdot\|_F$ is the Frobenius norm.

**Convergence result.** Now we state our main convergence result (proved in Appendix B) for minimizing $F_\lambda(\mathbf{x}, \mathbf{c})$ in (6) w.r.t. $(\mathbf{x}, \mathbf{c}) \in \mathbb{R}^{d+m}$ via Algorithm 1. This provides first-order guarantees for convergence of $(\mathbf{x}, \mathbf{c})$ to a stationary point and recovers the $\mathcal{O}\left(1/T\right)$ convergence rate of [1, 3].

**Theorem 1.** *Consider running Algorithm 1 for $T$ iterations for minimizing (6) with $\eta_1 = 1/2(L+GL_{Q_1}+G_{Q_1}Ll_{Q_1})$ and $\eta_2 = 1/2(GL_{Q_2}+G_{Q_2}Ll_{Q_2})$. For any $t \in [T]$, define $\mathbf{G}^t := [\nabla_{\mathbf{x}^{t+1}}F_\lambda\left(\mathbf{x}^{t+1}, \mathbf{c}^t\right)^T, \nabla_{\mathbf{c}^{t+1}}F_\lambda\left(\mathbf{x}^{t+1}, \mathbf{c}^{t+1}\right)^T]^T$. Then, under **A.1-A.5** and for $L_{\min} = \min\{\frac{1}{\eta_1}, \frac{1}{\eta_2}\}$, $L_{\max} = \max\{\frac{1}{\eta_1}, \frac{1}{\eta_2}\}$, we have:*

$$\frac{1}{T}\sum_{t=0}^{T-1}\|\mathbf{G}^t\|_2^2 = \mathcal{O}\left(\frac{L_{\max}^2\left(F_\lambda\left(\mathbf{x}^0, \mathbf{c}^0\right) - F_\lambda(\mathbf{x}^T, \mathbf{c}^T)\right)}{L_{\min}T}\right).$$

**Remark 2.** *In Theorem 1, we see that gradient norm decays without any constant error terms. The convergence rate depends on Lipschitz smoothness constants of $f$ and $f(\widetilde{Q}_{\mathbf{c}}(.))$ through $L_{\max}$ and $L_{\min}$. Choosing a smoother $\widetilde{Q}_{\mathbf{c}}(.)$ would speed up convergence; however, if chosen too small, this could result in an accuracy loss when hard-quantizing the parameters at the end of the algorithm.*

**Remark 3** (Number of centers and convergence)**.** *The number of quantization levels $m$ has a direct effect on convergence through the soft quantization function $\widetilde{Q}_{\mathbf{c}}(\mathbf{x})$ and Lipschitz constants. Note that as $m \to \infty$, we have $\widetilde{Q}_{\mathbf{c}}(\mathbf{x}) \to \mathbf{x}, \forall \mathbf{x}$. In this case, $\widetilde{Q}_{\mathbf{c}}(\mathbf{x})$ is $0$-smooth and $1$-Lipschitz w.r.t. all parameters. As a result, we would have $L_{\min} = L$ and $L_{\max} = 2L$. Note that the ratio $L_{\max}^2/L_{\min}$ increases as the quantization becomes more aggressive, and consequently, affects the convergence.*

## 4  Personalized Quantization for FL via Knowledge Distillation

We now consider the FL setting where we aim to learn quantized and personalized models for each client with different precision and model dimensions in heterogeneous data setting. Our proposed method QuPeD (Algorithm 2), utilizes the centralized scheme of Algorithm 1 locally at each client to minimize (3) over $\left(\{\mathbf{x}_i, \mathbf{c}_i\}_{i=1}^n, \mathbf{w}\right)$. Here, $\mathbf{x}_i, \mathbf{c}_i$, denote the model parameters and the quantization vector (centers) for client $i$, and $\mathbf{w}$ denotes the global model that facilitates collaboration among clients which is encouraged through the knowledge distillation (KD) loss in the local objectives (2).

---

[6]As a short notation, we use $\text{prox}_{\eta_1\lambda R_{\mathbf{c}^t}}$ to denote $\text{prox}_{\eta_1\lambda R(\cdot, \mathbf{c}^t)}$, and $\text{prox}_{\eta_2\lambda R_{\mathbf{x}^{t+1}}}$ for $\text{prox}_{\eta_2\lambda R(\mathbf{x}^{t+1}, \cdot)}$.

---

**Algorithm 2** QuPeD: Quantized Personalization via Distillation

---

**Input:** Regularization parameters $\lambda, \lambda_p$; synchronization gap $\tau$; for client $i \in [n]$, initialize full precision personalized models $\mathbf{x}_i^0$, quantization centers $\mathbf{c}_i^0$, local model $\mathbf{w}_i^0$, learning rates $\eta_1^{(i)}, \eta_2^{(i)}, \eta_3$; quantization enforcing penalty function $R(\mathbf{x}, \mathbf{c})$; soft quantizer $\widetilde{Q}_{\mathbf{c}}(\mathbf{x})$; number of clients to be sampled $K$.

1: **for** $t = 0$ **to** $T - 1$ **do**
2:   **if** $\tau$ divides $t$ **then**
3:     **On Server do:**
        Choose a subset of clients $\mathcal{K}_t \subseteq [n]$ with size $K$
4:     Broadcast $\mathbf{w}^t$ to all **Clients**
5:     **On Clients** $i \in \mathcal{K}_t$ **to** $n$ (in parallel) **do**:
6:       Receive $\mathbf{w}^t$ from **Server**; set $\mathbf{w}_i^t = \mathbf{w}^t$
7:   **end if**
8:   **On Clients** $i \in \mathcal{K}_t$ **to** $n$ (in parallel) **do**:
9:     Compute $\mathbf{g}_i^t := (1 - \lambda_p)(\nabla_{\mathbf{x}_i^t} f_i(\mathbf{x}_i^t) + \nabla_{\mathbf{x}_i^t} f_i(\widetilde{Q}_{\mathbf{c}_i^t}(\mathbf{x}_i^t))) + \lambda_p(\nabla_{\mathbf{x}_i^t} f_i^{KD}(\mathbf{x}_i^t, \mathbf{w}_i^t) + \nabla_{\mathbf{x}_i^t} f_i^{KD}(\widetilde{Q}_{\mathbf{c}_i^t}(\mathbf{x}_i^t), \mathbf{w}_i^t))$
10:    $\mathbf{x}_i^{t+1} = \text{prox}_{\eta_1^{(i)} \lambda R_{\mathbf{c}_i^t}}(\mathbf{x}_i^t - \eta_1^{(i)} \mathbf{g}_i^t)$
11:    Compute $\mathbf{h}_i^t := (1 - \lambda_p) \nabla_{\mathbf{c}_i^t} f_i(\widetilde{Q}_{\mathbf{c}_i^t}(\mathbf{x}_i^{t+1})) + \lambda_p \nabla_{\mathbf{c}_i^t} f_i^{KD}(\widetilde{Q}_{\mathbf{c}_i^t}(\mathbf{x}_i^{t+1}), \mathbf{w}_i^t)$
12:    $\mathbf{c}_i^{t+1} = \text{prox}_{\eta_2^{(i)} \lambda R_{\mathbf{x}_i^{t+1}}}(\mathbf{c}_i^t - \eta_2^{(i)} \mathbf{h}_i^t)$
13:    $\mathbf{w}_i^{t+1} = \mathbf{w}_i^t - \eta_3 \lambda_p(\nabla_{\mathbf{w}_i^t} f_i^{KD}(\mathbf{x}_i^{t+1}, \mathbf{w}_i^t) + \nabla_{\mathbf{w}_i^t} f_i^{KD}(\widetilde{Q}_{\mathbf{c}_i^{t+1}}(\mathbf{x}_i^{t+1}), \mathbf{w}_i^t))$
14:    **if** $\tau$ divides $t + 1$ **then**
15:      Clients send $\mathbf{w}_i^t$ to **Server**
16:      Server receives $\{\mathbf{w}_i^t\}$; computes $\mathbf{w}^{t+1} = \frac{1}{K} \sum_{i \in \mathcal{K}_t}^n \mathbf{w}_i^t$
17:    **end if**
18: **end for**
19: $\hat{\mathbf{x}}_i^T = Q_{\mathbf{c}_i^T}(\mathbf{x}_i^T)$ for all $i \in [n]$

**Output:** Quantized personalized models $\{\hat{\mathbf{x}}_i^T\}_{i=1}^n$

---

**Description of the algorithm.** Since clients perform local iterations, apart from maintaining $\mathbf{x}_i^t, \mathbf{c}_i^t$ at each client $i \in [n]$, it also maintains a model $\mathbf{w}_i^t$ that helps in utilizing other clients' data via collaboration. We call $\{\mathbf{w}_i^t\}_{i=1}^n$ local copies of the global model at clients at time $t$. At each iteration $t$ server samples a subset of all indices $\mathcal{K}_t \subseteq [n]$ with $K \leq n$ elements. Client $i$ updates $\mathbf{w}_i^t$ in between communication rounds based on its local data and synchronizes that with the server which aggregates them to update the global model. Note that the local objective in (2) can be split into the weighted average of loss functions $(1 - \lambda_p)(f_i(\mathbf{x}_i) + f_i(\widetilde{Q}_{\mathbf{c}_i}(\mathbf{x}_i))) + \lambda_p(f_i^{KD}(\mathbf{x}_i, \mathbf{w}_i) + f_i^{KD}(\widetilde{Q}_{\mathbf{c}_i}(\mathbf{x}_i), \mathbf{w}_i))$ and the term enforcing quantization $\lambda R(\mathbf{x}_i, \mathbf{c}_i)$. At any iteration $t$ that is not a communication round (line 3), if $i \in \mathcal{K}_t$, client $i$ first computes the gradient $\mathbf{g}_i^t$ of the loss function w.r.t. $\mathbf{x}_i^t$ (line 4) and then takes a gradient step followed by the proximal step using $R$ (line 5) to update from $\mathbf{x}_i^t$ to $\mathbf{x}_i^{t+1}$. Then it computes the gradient $\mathbf{h}_i^t$ of the loss function w.r.t. $\mathbf{c}_i^t$ (line 6) and updates the centers followed by the proximal step (line 7). Finally, it updates $\mathbf{w}_i^t$ to $\mathbf{w}_i^{t+1}$ by taking a gradient step of the loss function at $\mathbf{w}_i^t$ (line 8). Thus, the local training of $\mathbf{x}_i^t, \mathbf{c}_i^t$ also incorporates knowledge from other clients' data through $\mathbf{w}_i^t$. When $t$ is divisible by $\tau$, sampled clients upload $\{\mathbf{w}_i^t\}$ to the server (line 10) which aggregates them (line 15) and broadcasts the updated global model (line 16) to all the clients. At the end of training, clients learn their personalized models $\{\mathbf{x}_i^T\}_{i=1}^n$ and quantization centers $\{\mathbf{c}_i^T\}_{i=1}^n$. Finally, client $i$ quantizes $\mathbf{x}_i^T$ using $Q_{\mathbf{c}_i^T}$ (line 19).

**Assumptions.** In addition to assumptions **A.1 - A.5** (with **A.3** and **A.5** modified to have client specific gradient bounds $\{G^{(i)}, G_{Q_1}^{(i)}, G_{Q_2}^{(i)}\}$ as they have different model dimensions[7]), we assume:

**A.6** (*Bounded diversity*): At any $t \in \{0, \cdots, T - 1\}$ and any client $i \in [n]$, the variance of the local gradient (at client $i$) w.r.t. the global gradient is bounded, i.e., there exists $\kappa_i < \infty$, such that for every $\{\mathbf{x}_i^{t+1} \in \mathbb{R}^d, \mathbf{c}_i^{t+1} \in \mathbb{R}^{m_i} : i \in [n]\}$ and $\mathbf{w}^t \in \mathbb{R}^d$ generated according to Algorithm 2, we have:

---

[7]We keep smoothness constants to be the same across clients for notational simplicity, however, our result can easily be extended to that case.

$\|\nabla_{\mathbf{w}^t} F_i(\mathbf{x}_i^{t+1}, \mathbf{c}_i^{t+1}, \mathbf{w}^t) - \frac{1}{n}\sum_{j=1}^n \nabla_{\mathbf{w}^t} F_j(\mathbf{x}_j^{t+1}, \mathbf{c}_j^{t+1}, \mathbf{w}^t)\|^2 \leq \kappa_i$. This assumption is equivalent to the bounded diversity assumption in [7, 8]; see Appendix A.

**A.7** *(Smoothness of $f^{KD}$):* We assume $f_i^{KD}(\mathbf{x}, \mathbf{w})$ is $L_{D_1}$-smooth w.r.t. $\mathbf{x}$, $L_{D_2}$-smooth w.r.t. $\mathbf{w}$ for all $i \in [n]$; as a result it is $L_D$-smooth w.r.t. $[\mathbf{x}, \mathbf{w}]$ where $L_D = \max\{L_{D_1}, L_{D_2}\}$. Furthermore, we assume $f_i^{KD}(\widetilde{Q}_{\mathbf{c}}(\mathbf{x}), \mathbf{w})$ is $L_{DQ_1}$-smooth w.r.t. $\mathbf{x}$, $L_{DQ_2}$-smooth w.r.t. $\mathbf{c}$ and $L_{DQ_3}$-smooth w.r.t. $\mathbf{w}$ for all $i \in [n]$; as a result it is $L_{DQ}$-smooth w.r.t. $[\mathbf{x}, \mathbf{c}, \mathbf{w}]$ where $L_{DQ} = \max\{L_{DQ_1}, L_{DQ_2}, L_{DQ_3}\}$. In Appendix A we discuss how this assumption can actually be inferred from previous assumptions.

**Convergence result.** In Theorem 2 we present the convergence result when there is full client participation, i.e. $K = n$, in Appendix C we discuss the modification in convergence result under client sampling. The following result (proved in Appendix C) achieves a rate of $\mathcal{O}\left(1/\sqrt{T}\right)$ for finding a stationary point within an error that depends on the data heterogeneity, matching result in [7]:

**Theorem 2.** *Under assumptions* **A.1-A.7**, *consider running Algorithm 2 for $T$ iterations for minimizing* (3) *with* $\tau \leq \sqrt{T}$, $\eta_1^{(i)} = 1/2(\lambda_p(2+L_{D_1}+L_{DQ_1})+(1-\lambda_p)(L+G^{(i)}L_{Q_1}+G_{Q_1}^{(i)}Ll_{Q_1}))$, $\eta_2^{(i)} = 1/2(\lambda_p(1+L_{DQ_2})+(1-\lambda_p)(G^{(i)}L_{Q_2}+G_{Q_2}^{(i)}Ll_{Q_2}))$, *and* $\eta_3 = 1/4(\lambda_p L_w \sqrt{C_L}\sqrt{T})$ *where* $L_w = L_{D_2} + L_{DQ_3}$. *Let* $\mathbf{G}_i^t := [\nabla_{\mathbf{x}_i^{t+1}} F_i(\mathbf{x}_i^{t+1}, \mathbf{c}_i^t, \mathbf{w}^t)^T, \nabla_{\mathbf{c}_i^{t+1}} F_i(\mathbf{x}_i^{t+1}, \mathbf{c}_i^t, \mathbf{w}^t)^T, \nabla_{\mathbf{w}^t} F_i(\mathbf{x}_i^{t+1}, \mathbf{c}_i^{t+1}, \mathbf{w}^t)^T]^T$. *Then*

$$\frac{1}{T}\sum_{t=0}^{T-1}\frac{1}{n}\sum_{i=1}^n \|\mathbf{G}_i^t\|^2 = \mathcal{O}\left(\frac{\tau^2\overline{\kappa}+\overline{\Delta}_F}{\sqrt{T}} + \tau^2\overline{\kappa}\left(\frac{C_1}{T}+\frac{C_2}{T^{\frac{3}{2}}}\right) + \overline{\kappa}\right),$$

*for some constants $C_1, C_2$, where* $\overline{\Delta}_F = \frac{1}{n}\sum_{i=1}^n (L_{\max}^{(i)})^2 \left(F_i(\mathbf{x}_i^0, \mathbf{c}_i^0, \mathbf{w}_i^0) - F_i(\mathbf{x}_i^T, \mathbf{c}_i^T, \mathbf{w}_i^T)\right)$, $\overline{\kappa} = \frac{1}{n}\sum_{i=1}^n (L_{\max}^{(i)})^2\kappa_i$, $C_L = 1 + \frac{\frac{1}{n}\sum_{i=1}^n (L_{\max}^{(i)})^2}{(\min_i\{L_{\max}^{(i)}\})^2}$, *and $L_{\max}^{(i)}$ depends on the smoothness parameters and the gradient bound at client $i$ – the expression can be found in Appendix C.*

**Remark 4** (Resource and data heterogeneity.)**.** *Firstly, our observation from Remark 3 holds here as well. Aggressive quantization has a scaling effect on all the terms through $L_{\max}^{(i)}$. Now the interesting question is: how does having different model structures across clients affect the convergence rate of Theorem 2? Note that in Assumptions* **A.3, A.5**, *we assume clients have different client-specific gradient bounds; this results in client specific $L_{\max}^{(i)}$, and consequently $\overline{\kappa}$, which couples resource and data heterogeneity across clients. Here we make an important* first *observation regarding the coupled effect of data and resource heterogeneity on the convergence rate. Suppose data distributions are fixed across clients (i.e., $\kappa_i$'s are fixed) and we need to choose models for each client in the federated ecosystem. Then, for a fast convergence, for the clients that have local data that is not a representative of the general distribution (large $\kappa_i$), it is critical to choose models with small smoothness parameter (e.g., choosing a less aggressive quantization); whereas, clients with data that is representative of the overall data distribution (small $\kappa_i$) can tolerate having a less smooth model.*

## 5 Experiments

In this section, we first briefly compare numerical results for our underlying model quantization scheme (Algorithm 1) in a centralized case against related works [1, 33]. For a major part of the section, we then compare QuPeD (Algorithm 2) against other personalization schemes [7, 8, 29] for data heterogeneous clients and demonstrate its effectiveness in resource heterogeneous environments. Additional experimental results on a language task are provided in Appendix E. Implementation details, and further experiments with modifications to settings in this section, are also in Appendix E.

**Centralized Training:** We compare Algorithm 1 with [1, 33] for ResNet-20 and ResNet-32 [13] models trained on CIFAR-10 [18] dataset.

While both [1, 33] are limited to binary quantization, [1] can be seen as a specific case of our centralized method where the centers do not get updated. From Table 1, we see that updating centers (Algorithm 1) significantly improves the performance (0.48% increase in test accuracy). Allowing quantization with 2 bits instead of 1bit for Algorithm 1 further increases the test accuracy.

**Table 1** Test accuracy (in %) on CIFAR-10.

| Method | ResNet-20 | ResNet-32 |
|---|---|---|
| Full Precision (FP) | 92.05 | 92.95 |
| ProxQuant [1] (1bit) | 90.69 | 91.55 |
| BinaryRelax [33] (1bit) | 87.82 | 90.65 |
| Algorithm1 (1bit) | 91.17 | 92.20 |
| Algorithm1 (2bits) | 91.45 | 92.47 |

**Table 2** Test accuracy (in %) for CNN1 model at all clients.[7]

| Method | FEMNIST | CIFAR-10 |
|---|---|---|
| FedAvg (FP) | $94.92 \pm 0.04$ | $61.40 \pm 0.29$ |
| Local Training (FP) | $94.86 \pm 0.93$ | $71.57 \pm 0.28$ |
| Local Training (2 Bits) | $93.95 \pm 0.23$ | $70.87 \pm 0.15$ |
| Local Training (1 Bit) | $93.00 \pm 0.50$ | $69.05 \pm 0.13$ |
| **QuPeD** (FP) | $\mathbf{97.31} \pm 0.12$ | $\mathbf{75.06} \pm 0.40$ |
| **QuPeD** (2 Bits) | $96.73 \pm 0.27$ | $74.58 \pm 0.44$ |
| **QuPeD** (1 Bit) | $95.15 \pm 0.21$ | $71.20 \pm 0.33$ |
| QuPeL (2 Bits) | $96.10 \pm 0.14$ | $73.52 \pm 0.51$ |
| QuPeL (1 Bits) | $94.06 \pm 0.28$ | $71.01 \pm 0.32$ |
| pFedMe (FP) [7] | $96.60 \pm 0.37$ | $73.66 \pm 0.65$ |
| Per-FedAvg (FP) [8] | $97.16 \pm 0.21$ | $74.15 \pm 0.41$ |
| Federated ML (FP) [29] | $96.32 \pm 0.32$ | $74.34 \pm 0.30$ |

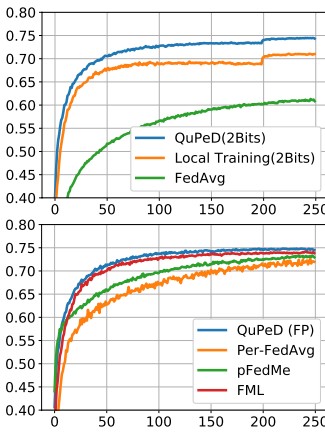

**Figure 1** Test Acc. vs epoch (CIFAR-10)

**Personalized Training:** We consider an image classification task on FEMNIST [4] and CIFAR-10 [18] datasets. We consider two CNN architectures: (i) CNN1 (used in [26]): has 2 convolutional and 3 fully connected layers, (ii) CNN2: this is CNN1 with an additional convolutional layer with 32 filters and $5 \times 5$ kernel size. For CIFAR-10 we choose a batch size of 25. For FEMNIST, we choose variable batch sizes to have 60 iterations for all clients per epoch. We train each algorithm for 250 epochs on CIFAR-10 and 30 epochs on FEMNIST. For quantized training, as standard practice [28], we let the first and last layers of networks to be in full precision. We use last 50 epochs on CIFAR-10, and 5 epochs on FEMNIST for the fine-tuning phase.

*Data Heterogeneity (DH)*: We consider $n = 50$ clients for CIFAR-10 and $n = 66$ for FEMNIST. To simulate data heterogeneity on CIFAR-10, similar to [26], we allow each client to have access to data samples from only 4 randomly chosen classes. Thus, each client has 1000 training samples and 200 test samples. On FEMNIST, we use a subset of 198 writers from the dataset and distribute the data so that each client has access to data samples written by 3 randomly chosen writers. The number of training samples per client varies between 203-336 and test samples per client varies between 25-40. Test samples are sampled from the same class/writer that training samples are sampled from, in parallel with previous works in heterogeneous FL.

**Table 3** Test accuracy (in %) for CNN1 model at all clients, with client sampling.

| Method | MNIST ($\frac{K}{n} = 0.1$) | FEMNIST($\frac{K}{n} = \frac{1}{3}$) |
|---|---|---|
| FedAvg (FP) | $92.87 \pm 0.05$ | $91.30 \pm 0.43$ |
| **QuPeD** (FP) | $\mathbf{98.17} \pm 0.32$ | $\mathbf{94.93} \pm 0.25$ |
| **QuPeD** (2 Bits) | $98.01 \pm 0.15$ | $94.56 \pm 0.18$ |
| **QuPeD** (1 Bit) | $97.58 \pm 0.23$ | $92.52 \pm 0.64$ |
| pFedMe (FP) [7] | $97.79 \pm 0.03$ | $93.70 \pm 0.39$ |
| Per-FedAvg (FP) [8] | $95.80 \pm 0.29$ | $92.10 \pm 0.22$ |
| Federated ML (FP) [29] | $98.03 \pm 0.31$ | $92.73 \pm 0.36$ |

*Resource Heterogeneity (RH)*: To simulate resource heterogeneity for QuPeD, we consider 4 settings: (i) half of the clients have CNN1 in full precision (FP) and the other half CNN2 in FP, (ii) half of the clients have CNN1 in 2 bits and the other half CNN2 in FP, (iii) half of the clients have CNN1 in 2 bits and the other in FP, (iv) half of the clients have CNN1 in 2 bits and the other half CNN2 in 2 bits.

*Results (DH):* We compare QuPeD against FedAvg [26], local training of clients (without any collaboration), and personalized FL methods: pFedMe [7], Per-FedAvg [8], Federated Mutual Learning [29], and QuPeL (Footnote 5). For all methods, if applicable, we set $\tau = 10$ local iterations, use learning rate decay 0.99 and use weight decay of $10^{-4}$; we fine tune the initial learning rate for each method independently, see Appendix E for details. The results are provided in Table 2 with full client participation ($K = n$), plotted in Figure 1 for CIFAR-10, and in Table 3 with client sampling where we state average results over 3 runs; all clients train CNN1 (see Appendix E for CNN2) and quantization values are indicated in parenthesis. Thus, we only consider model personalization for

---

[7]Here QuPeD (FP) corresponds to changing alternating proximal gradient updates with SGD update on model parameters in Algorithm 2, Local Training (FP) corresponds to SGD updates without communication.

**Table 4** Test accuracy (in %) on FEMNIST and CIFAR-10 for heterogeneous resource distribution among clients.

| Resource Heterogeneity | FEMNIST | | CIFAR-10 | |
|---|---|---|---|---|
| | Local Training | QuPeD | Local Training | QuPeD |
| CNN1(FP) + CNN2(FP) | $93.41 \pm 0.82$ | $97.44 \pm 0.14$ | $72.81 \pm 0.03$ | $75.50 \pm 0.25$ |
| CNN1(2 Bits)+CNN2(FP) | $92.70 \pm 1.09$ | $97.01 \pm 0.05$ | $72.42 \pm 0.17$ | $75.08 \pm 0.18$ |
| CNN1(2 Bits)+CNN1(FP) | $93.56 \pm 0.38$ | $96.96 \pm 0.15$ | $71.23 \pm 0.08$ | $74.84 \pm 0.30$ |
| CNN1(2 Bits)+CNN2(2 Bits) | $91.11 \pm 0.23$ | $96.64 \pm 0.31$ | $72.15 \pm 0.47$ | $74.64 \pm 0.27$ |

data heterogeneity. In Table 2, we observe that full precision QuPeD consistently outperforms all other methods for both datasets. Furthermore, we observe QuPeD with 2-bit quantization is the second best performing method on CIFAR-10 (after QuPeD (FP)) and third best performing method on FEMNIST despite the loss due to quantization. Hence, QuPeD is highly effective for quantized training in data heterogeneous settings. Since QuPeD outperforms QuPeL, we can also (empirically) claim that considering KD loss to encourage collaboration is superior to $\ell_2$ distance loss. Lastly, we observe from Table 3 that QuPeD continues to outperform other methods under client sampling.

*Results (DH+RH)*: We now discuss personalized FL setting with both data and resource heterogeneity across clients. Note that since FedAvg, pFedMe, and Per-FedAvg cannot work in settings where clients have different model dimensions, we only provide comparisons of QuPeD with local training (no collaboration) to demonstrate its effectiveness. The results are given in Table 4. We observe that QuPeD (collaborative training) significantly outperforms local training in all cases (about $3.5\%$ or higher on FEMNIST and $2.5\%$ or higher on CIFAR-10) and works remarkably well even in cases where clients have quantized models without any significant loss in performance.

## 6 Discussion

In this work, we introduced QuPeD: an algorithm for training of quantized and personalized models in heterogeneous Federated Learning (FL) settings. Our proposed scheme tackles two of the major problems in practical FL applications: resource and data heterogeneity among participating clients. QuPeD allows clients to learn personalized models based on their local data distributions while simultaneously allowing these models to be different in architecture and weight precision. This is unlike existing personalized FL algorithms in literature which only consider the data heterogeneity aspect of FL. Our theoretical results on the convergence rates recovers previous results, which are in simpler settings, when the clients have identical resources, i.e., same model size and precision. When clients have distinct resources, our result ties convergence speed to combination of resource and data heterogeneity revealing additional insights. Our experimental results demonstrate that QuPeD delivers superior empirical performance even under aggressive quantization.

Convergence results stated in Theorems 1, 2, use bounded gradients assumptions **A.3, A.5** for non-convex objectives considered in this paper. Incorporating recent progress on weakening gradient assumptions in convergence analyses for FL (*e.g.,* [16, 30] and references therein) into our framework is part of future work. We remark that our current assumptions are fairly standard and applicable to our new optimization formulation to tackle data and resource heterogeneity of the clients, which is the principal focus and contribution of our work. Simulations consisting of tens of thousands (or more) of clients and lower client sampling ratios are also crucial, and are left for potential future investigations that can shed further light on our methods at massive scale.

*Societal Impact:* This paper considers collaborative personalization in learning, which can significantly improve learning performance. However, such personalization is only as good as the data used for training, and if not used properly could lead to information bubbles and disjointed views for each client, *e.g.,* clustering methods that put together clients with similar data statistics. We ameliorate that by not clustering similar data and models, but this is an aspect that might need more examination for social consequences. We explicitly consider diverse resources in our designs, potentially enabling clients with fewer resources (devices with less capabilities) to benefit from richer resources; potentially a positive impact in using resources more equitably.

## 7 Acknowledgments

This work was partially supported by NSF grants #2007714 and #1955632, and by Army Research Laboratory under Cooperative Agreement W911NF-17-2-0196. The views and conclusions contained in this document are those of the authors and should not be interpreted as representing the official policies, either expressed or implied, of the Army Research Laboratory or the U.S. Government. The U.S. Government is authorized to reproduce and distribute reprints for Government purposes notwithstanding any copyright notation here on. We also thank the reviewers for detailed inputs that helped improve the presentation of our results.

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
