# A Preliminaries

## A.1 Notation

- Given a composite function $g(\mathbf{x}, \mathbf{y})$ we will denote $\nabla g(\mathbf{x}, \mathbf{y})$ or $\nabla_{(\mathbf{x}, \mathbf{y})} g(\mathbf{x}, \mathbf{y})$ as the gradient; $\nabla_{\mathbf{x}} g(\mathbf{x}, \mathbf{y})$ and $\nabla_{\mathbf{y}} g(\mathbf{x}, \mathbf{y})$ as the partial gradients with respect to $\mathbf{x}$ and $\mathbf{y}$.

- For a vector $\mathbf{u}$, $\|\mathbf{u}\|$ denotes the $\ell_2$-norm $\|\mathbf{u}\|_2$. For a matrix $\mathbf{A}$, $\|\mathbf{A}\|_F$ denotes the Frobenius norm.

- Unless otherwise stated, for a given vector $\mathbf{x}$, $x_i$ denotes the $i'th$ element in vector $\mathbf{x}$; and $\mathbf{x}_i$ denotes that the vector belongs to client $i$. Furthermore, $\mathbf{x}_i^t$ denotes a vector that belongs to client $i$ at time $t$.

## A.2 Equivalence of Assumption A.6 to Assumptions in Related Work

In particular, diversity assumption (Assumption 5) in [8] is as follows:

$$\frac{1}{n} \sum_{i=1}^{n} \left\| \nabla_{\mathbf{w}} f_i(\mathbf{w}) - \nabla_{\mathbf{w}} f(\mathbf{w}) \right\|^2 \leq B,$$

where $f_i$ is local function, $B$ is a constant and $\nabla_{\mathbf{w}} f(\mathbf{w}) = \frac{1}{n} \sum_{j=1}^{n} \nabla_{\mathbf{w}} f_i(\mathbf{w})$. Now we will show the equivalence to our stated assumption **A.6**. Let us define,

$$x_i(\mathbf{w}^t) := \underset{\mathbf{x} \in \mathbb{R}^d}{\arg\min} \left\{ \left\langle \mathbf{x} - \mathbf{x}_i^t, \nabla f_i\left(\mathbf{x}_i^t\right) \right\rangle + \left\langle \mathbf{x} - \mathbf{x}_i^t, \nabla_{\mathbf{x}_i^t} f_i(\widetilde{Q}_{\mathbf{c}_i^t}(\mathbf{x}_i^t)) \right\rangle + \left\langle \mathbf{x} - \mathbf{x}_i^t, \lambda_p(\mathbf{x}_i^t - \mathbf{w}^t) \right\rangle \right.$$

$$\left. + \frac{1}{2\eta_1} \left\| \mathbf{x} - \mathbf{x}_i^t \right\|_2^2 + \lambda R(\mathbf{x}, \mathbf{c}_i^t) \right\}$$

$$c_i(\mathbf{w}^t) := \underset{\mathbf{c} \in \mathbb{R}^m}{\arg\min} \left\{ \left\langle \mathbf{c} - \mathbf{c}_i^t, \nabla_{\mathbf{c}_i^t} f_i(\widetilde{Q}_{\mathbf{c}_i^t}(x_i(\mathbf{w}^t))) \right\rangle + \frac{1}{2\eta_2} \left\| \mathbf{c} - \mathbf{c}_i^t \right\|_2^2 + \lambda R(x_i(\mathbf{w}^t), \mathbf{c}) \right\}$$

Then we can define,

$$\psi_i(x_i(\mathbf{w}^t), c_i(\mathbf{w}^t), \mathbf{w}^t) := F_i(\mathbf{x}_i^{t+1}, \mathbf{c}_i^{t+1}, \mathbf{w}^t)$$

as a result, we can further define $g_i(\mathbf{w}^t) := \psi_i(x_i(\mathbf{w}^t), c_i(\mathbf{w}^t), \mathbf{w}^t)$. Therefore, our assumption **A.6** is equivalent to stating the following assumption: At any $t \in [T]$ and any client $i \in [n]$, the variance of the local gradient (at client $i$) w.r.t. the global gradient is bounded, i.e., there exists $\kappa_i < \infty$, such that for every $\mathbf{w}^t \in \mathbb{R}^d$, we have:

$$\left\| \nabla_{\mathbf{w}^t} g_i(\mathbf{w}^t) - \frac{1}{n} \sum_{j=1}^{n} \nabla_{\mathbf{w}^t} g_j(\mathbf{w}^t) \right\|^2 \leq \kappa_i,$$

And we also define $\kappa := \frac{1}{n} \sum_{i=1}^{n} \kappa_i$ and then,

$$\frac{1}{n} \sum_{i=1}^{n} \left\| \nabla_{\mathbf{w}^t} g_i(\mathbf{w}^t) - \nabla_{\mathbf{w}^t} g(\mathbf{w}^t) \right\|^2 \leq \kappa,$$

here $\nabla_{\mathbf{w}^t} g(\mathbf{w}^t) = \frac{1}{n} \sum_{j=1}^{n} \nabla_{\mathbf{w}^t} g_j(\mathbf{w}^t)$. Hence, our assumption is equivalent to assumptions that are found in aforementioned works.

## A.3 Alternating Proximal Steps

We define the following functions: $f : \mathbb{R}^d \to \mathbb{R}, \widetilde{Q} : \mathbb{R}^{d+m} \to \mathbb{R}^d$, and we also define $h(\mathbf{x}, \mathbf{c}) = f(\widetilde{Q}_{\mathbf{c}}(\mathbf{x})), h : \mathbb{R}^{d+m} \to \mathbb{R}$ where $\mathbf{x} \in \mathbb{R}^d$ and $\mathbf{c} \in \mathbb{R}^m$. Note that here $\widetilde{Q}_{\mathbf{c}}(\mathbf{x})$ denotes $\widetilde{Q}(\mathbf{x}, \mathbf{c})$. Throughout our paper we will use $\widetilde{Q}_{\mathbf{c}}(\mathbf{x})$ to denote $\widetilde{Q}(\mathbf{x}, \mathbf{c})$, in other words both $\mathbf{c}$ and $\mathbf{x}$ are inputs to the function $\widetilde{Q}_{\mathbf{c}}(\mathbf{x})$. We propose an alternating proximal gradient algorithm. Our updates are as follows:

$$\mathbf{x}^{t+1} = \text{prox}_{\eta_1 \lambda R(\cdot, \mathbf{c}^t)}(\mathbf{x}^t - \eta_1 \nabla f(\mathbf{x}^t) - \eta_1 \nabla_{\mathbf{x}^t} f(\widetilde{Q}_{\mathbf{c}^t}(\mathbf{x}^t))) \tag{1}$$

$$\mathbf{c}^{t+1} = \text{prox}_{\eta_2 \lambda R(\mathbf{x}^{t+1}, \cdot)}(\mathbf{c}^t - \eta_2 \nabla_{\mathbf{c}^t} f(\widetilde{Q}_{\mathbf{c}^t}(\mathbf{x}^{t+1})))$$

For simplicity we assume the functions in the objective function are differentiable, however, our analysis could also be done using subdifferentials.

Our method is inspired by [3] where the authors introduce an alternating proximal minimization algorithm to solve a broad class of non-convex problems as an alternative to coordinate descent methods. In this work we construct another optimization problem that can be used as a surrogate in learning quantized networks where both model parameters and quantization levels are subject to optimization. In particular, [3] considers a general objective function of the form $F(\mathbf{x}, \mathbf{y}) = f(\mathbf{x}) + g(\mathbf{y}) + \lambda H(\mathbf{x}, \mathbf{y})$, whereas, our objective function is tailored for learning quantized networks: $F_\lambda(\mathbf{x}, \mathbf{c}) = f(\mathbf{x}) + f(\widetilde{Q}_{\mathbf{c}}(\mathbf{x})) + \lambda R(\mathbf{x}, \mathbf{c})$. Furthermore, they consider updates where the proximal mappings are with respect to functions $f, g$, whereas in our case the proximal mappings are with respect to the distance function $R(\mathbf{x}, \mathbf{c})$ to capture the soft projection.

### A.4 A Soft Quantization Function

In this section we give an example of the soft quantization function that can be used in previous sections. In particular, we can define the following soft quantization function: $\widetilde{Q}_{\mathbf{c}}(\mathbf{x}) : \mathbb{R}^{d+m} \to \mathbb{R}^d$ and $\widetilde{Q}_{\mathbf{c}}(\mathbf{x})_i := \sum_{j=2}^m (c_j - c_{j-1})\sigma(P(x_i - \frac{c_j + c_{j-1}}{2})) + c_1$ where $\sigma$ denotes the sigmoid function and $P$ is a parameter controlling how closely $\widetilde{Q}_{\mathbf{c}}(\mathbf{x})$ approximates $Q_{\mathbf{c}}(\mathbf{x})$. Note that as $P \to \infty$, $\widetilde{Q}_{\mathbf{c}}(\mathbf{x}) \to Q_{\mathbf{c}}(\mathbf{x})$. This function can be seen as a simplification of the function that was used in [32].

**Assumption.** For all $j \in [m]$, $c_j$ is in a compact set. In other words, there exists a finite $c_{\max}$ such that $|c_j| \le c_{\max}$ for all $j \in [m]$.

In addition, we assume that the centers are sorted, i.e., $c_1 < \cdots < c_m$. Now, we state several useful facts.

**Fact 1.** $\widetilde{Q}_{\mathbf{c}}(\mathbf{x})$ *is continuously and infinitely differentiable everywhere.*

**Fact 2.** $\sigma(\mathbf{x})$ *is a Lipschitz continuous function.*

**Fact 3.** *Sum of Lipschitz continuous functions is also Lipschitz continuous.*

**Fact 4.** *Product of bounded and Lipschitz continuous functions is also Lipschitz continuous.*

**Fact 5.** *Let $g : \mathbb{R}^n \to \mathbb{R}^m$. Then, the coordinate-wise Lipschitz continuity implies overall Lipschitz continuity. In other words, let $g_i$ be the i'th output then if $g_i$ is Lipschitz continuous for all $i$, then $g$ is also Lipschitz continuous.*

In our convergence analysis, we require that $\widetilde{Q}_{\mathbf{c}}(\mathbf{x})$ is Lipschitz continuous as well as smooth with respect to both $\mathbf{x}$ and $\mathbf{c}$.

**Claim 1.** $\widetilde{Q}_{\mathbf{c}}(\mathbf{x})$ *is $l_{Q_1}$-Lipschitz continuous and $L_{Q_1}$-smooth with respect to $\mathbf{x}$.*

*Proof.* First we prove Lipschitz continuity. Note,

$$\frac{\partial \widetilde{Q}_{\mathbf{c}}(\mathbf{x})_i}{\partial x_j} = \begin{cases} 0, \text{ if } i \ne j \\ P \sum_{j=2}^m (c_j - c_{j-1})\sigma(P(x_i - \frac{c_j + c_{j-1}}{2}))(1 - \sigma(P(x_i - \frac{c_j + c_{j-1}}{2}))), \text{ if } i = j \end{cases} \tag{2}$$

As a result, $\|\frac{\partial \widetilde{Q}_{\mathbf{c}}(\mathbf{x})_i}{\partial x_j}\| \le \frac{P}{4}(c_m - c_1) \le \frac{P}{2}c_{max}$. The norm of the gradient of $\widetilde{Q}_{\mathbf{c}}(\mathbf{x})_i$ with respect to $x$ is bounded which implies there exists $l_{Q_1}^{(i)}$ such that $\|\widetilde{Q}_{\mathbf{c}}(\mathbf{x})_i - \widetilde{Q}_{\mathbf{c}}(\mathbf{x}')_i\| \le l_{Q_1}^{(i)} \|\mathbf{x} - \mathbf{x}'\|$; using Fact 5 and the fact that $i$ was arbitrary, there exists $l_{Q_1}$ such that $\|\widetilde{Q}_{\mathbf{c}}(\mathbf{x}) - \widetilde{Q}_{\mathbf{c}}(\mathbf{x}')\| \le l_{Q_1} \|\mathbf{x} - \mathbf{x}'\|$. In other words, $\widetilde{Q}_{\mathbf{c}}(\mathbf{x})$ is Lipschitz continuous.

For smoothness note that, $\nabla_{\mathbf{x}} \widetilde{Q}_{\mathbf{c}}(\mathbf{x}) = \nabla \widetilde{Q}_{\mathbf{c}}(\mathbf{x})_{1:d,:}$. Now we focus on an arbitrary term of $\nabla_{\mathbf{x}} \widetilde{Q}_{\mathbf{c}}(\mathbf{x})_{j,i}$. From (2) we know that this term is 0 if $i \ne j$, and a weighted sum of product of sigmoid functions if $i = j$. Then, using the Facts 1-4 the function $\nabla_{\mathbf{x}} \widetilde{Q}_{\mathbf{c}}(\mathbf{x})_{j,i}$ is Lipschitz continuous. Since $i, j$ were arbitrarily chosen, $\nabla_{\mathbf{x}} \widetilde{Q}_{\mathbf{c}}(\mathbf{x})_{j,i}$ is Lipschitz continuous for all $i, j$. Then,

by Fact 5, $\nabla_{\mathbf{x}}\widetilde{Q}_{\mathbf{c}}(\mathbf{x})$ is Lipschitz continuous, which implies that $\widetilde{Q}_{\mathbf{c}}(\mathbf{x})$ is $L_{Q_1}$-smooth for some coefficient $L_{Q_1} < \infty$. □

**Claim 2.** $\widetilde{Q}_{\mathbf{c}}(\mathbf{x})$ *is* $l_{Q_2}$*-Lipschitz continuous and* $L_{Q_2}$*-smooth with respect to* $\mathbf{c}$.

*Proof.* For Lipschitz continuity we have,

$$
\begin{aligned}
\frac{\partial \widetilde{Q}_{\mathbf{c}}(\mathbf{x})_i}{\partial c_j} = &\ \sigma(P(x_i - \frac{c_j + c_{j-1}}{2})) - \sigma(P(x_i - \frac{c_j + c_{j+1}}{2})) \\
&+ (c_j - c_{j+1})\frac{P}{2}\sigma(P(x_i - \frac{c_j + c_{j+1}}{2}))(1 - \sigma(P(x_i - \frac{c_j + c_{j+1}}{2}))) \\
&- (c_j - c_{j-1})\frac{P}{2}\sigma(P(x_i - \frac{c_j + c_{j-1}}{2}))(1 - \sigma(P(x_i - \frac{c_j + c_{j-1}}{2})))
\end{aligned}
$$

As a result, $\|\frac{\partial \widetilde{Q}_{\mathbf{c}}(\mathbf{x})_i}{\partial c_j}\| \le 2 + c_{max}\frac{P}{2}$. Similar to Claim 1 using the facts that $i$ is arbitrary and the Fact 5, we find there exists $l_{Q_2}$ such that $\|\widetilde{Q}_{\mathbf{c}}(\mathbf{x}) - \widetilde{Q}_{\mathbf{d}}(\mathbf{x})\| \le l_{Q_1}\|\mathbf{c} - \mathbf{d}\|$. In other words, $\widetilde{Q}_{\mathbf{c}}(\mathbf{x})$ is Lipschitz continuous. And for the smoothness, following the same idea from the proof of Claim 2 we find $\widetilde{Q}_{\mathbf{c}}(\mathbf{x})$ is $L_{Q_2}$-smooth with respect to $\mathbf{c}$. □

The example we gave in this section is simple yet provides technical necessities we require in the analysis. Other examples can also be used as long as they provide the smoothness properties that we utilize in the next sections.

## A.5 Lipschitz Relations

In this section we will use the assumptions **A.1-5** and show useful relations for partial gradients derived from the assumptions. We have the following gradient for the composite function:

$$
\nabla_{(\mathbf{x},\mathbf{c})}h(\mathbf{x}, \mathbf{c}) = \nabla_{(\mathbf{x},\mathbf{c})}f(\widetilde{Q}_{\mathbf{c}}(\mathbf{x})) = \nabla_{(\mathbf{x},\mathbf{c})}\widetilde{Q}_{\mathbf{c}}(\mathbf{x})\nabla_{\widetilde{Q}_{\mathbf{c}}(\mathbf{x})}f(\widetilde{Q}_{\mathbf{c}}(\mathbf{x})) \tag{3}
$$

where $dim(\nabla h(\mathbf{x}, \mathbf{c})) = (d + m) \times 1$, $dim(\nabla_{\widetilde{Q}_{\mathbf{c}}(\mathbf{x})}f(\widetilde{Q}_{\mathbf{c}}(\mathbf{x}))) = d \times 1$, $dim(\nabla\widetilde{Q}_{\mathbf{c}}(\mathbf{x})) = (d + m) \times d$. Note that the soft quantization functions of our interest are elementwise which implies $\frac{\partial \widetilde{Q}_{\mathbf{c}}(\mathbf{x})_i}{\partial x_j} = 0$ if $i \ne j$. In particular, for the gradient of the quantization function we have,

$$
\nabla_{(\mathbf{x},\mathbf{c})}\widetilde{Q}_{\mathbf{c}}(\mathbf{x}) =
\begin{bmatrix}
\frac{\partial \widetilde{Q}_{\mathbf{c}}(\mathbf{x})_1}{\partial x_1} & 0 & \cdots \\
0 & \frac{\partial \widetilde{Q}_{\mathbf{c}}(\mathbf{x})_2}{\partial x_2} & 0 \cdots \\
\vdots & \vdots & \vdots \\
\frac{\partial \widetilde{Q}_{\mathbf{c}}(\mathbf{x})_1}{\partial c_1} & \frac{\partial \widetilde{Q}_{\mathbf{c}}(\mathbf{x})_2}{\partial c_1} & \cdots \\
\vdots & \vdots & \vdots \\
\frac{\partial \widetilde{Q}_{\mathbf{c}}(\mathbf{x})_1}{\partial c_m} & \frac{\partial \widetilde{Q}_{\mathbf{c}}(\mathbf{x})_2}{\partial c_m} & \cdots
\end{bmatrix}
\tag{4}
$$

Moreover for the composite function we have,

$$\nabla h(\mathbf{x}, \mathbf{c}) = \begin{bmatrix} \frac{\partial f}{\partial \widetilde{Q}_{\mathbf{c}}(\mathbf{x})_1} \frac{\partial \widetilde{Q}_{\mathbf{c}}(\mathbf{x})_1}{\partial x_1} & +0 & +\dots \\ 0 & +\frac{\partial f}{\partial \widetilde{Q}_{\mathbf{c}}(\mathbf{x})_2} \frac{\partial \widetilde{Q}_{\mathbf{c}}(\mathbf{x})_2}{\partial x_2} & +\dots \\ \vdots & & \\ \frac{\partial f}{\partial \widetilde{Q}_{\mathbf{c}}(\mathbf{x})_1} \frac{\partial \widetilde{Q}_{\mathbf{c}}(\mathbf{x})_1}{\partial c_1} & +\frac{\partial f}{\partial \widetilde{Q}_{\mathbf{c}}(\mathbf{x})_2} \frac{\partial \widetilde{Q}_{\mathbf{c}}(\mathbf{x})_2}{\partial c_1} & +\dots \\ \vdots & & \\ \frac{\partial f}{\partial \widetilde{Q}_{\mathbf{c}}(\mathbf{x})_1} \frac{\partial \widetilde{Q}_{\mathbf{c}}(\mathbf{x})_1}{\partial c_m} & +\frac{\partial f}{\partial \widetilde{Q}_{\mathbf{c}}(\mathbf{x})_2} \frac{\partial \widetilde{Q}_{\mathbf{c}}(\mathbf{x})_2}{\partial c_m} & +\dots \end{bmatrix} \tag{5}$$

In (5) and (4) we use $x_i$, $c_j$ to denote $(\mathbf{x})_i$ and $(\mathbf{c})_j$ (i'th, j'th element respectively) for notational simplicity. Now, we prove two claims that will be useful in the main analysis.

**Claim 3.**

$$\|\nabla_{\mathbf{x}} f(\widetilde{Q}_{\mathbf{c}}(\mathbf{x})) - \nabla_{\mathbf{y}} f(\widetilde{Q}_{\mathbf{c}}(\mathbf{y}))\| = \|\nabla h(\mathbf{x}, \mathbf{c})_{1:d} - \nabla h(\mathbf{y}, \mathbf{c})_{1:d}\| \le (GL_{Q_1} + G_{Q_1} L l_{Q_1})\|\mathbf{x} - \mathbf{y}\|$$

*Proof.*

$$\begin{aligned}
\|\nabla h(\mathbf{x}, \mathbf{c})_{1:d} - \nabla h(\mathbf{y}, \mathbf{c})_{1:d}\| &= \|\nabla_{\mathbf{x}} f(\widetilde{Q}_{\mathbf{c}}(\mathbf{x})) - \nabla_{\mathbf{y}} f(\widetilde{Q}_{\mathbf{c}}(\mathbf{y}))\| \\
&= \|\nabla_{\widetilde{Q}_{\mathbf{c}}(\mathbf{x})} f(\widetilde{Q}_{\mathbf{c}}(\mathbf{x})) \nabla_{\mathbf{x}} \widetilde{Q}_{\mathbf{c}}(\mathbf{x}) - \nabla_{\widetilde{Q}_{\mathbf{c}}(\mathbf{y})} f(\widetilde{Q}_{\mathbf{c}}(\mathbf{y})) \nabla_{\mathbf{y}} \widetilde{Q}_{\mathbf{c}}(\mathbf{y})\| \\
&= \|\nabla_{\widetilde{Q}_{\mathbf{c}}(\mathbf{x})} f(\widetilde{Q}_{\mathbf{c}}(\mathbf{x})) \nabla_{\mathbf{x}} \widetilde{Q}_{\mathbf{c}}(\mathbf{x}) - \nabla_{\widetilde{Q}_{\mathbf{c}}(\mathbf{x})} f(\widetilde{Q}_{\mathbf{c}}(\mathbf{x})) \nabla_{\mathbf{y}} \widetilde{Q}_{\mathbf{c}}(\mathbf{y}) \\
&\quad + \nabla_{\widetilde{Q}_{\mathbf{c}}(\mathbf{x})} f(\widetilde{Q}_{\mathbf{c}}(\mathbf{x})) \nabla_{\mathbf{y}} \widetilde{Q}_{\mathbf{c}}(\mathbf{y}) - \nabla_{\widetilde{Q}_{\mathbf{c}}(\mathbf{y})} f(\widetilde{Q}_{\mathbf{c}}(\mathbf{y})) \nabla_{\mathbf{y}} \widetilde{Q}_{\mathbf{c}}(\mathbf{y})\| \\
&\overset{(a)}{\le} \|\nabla_{\widetilde{Q}_{\mathbf{c}}(\mathbf{x})} f(\widetilde{Q}_{\mathbf{c}}(\mathbf{x}))\| \|\nabla_{\mathbf{x}} \widetilde{Q}_{\mathbf{c}}(\mathbf{x}) - \nabla_{\mathbf{y}} \widetilde{Q}_{\mathbf{c}}(\mathbf{y})\|_F \\
&\quad + \|\nabla_{\mathbf{y}} \widetilde{Q}_{\mathbf{c}}(\mathbf{y})\|_F \|\nabla_{\widetilde{Q}_{\mathbf{c}}(\mathbf{x})} f(\widetilde{Q}_{\mathbf{c}}(\mathbf{x})) - \nabla_{\widetilde{Q}_{\mathbf{c}}(\mathbf{y})} f(\widetilde{Q}_{\mathbf{c}}(\mathbf{y}))\| \\
&\le GL_{Q_1}\|\mathbf{x} - \mathbf{y}\| + G_{Q_1} L \|\widetilde{Q}_{\mathbf{c}}(\mathbf{x}) - \widetilde{Q}_{\mathbf{c}}(\mathbf{y})\| \\
&\le GL_{Q_1}\|\mathbf{x} - \mathbf{y}\| + G_{Q_1} L l_{Q_1} \|\mathbf{x} - \mathbf{y}\| \\
&= (GL_{Q_1} + G_{Q_1} L l_{Q_1})\|\mathbf{x} - \mathbf{y}\|
\end{aligned}$$

To obtain (a) we have used the fact $\|\mathbf{A}\mathbf{x}\|_2 \le \|\mathbf{A}\|_F \|\mathbf{x}\|_2$. $\qquad\square$

**Claim 4.**

$$\begin{aligned}
\|\nabla_{\mathbf{c}} f(\widetilde{Q}_{\mathbf{c}}(\mathbf{x})) - \nabla_{\mathbf{d}} f(\widetilde{Q}_{\mathbf{d}}(\mathbf{x}))\| &= \|\nabla h(\mathbf{x}, \mathbf{c})_{d+1:m} - \nabla h(\mathbf{x}, \mathbf{d})_{d+1:m}\| \\
&\le (GL_{Q_2} + G_{Q_2} L l_{Q_2})\|\mathbf{c} - \mathbf{d}\|
\end{aligned}$$

*Proof.* We can follow similar steps,

$$\begin{aligned}
\|\nabla h(\mathbf{x}, \mathbf{c})_{d+1:m} - \nabla h(\mathbf{x}, \mathbf{d})_{d+1:m}\| &= \|\nabla_{\mathbf{c}} f(\widetilde{Q}_{\mathbf{c}}(\mathbf{x})) - \nabla_{\mathbf{d}} f(\widetilde{Q}_{\mathbf{d}}(\mathbf{x}))\| \\
&= \|\nabla_{\widetilde{Q}_{\mathbf{c}}(\mathbf{x})} f(\widetilde{Q}_{\mathbf{c}}(\mathbf{x})) \nabla_{\mathbf{c}} \widetilde{Q}_{\mathbf{c}}(\mathbf{x}) - \nabla_{\widetilde{Q}_{\mathbf{d}}(\mathbf{y})} f(\widetilde{Q}_{\mathbf{d}}(\mathbf{y})) \nabla_{\mathbf{d}} \widetilde{Q}_{\mathbf{d}}(\mathbf{y})\| \\
&= \|\nabla_{\widetilde{Q}_{\mathbf{c}}(\mathbf{x})} f(\widetilde{Q}_{\mathbf{c}}(\mathbf{x})) \nabla_{\mathbf{c}} \widetilde{Q}_{\mathbf{c}}(\mathbf{x}) - \nabla_{\widetilde{Q}_{\mathbf{c}}(\mathbf{x})} f(\widetilde{Q}_{\mathbf{c}}(\mathbf{x})) \nabla_{\mathbf{d}} \widetilde{Q}_{\mathbf{d}}(\mathbf{x}) \\
&\quad + \nabla_{\widetilde{Q}_{\mathbf{c}}(\mathbf{x})} f(\widetilde{Q}_{\mathbf{c}}(\mathbf{x})) \nabla_{\mathbf{d}} \widetilde{Q}_{\mathbf{d}}(\mathbf{y}) - \nabla_{\widetilde{Q}_{\mathbf{d}}(\mathbf{x})} f(\widetilde{Q}_{\mathbf{d}}(\mathbf{y})) \nabla_{\mathbf{d}} \widetilde{Q}_{\mathbf{d}}(\mathbf{y})\| \\
&\le \|\nabla_{\widetilde{Q}_{\mathbf{c}}(\mathbf{x})} f(\widetilde{Q}_{\mathbf{c}}(\mathbf{x}))\| \|\nabla_{\mathbf{c}} \widetilde{Q}_{\mathbf{c}}(\mathbf{x}) - \nabla_{\mathbf{d}} \widetilde{Q}_{\mathbf{d}}(\mathbf{y})\|_F \\
&\quad + \|\nabla_{\mathbf{d}} \widetilde{Q}_{\mathbf{d}}(\mathbf{y})\|_F \|\nabla_{\widetilde{Q}_{\mathbf{c}}(\mathbf{x})} f(\widetilde{Q}_{\mathbf{c}}(\mathbf{x})) - \nabla_{\widetilde{Q}_{\mathbf{d}}(\mathbf{y})} f(\widetilde{Q}_{\mathbf{d}}(\mathbf{y}))\| \\
&\le GL_{Q_2}\|\mathbf{c} - \mathbf{d}\| + G_{Q_2} L \|\widetilde{Q}_{\mathbf{c}}(\mathbf{x}) - \widetilde{Q}_{\mathbf{d}}(\mathbf{x})\| \\
&\le GL_{Q_2}\|\mathbf{c} - \mathbf{d}\| + G_{Q_2} L l_{Q_2} \|\mathbf{c} - \mathbf{d}\| \\
&= (GL_{Q_2} + G_{Q_2} L l_{Q_2})\|\mathbf{c} - \mathbf{d}\|
\end{aligned}$$

where $\nabla_{\mathbf{c}} \widetilde{Q}_{\mathbf{c}}(\mathbf{x}) = \nabla \widetilde{Q}_{\mathbf{c}}(\mathbf{x})_{(d+1:d+m,:)}$. $\qquad\square$

## A.6 Assumption A.7 is a Corollary of other Assumptions

In this section we discuss how Assumption **A.7** can be inferred from **A.1**-**A.5**. Here we drop client indices $i$ for notational simplicity. Let us define $f_w(\mathbf{w})$ as the neural network loss function with model $\mathbf{w}$. First we will argue that $f^{KD}(\mathbf{w}, \mathbf{x})$ is smooth, given that $f_w(\mathbf{w})$ and $f(\mathbf{x})$ are two smooth neural network loss functions with Cross Entropy as the loss function, and that $f_w(\mathbf{w})$ and $f(\mathbf{x})$ have bounded gradients. These two standard assumptions imply the smoothness of $f^{KD}(\mathbf{w}, \mathbf{x})$ individually with respect to both input parameters.

**Proposition 1.** $f^{KD}(\mathbf{w}, \mathbf{x})$ *is* $L_{D_1}$*-smooth with respect to* $\mathbf{x}$ *and* $L_{D_2}$*-smooth with respect to* $\mathbf{w}$ *for some positive constants* $L_{D_1}, L_{D_2}$.

*Proof.* Note that $f(\mathbf{x}) = \frac{1}{N}\sum_{i=1}^{N} \mathbf{y}_i^T \log(\frac{1}{s(\mathbf{x};\xi_i)})$ where $i$ denotes the index of data sample, $\mathbf{y}_i$ is the one hot encoding label vector, $N$ is the total number of data samples, $\log$ denotes elementwise logarithm and $\frac{1}{s(\mathbf{x};\xi_i)}$ denotes elementwise inverse; softmax function $s(\mathbf{x}) : \mathbb{R}^d \to \mathbb{R}^K$, where $K$ denotes the number of classes (similarly $s^w(\mathbf{w}) : \mathbb{R}^d \to \mathbb{R}^K$ is the function whose output is a vector of softmax probabilities and input is global model), is defined in Section 2, here we explicitly state that data samples $\xi_i$ is a parameterization of $s$. Assuming $f(\mathbf{x})$ is smooth for any possible pair of $(\mathbf{y}_i, \xi_i)$ implies $\log(\frac{1}{s(\mathbf{x})_j})$ is $C_x$-smooth for some constant $C_x$, here we used $s(\mathbf{x})_j$ to denote $j$'th output of $s(\mathbf{x})$ and we omitted $\xi_i$ since $\log(\frac{1}{s(\mathbf{x})_j})$ is smooth independent of $\xi_i$. Note that $s(\mathbf{x})_j : \mathbb{R}^d \to \mathbb{R}$. We have,

$$f^{KD}(\mathbf{w}, \mathbf{x}) = \frac{1}{N}\sum_{i=1}^{N}(s^w(\mathbf{w};\xi_i))^T \log(\frac{s^w(\mathbf{w};\xi_i)}{s(\mathbf{x};\xi_i)}) = \frac{1}{N}\sum_{i=1}^{N}(s^w(\mathbf{w};\xi_i))^T \log(s^w(\mathbf{w};\xi_i))$$

$$+ \frac{1}{N}\sum_{i=1}^{N}(s^w(\mathbf{w};\xi_i))^T \log(\frac{1}{s(\mathbf{x};\xi_i)})$$

where the operations are elementwise as before. In this expression only the last term depends on $\mathbf{x}$ and for each $i$, since $s^w(\mathbf{w};\xi_i)_j \leq 1$ with $\sum_j s^w(\mathbf{w};\xi_i)_j = 1$, the expression is a weighted average of smooth functions $\log(\frac{1}{s(\mathbf{x})_j})$; as a result, $f^{KD}(\mathbf{w}, \mathbf{x})$ is $L_{D_1}$-smooth with respect to $\mathbf{x}$ for some constant $L_{D_1}$.

Now we investigate smoothness with respect to $\mathbf{w}$. First, note that we can assume for all $j$ and $\mathbf{w}$, $s^w(\mathbf{w})_j$ is lower bounded by a positive constant $M > 0$ and upper bounded by a positive constant $P < 1$ since, by definition, output vector of the softmax function contains values between 0 and 1 (we ignore the limiting case when a logit is infinitely large, this is also implied by the smoothness assumptions on $f_w(\mathbf{w})$ and $f(\mathbf{x})$). Then, note that assuming a gradient bound on $f_w(\mathbf{w}) = \frac{1}{N}\sum_{i=1}^{N}\mathbf{y}_i^T \log(\frac{1}{s(\mathbf{w};\xi_i)})$ implies that for all $j$ $\|\nabla \log(\frac{1}{s^w(\mathbf{w})_j})\| = \|\frac{\nabla s^w(\mathbf{w})_j}{s^w(\mathbf{w})_j}\| \leq G_w$ for some constant $G_w$ (again the division operation is elementwise); since $0 < s^w(\mathbf{w})_j < 1$ we have $\|\nabla s^w(\mathbf{w})_j\| \leq G_w$. Moreover, similar to the first part, by assuming $f_w(\mathbf{w})$ is smooth we obtain that $\log(\frac{1}{s^w(\mathbf{w})_j})$ is smooth for all $j$. This implies having a bounded Hessian:

for some constant $C$ we have $C \geq \left\|\nabla^2 \log(\frac{1}{s^w(\mathbf{w})_j})\right\|_F = \left\|\frac{\nabla s^w(\mathbf{w})_j \nabla s^w(\mathbf{w})_j^T}{s^w(\mathbf{w})_j^2} - \frac{\nabla^2 s^w(\mathbf{w})_j}{s^w(\mathbf{w})_j}\right\|_F$

$$\overset{(a)}{\geq} \left|\left\|\frac{\nabla s^w(\mathbf{w})_j \nabla s^w(\mathbf{w})_j^T}{s^w(\mathbf{w})_j}\right\|_F - \left\|\nabla^2 s^w(\mathbf{w})_j\right\|_F\right|$$

$$= \left|\frac{G_w^2}{s^w(\mathbf{w})_j} - \left\|\nabla^2 s^w(\mathbf{w})_j\right\|_F\right|$$

$$\geq \left\|\nabla^2 s^w(\mathbf{w})_j\right\|_F - \frac{G_w^2}{s^w(\mathbf{w})_j}$$

$$\geq \left\|\nabla^2 s^w(\mathbf{w})_j\right\|_F - \frac{G_w^2}{M}$$

where (a) is due to reverse triangular inequality and $0 < s^w(\mathbf{w})_j < 1$. As a result we obtain $C + \frac{G_w^2}{M} \geq \left\| \nabla^2 s^w(\mathbf{w})_j \right\|_F$ for all $j$, i.e., $s^w(\mathbf{w})_j$ is $L_S$-smooth with some constant $L_S \leq C + \frac{G_w^2}{M}$. Thus, both $s^w(\mathbf{w})_j$ and $\log(s^w(\mathbf{w})_j)$ are smooth functions. Note both $s^w(\mathbf{w})_j$ and $\log(s^w(\mathbf{w})_j)$ are bounded functions. Consequently, the first summation term in the definition of $f^{KD}(\mathbf{w}, \mathbf{x})$ consists of the sum of product of bounded and smooth functions and the second term consists of sum of smooth functions multiplied with positive constants (as $\log(\frac{1}{s(\mathbf{x};\xi_i)})$ does not depend on $\mathbf{w}$). Using Fact 3 and Fact 4 we conclude there exists a constant $L_{D_2}$ such that $f^{KD}(\mathbf{w}, \mathbf{x})$ is $L_{D_2}$-smooth w.r.t $\mathbf{w}$. $\qquad\square$

**Proposition 2.** $f^{KD}(\mathbf{w}, \widetilde{Q}_\mathbf{c}(\mathbf{x}))$ *is* $L_{DQ_1}$*-smooth with respect to* $\mathbf{x}$, $L_{DQ_2}$*-smooth with respect to* $\mathbf{c}$, *and* $L_{DQ_3}$*-smooth with respect to* $\mathbf{w}$ *for some constants* $L_{DQ_1}, L_{DQ_2}, L_{DQ_3}$.

*Proof.* The proof is exactly the same as in Proposition 1, instead of using smoothness of $f(\mathbf{x})$, using smoothness of $f(\widetilde{Q}_\mathbf{c}(\mathbf{x}))$ with respect to $\mathbf{x}, \mathbf{c}$ from Claims 3 and 4 gives the result. $\qquad\square$

# B   Proof of Theorem 1

This proof consists of two parts. First we show the sufficient decrease property by sequentially using Lipschitz properties for each update step in Algorithm 1. For each variable $\mathbf{x}$ and $\mathbf{c}$ we find the decrease inequalities and then combine them to obtain an overall sufficient decrease. Then we bound the norm of the gradient using optimality conditions of the proximal updates in Algorithm 1. Using sufficient decrease and bound on the gradient we arrive at the result. We leave some of the derivations and proof of the claims to Appendix B.3.

**Alternating updates.** Remember that for the Algorithm 1 we have the following alternating updates:

$$\mathbf{x}^{t+1} = \text{prox}_{\eta_1 \lambda R_{\mathbf{c}^t}}(\mathbf{x}^t - \eta_1 \nabla f(\mathbf{x}^t) - \eta_1 \nabla_{\mathbf{x}^t} f(\widetilde{Q}_{\mathbf{c}^t}(\mathbf{x}^t)))$$

$$\mathbf{c}^{t+1} = \text{prox}_{\eta_2 \lambda R_{\mathbf{x}^{t+1}}}(\mathbf{c}^t - \eta_2 \nabla_{\mathbf{c}^t} f(\widetilde{Q}_{\mathbf{c}^t}(\mathbf{x}^{t+1})))$$

These translate to following optimization problems for $\mathbf{x}$ and $\mathbf{c}$ respectively (see end of the section for derivation):

$$\mathbf{x}^{t+1} = \underset{\mathbf{x} \in \mathbb{R}^d}{\arg\min} \left\{ \left\langle \mathbf{x} - \mathbf{x}^t, \nabla_{\mathbf{x}^t} f(\mathbf{x}^t) \right\rangle + \left\langle \mathbf{x} - \mathbf{x}^t, \nabla_{\mathbf{x}^t} f(\widetilde{Q}_{\mathbf{c}^t}(\mathbf{x}^t)) \right\rangle + \frac{1}{2\eta_1} \left\| \mathbf{x} - \mathbf{x}^t \right\|_2^2 + \lambda R(\mathbf{x}, \mathbf{c}^t) \right\}$$
(6)

$$\mathbf{c}^{t+1} = \underset{\mathbf{c} \in \mathbb{R}^m}{\arg\min} \left\{ \left\langle \mathbf{c} - \mathbf{c}^t, \nabla_{\mathbf{c}^t} f(\widetilde{Q}_{\mathbf{c}^t}(\mathbf{x}^{t+1})) \right\rangle + \frac{1}{2\eta_2} \left\| \mathbf{c} - \mathbf{c}^t \right\|_2^2 + \lambda R(\mathbf{x}^{t+1}, \mathbf{c}) \right\}$$
(7)

## B.1   Sufficient Decrease

This section is divided into two, first we will show sufficient decrease property with respect to $\mathbf{x}$, then we will show sufficient decrease property with respect to $\mathbf{c}$.

### B.1.1   Sufficient Decrease Due to x

**Claim 5.** $f(\mathbf{x}) + f(\widetilde{Q}_\mathbf{c}(\mathbf{x}))$ *is* $(L + GL_{Q_1} + G_{Q_1} LL_{Q_1})$*-smooth with respect to* $\mathbf{x}$.

Using Claim 5 we have,

$$F_\lambda(\mathbf{x}^{t+1}, \mathbf{c}^t) + (\frac{1}{2\eta_1} - \frac{L + GL_{Q_1} + G_{Q_1} LL_{Q_1}}{2}) \|\mathbf{x}^{t+1} - \mathbf{x}^t\|^2$$

$$= f(\mathbf{x}^{t+1}) + f(\widetilde{Q}_{\mathbf{c}^t}(\mathbf{x}^{t+1})) + \lambda R(\mathbf{x}^{t+1}, \mathbf{c}^t) \qquad (8)$$
$$+ (\frac{1}{2\eta_1} - \frac{L + GL_{Q_1} + G_{Q_1} LL_{Q_1}}{2}) \|\mathbf{x}^{t+1} - \mathbf{x}^t\|^2$$
$$\leq f(\mathbf{x}^t) + f(\widetilde{Q}_{\mathbf{c}^t}(\mathbf{x}^t)) + \lambda R(\mathbf{x}^{t+1}, \mathbf{c}^t) + \left\langle \nabla f(\mathbf{x}^t), \mathbf{x}^{t+1} - \mathbf{x}^t \right\rangle$$

$$+ \left\langle \nabla_{\mathbf{x}^t} f(\widetilde{Q}_{\mathbf{c}^t}(\mathbf{x}^t)), \mathbf{x}^{t+1} - \mathbf{x}^t \right\rangle + \frac{1}{2\eta_1} \|\mathbf{x}^{t+1} - \mathbf{x}^t\|^2 \qquad (9)$$

**Claim 6.** *Let*

$$A(\mathbf{x}^{t+1}) := \lambda R(\mathbf{x}^{t+1}, \mathbf{c}^t) + \left\langle \nabla f(\mathbf{x}^t), \mathbf{x}^{t+1} - \mathbf{x}^t \right\rangle + \left\langle \nabla_{\mathbf{x}^t} f(\widetilde{Q}_{\mathbf{c}^t}(\mathbf{x}^t)), \mathbf{x}^{t+1} - \mathbf{x}^t \right\rangle$$

$$+ \frac{1}{2\eta_1} \|\mathbf{x}^{t+1} - \mathbf{x}^t\|^2$$

$$A(\mathbf{x}^t) := \lambda R(\mathbf{x}^t, \mathbf{c}^t).$$

*Then $A(\mathbf{x}^{t+1}) \leq A(\mathbf{x}^t)$.*

Now we use Claim 6 and get,

$$f\left(\mathbf{x}^t\right) + \left\langle \mathbf{x}^{t+1} - \mathbf{x}^t, \nabla f\left(\mathbf{x}^t\right)\right\rangle + \frac{1}{2\eta_1} \left\|\mathbf{x}^{t+1} - \mathbf{x}^t\right\|_2^2 + \lambda R\left(\mathbf{x}^{t+1}, \mathbf{c}^t\right) + f(\widetilde{Q}_{\mathbf{c}^t}(\mathbf{x}^t))$$

$$+ \left\langle \nabla_{\mathbf{x}^t} f(\widetilde{Q}_{\mathbf{c}^t}(\mathbf{x}^t)), \mathbf{x}^{t+1} - \mathbf{x}^t \right\rangle \leq f\left(\mathbf{x}^t\right) + f(\widetilde{Q}_{\mathbf{c}^t}(\mathbf{x}^t)) + \lambda R\left(\mathbf{x}^t, \mathbf{c}^t\right)$$

$$= F_\lambda\left(\mathbf{x}^t, \mathbf{c}^t\right).$$

Using (9) we have,

$$F_\lambda(\mathbf{x}^{t+1}, \mathbf{c}^t) + \left(\frac{1}{2\eta_1} - \frac{L + GL_{Q_1} + G_{Q_1} LL_{Q_1}}{2}\right)\|\mathbf{x}^{t+1} - \mathbf{x}^t\|^2 \leq F_\lambda\left(\mathbf{x}^t, \mathbf{c}^t\right).$$

Now, we choose $\eta_1 = \frac{1}{2(L + GL_{Q_1} + G_{Q_1} Ll_{Q_1})}$ and obtain the decrease property for $\mathbf{x}$:

$$F_\lambda\left(\mathbf{x}^{t+1}, \mathbf{c}^t\right) + \frac{L + GL_{Q_1} + G_{Q_1} Ll_{Q_1}}{2}\|\mathbf{x}^{t+1} - \mathbf{x}^t\|^2 \leq F_\lambda\left(\mathbf{x}^t, \mathbf{c}^t\right). \qquad (10)$$

### B.1.2  Sufficient Decrease Due to c

From Claim 4 we have $f(\widetilde{Q}_{\mathbf{c}}(\mathbf{x}))$ is $(GL_{Q_2} + G_{Q_2} LL_{Q_2})$-smooth with respect to $\mathbf{c}$. Using Claim 4,

$$F_\lambda(\mathbf{x}^{t+1}, \mathbf{c}^{t+1}) + \left(\frac{1}{2\eta_2} - \frac{GL_{Q_2} + G_{Q_2} LL_{Q_2}}{2}\right)\|\mathbf{c}^{t+1} - \mathbf{c}^t\|^2$$

$$= f(\mathbf{x}^{t+1}) + f(\widetilde{Q}_{\mathbf{c}^{t+1}}(\mathbf{x}^{t+1})) + \lambda R(\mathbf{x}^{t+1}, \mathbf{c}^{t+1}) + \left(\frac{1}{2\eta_2} - \frac{GL_{Q_2} + G_{Q_2} LL_{Q_2}}{2}\right)\|\mathbf{c}^{t+1} - \mathbf{c}^t\|^2$$

$$\leq f(\mathbf{x}^{t+1}) + f(\widetilde{Q}_{\mathbf{c}^t}(\mathbf{x}^{t+1})) + \lambda R(\mathbf{x}^{t+1}, \mathbf{c}^{t+1}) + \left\langle \nabla_{\mathbf{c}^t} f(\widetilde{Q}_{\mathbf{c}^t}(\mathbf{x}^{t+1})), \mathbf{c}^{t+1} - \mathbf{c}^t \right\rangle$$

$$+ \frac{1}{2\eta_2}\|\mathbf{c}^{t+1} - \mathbf{c}^t\|^2 \qquad (11)$$

Now we state the counterpart of Claim 6 for $\mathbf{c}$.

**Claim 7.** *Let*

$$B(\mathbf{c}^{t+1}) := \lambda R(\mathbf{x}^{t+1}, \mathbf{c}^{t+1}) + \left\langle \nabla_{\mathbf{c}^t} f(\widetilde{Q}_{\mathbf{c}^t}(\mathbf{x}^{t+1})), \mathbf{c}^{t+1} - \mathbf{c}^t \right\rangle + \frac{1}{2\eta_1}\|\mathbf{c}^{t+1} - \mathbf{c}^t\|^2$$

$$B(\mathbf{c}^t) := \lambda R(\mathbf{x}^{t+1}, \mathbf{c}^t).$$

*Then $B(\mathbf{c}^{t+1}) \leq B(\mathbf{c}^t)$.*

Now using Claim 7,

$$f(\mathbf{x}^{t+1}) + \eta_2 \left\|\mathbf{c}^{t+1} - \mathbf{c}^t\right\|_2^2 + \lambda R(\mathbf{x}^{t+1}, \mathbf{c}^{t+1}) + f(\widetilde{Q}_{\mathbf{c}^t}(\mathbf{x}^{t+1})) + \left\langle \mathbf{c}^{t+1} - \mathbf{c}^t, \nabla_{\mathbf{c}^t} f(\widetilde{Q}_{\mathbf{c}^t}(\mathbf{x}^{t+1})) \right\rangle$$

$$\leq f\left(\mathbf{x}^{t+1}\right) + f(\widetilde{Q}_{\mathbf{c}^t}(\mathbf{x}^{t+1})) + \lambda R\left(\mathbf{x}^{t+1}, \mathbf{c}^t\right) = F_\lambda\left(\mathbf{x}^{t+1}, \mathbf{c}^t\right)$$

Setting $\eta_2 = \frac{1}{2(GL_{Q_2} + G_{Q_2} Ll_{Q_2})}$ and using the bound in (11), we obtain the sufficient decrease for $\mathbf{c}$:

$$F_\lambda\left(\mathbf{x}^{t+1}, \mathbf{c}^{t+1}\right) + \frac{GL_{Q_2} + G_{Q_2} Ll_{Q_2}}{2}\|\mathbf{c}^{t+1} - \mathbf{c}^t\|^2 \leq F_\lambda\left(\mathbf{x}^{t+1}, \mathbf{c}^t\right) \qquad (12)$$

### B.1.3 Overall Decrease

Summing the bounds in (10) and (12), we have the overall decrease property:

$$F_\lambda(\mathbf{x}^{t+1}, \mathbf{c}^{t+1}) + \frac{L + GL_{Q_1} + G_{Q_1}Ll_{Q_1}}{2}\|\mathbf{x}^{t+1} - \mathbf{x}^t\|^2 + \frac{GL_{Q_2} + G_{Q_2}Ll_{Q_2}}{2}\|\mathbf{c}^{t+1} - \mathbf{c}^t\|^2$$
$$\leq F_\lambda\left(\mathbf{x}^t, \mathbf{c}^t\right) \qquad (13)$$

Let us define $L_{\min} = \min\{L + GL_{Q_1} + G_{Q_1}Ll_{Q_1}, GL_{Q_2} + G_{Q_2}Ll_{Q_2}\}$, and $\mathbf{z}^t = (\mathbf{x}^t, \mathbf{c}^t)$. Then from (13):

$$F_\lambda\left(\mathbf{z}^{t+1}\right) + \frac{L_{\min}}{2}(\|\mathbf{z}^{t+1} - \mathbf{z}^t\|^2) = F_\lambda\left(\mathbf{x}^{t+1}, \mathbf{c}^{t+1}\right) + \frac{L_{\min}}{2}(\|\mathbf{x}^{t+1} - \mathbf{x}^t\|^2 + \|\mathbf{c}^{t+1} - \mathbf{c}^t\|^2)$$
$$\leq F_\lambda\left(\mathbf{x}^t, \mathbf{c}^t\right) = F_\lambda(\mathbf{z}^t)$$

Telescoping the above bound for $t = 0, \ldots, T-1$, and dividing by $T$:

$$\frac{1}{T}\sum_{t=0}^{T-1}(\|\mathbf{z}^{t+1} - \mathbf{z}^t\|_2^2) \leq \frac{2\left(F_\lambda\left(\mathbf{z}^0\right) - F_\lambda\left(\mathbf{z}^T\right)\right)}{L_{\min}T} \qquad (14)$$

### B.2 Bound on the Gradient

We now find the first order stationarity guarantee. Taking the derivative of (6) with respect to $\mathbf{x}$ at $\mathbf{x} = \mathbf{x}^{t+1}$ and setting it to 0 gives us the first order optimality condition:

$$\nabla f(\mathbf{x}^t) + \nabla_{\mathbf{x}^t} f(\widetilde{Q}_{\mathbf{c}^t}(\mathbf{x}^t)) + \frac{1}{\eta_1}\left(\mathbf{x}^{t+1} - \mathbf{x}^t\right) + \lambda\nabla_{\mathbf{x}^{t+1}} R\left(\mathbf{x}^{t+1}, \mathbf{c}^t\right) = 0 \qquad (15)$$

Combining the above equality and Claim 5:

$$\left\|\nabla_{\mathbf{x}^{t+1}} F_\lambda(\mathbf{x}^{t+1}, \mathbf{c}^t)\right\|_2 = \left\|\nabla f(\mathbf{x}^{t+1}) + \nabla_{\mathbf{x}^{t+1}} f(\widetilde{Q}_{\mathbf{c}^t}(\mathbf{x}^{t+1})) + \lambda\nabla_{\mathbf{x}^{t+1}} R(\mathbf{x}^{t+1}, \mathbf{c}^t)\right\|_2$$
$$\stackrel{(a)}{=} \left\|\frac{1}{\eta}\left(\mathbf{x}^t - \mathbf{x}^{t+1}\right) + \nabla f(\mathbf{x}^{t+1}) - \nabla f(\mathbf{x}^t) + \nabla_{\mathbf{x}^{t+1}} f(\widetilde{Q}_{\mathbf{c}^t}(\mathbf{x}^{t+1}))\right.$$
$$\left. - \nabla_{\mathbf{x}^t} f(\widetilde{Q}_{\mathbf{c}^t}(\mathbf{x}^t))\right\|_2$$
$$\leq (\frac{1}{\eta_1} + L + GL_{Q_1} + G_{Q_1}Ll_{Q_1})\left\|\mathbf{x}^{t+1} - \mathbf{x}^t\right\|_2$$
$$\stackrel{(b)}{=} 3(L + GL_{Q_1} + G_{Q_1}Ll_{Q_1})\left\|\mathbf{x}^{t+1} - \mathbf{x}^t\right\|_2$$
$$\leq 3(L + GL_{Q_1} + G_{Q_1}Ll_{Q_1})\left\|\mathbf{z}_{t+1} - \mathbf{z}_t\right\|_2$$

where (a) is from (15) and (b) is because we chose $\eta_1 = \frac{1}{2(L+GL_{Q_1}+G_{Q_1}Ll_{Q_1})}$. First order optimality condition in (7) for $\mathbf{c}^{t+1}$ gives:

$$\nabla_{\mathbf{c}^{t+1}} f(\widetilde{Q}_{\mathbf{c}^{t+1}}(\mathbf{x}^{t+1})) + \frac{1}{\eta_2}(\mathbf{c}^{t+1} - \mathbf{c}^t) + \lambda\nabla_{\mathbf{c}^{t+1}} R(\mathbf{x}^{t+1}, \mathbf{c}^{t+1}) = 0$$

Combining the above equality and Claim 4:

$$\left\|\nabla_{\mathbf{c}^{t+1}} F_\lambda(\mathbf{x}^{t+1}, \mathbf{c}^{t+1})\right\|_2 = \left\|\nabla_{\mathbf{c}^{t+1}} f(\widetilde{Q}_{\mathbf{c}^{t+1}}(\mathbf{x}^{t+1})) + \lambda\nabla_{\mathbf{c}^{t+1}} R(\mathbf{x}^{t+1}, \mathbf{c}^{t+1})\right\|_2$$
$$= \left\|\frac{1}{\eta_2}\left(\mathbf{c}^t - \mathbf{c}^{t+1}\right) + \nabla_{\mathbf{c}^{t+1}} f(\widetilde{Q}_{\mathbf{c}^{t+1}}(\mathbf{x}^{t+1})) - \nabla_{\mathbf{c}^t} f(\widetilde{Q}_{\mathbf{c}^t}(\mathbf{x}^{t+1}))\right\|_2$$
$$\leq (\frac{1}{\eta_2} + GL_{Q_2} + G_{Q_2}Ll_{Q_2})\left\|\mathbf{c}^{t+1} - \mathbf{c}^t\right\|_2$$
$$\stackrel{(a)}{=} 3(GL_{Q_2} + G_{Q_2}Ll_{Q_2})\left\|\mathbf{c}^{t+1} - \mathbf{c}^t\right\|_2$$

$$\leq 3(GL_{Q_2} + G_{Q_2}Ll_{Q_2}) \left\|\mathbf{z}^{t+1} - \mathbf{z}^t\right\|_2$$

where (a) is because we set $\eta_2 = \frac{1}{2(GL_{Q_2} + G_{Q_2}Ll_{Q_2})}$. Then:

$$\left\|[\nabla_{\mathbf{x}^{t+1}}F_\lambda(\mathbf{x}^{t+1}, \mathbf{c}^t)^T, \nabla_{\mathbf{c}^{t+1}}F_\lambda(\mathbf{x}^{t+1}, \mathbf{c}^{t+1})^T]^T\right\|_2^2 = \left\|\nabla_{\mathbf{x}}F_\lambda\left(\mathbf{x}^{t+1}, \mathbf{c}^t\right)\right\|_2^2$$
$$+ \left\|\nabla_{\mathbf{c}}F_\lambda\left(\mathbf{x}^{t+1}, \mathbf{c}^{t+1}\right)\right\|_2^2$$
$$\leq 3^2(G(L_{Q_1} + L_{Q_2}) + L(1 + G_{Q_1}l_{Q_1} + G_{Q_2}l_{Q_2}))^2\|\mathbf{z}^{t+1} - \mathbf{z}^t\|^2$$

Letting $L_{\max} = \max\{L + GL_{Q_1} + G_{Q_1}Ll_{Q_1}, GL_{Q_2} + G_{Q_2}Ll_{Q_2}\}$ we have:

$$\left\|[\nabla_{\mathbf{x}^{t+1}}F_\lambda(\mathbf{x}^{t+1}, \mathbf{c}^t)^T, \nabla_{\mathbf{c}^{t+1}}F_\lambda(\mathbf{x}^{t+1}, \mathbf{c}^{t+1})^T]^T\right\|_2^2 \leq 9L_{\max}^2\left\|\mathbf{z}^{t+1} - \mathbf{z}^t\right\|_2^2$$

Summing over all time points and dividing by $T$:

$$\frac{1}{T}\sum_{t=0}^{T-1}\left\|[\nabla_{\mathbf{x}^{t+1}}F_\lambda(\mathbf{x}^{t+1}, \mathbf{c}^t)^T, \nabla_{\mathbf{c}^{t+1}}F_\lambda(\mathbf{x}^{t+1}, \mathbf{c}^{t+1})^T]^T\right\|_2^2 \leq \frac{1}{T}\sum_{t=0}^{T-1}9L_{\max}^2(\left\|\mathbf{z}^{t+1} - \mathbf{z}^t\right\|_2)$$
$$\leq \frac{18L_{\max}^2\left(F_\lambda\left(\mathbf{z}^0\right) - F_\lambda(\mathbf{z}^T)\right)}{L_{\min}T},$$

where in the last inequality we use (14). This concludes the proof of Theorem 1.

### B.3 Omitted Details

First we derive the optimization problems that the alternating updates correspond to. Remember we had the following alternating updates:

$$\mathbf{x}^{t+1} = \text{prox}_{\eta_1\lambda R_{\mathbf{c}^t}}(\mathbf{x}^t - \eta_1\nabla f(\mathbf{x}^t) - \eta_1\nabla_{\mathbf{x}^t}f(\widetilde{Q}_{\mathbf{c}^t}(\mathbf{x}^t)))$$
$$\mathbf{c}^{t+1} = \text{prox}_{\eta_2\lambda R_{\mathbf{x}^{t+1}}}(\mathbf{c}^t - \eta_2\nabla_{\mathbf{c}^t}f(\widetilde{Q}_{\mathbf{c}^t}(\mathbf{x}^{t+1})))$$

For $\mathbf{x}^{t+1}$, from the definition of proximal mapping we have:

$$\mathbf{x}^{t+1} = \underset{\mathbf{x}\in\mathbb{R}^d}{\arg\min}\left\{\frac{1}{2\eta_1}\left\|\mathbf{x} - \mathbf{x}^t + \eta_1\nabla_{\mathbf{x}^t}f\left(\mathbf{x}^t\right) + \eta_1\nabla_{\mathbf{x}^t}f(\widetilde{Q}_{\mathbf{c}^t}(\mathbf{x}^t))\right\|_2^2 + \lambda R(\mathbf{x}, \mathbf{c}^t)\right\}$$
$$= \underset{\mathbf{x}\in\mathbb{R}^d}{\arg\min}\left\{\left\langle\mathbf{x} - \mathbf{x}^t, \nabla_{\mathbf{x}^t}f\left(\mathbf{x}^t\right)\right\rangle + \left\langle\mathbf{x} - \mathbf{x}^t, \nabla_{\mathbf{x}^t}f(\widetilde{Q}_{\mathbf{c}^t}(\mathbf{x}^t))\right\rangle + \frac{1}{2\eta_1}\left\|\mathbf{x} - \mathbf{x}^t\right\|_2^2\right.$$
$$\left. + \frac{\eta_1}{2}\|\nabla_{\mathbf{x}^t}f(\mathbf{x}^t) + \nabla_{\mathbf{x}^t}f(\widetilde{Q}_{\mathbf{c}^t}(\mathbf{x}^t))\|^2 + \lambda R(\mathbf{x}, \mathbf{c}^t)\right\}$$
$$= \underset{\mathbf{x}\in\mathbb{R}^d}{\arg\min}\left\{\left\langle\mathbf{x} - \mathbf{x}^t, \nabla_{\mathbf{x}^t}f\left(\mathbf{x}^t\right)\right\rangle + \left\langle\mathbf{x} - \mathbf{x}^t, \nabla_{\mathbf{x}^t}f(\widetilde{Q}_{\mathbf{c}^t}(\mathbf{x}^t))\right\rangle + \frac{1}{2\eta_1}\left\|\mathbf{x} - \mathbf{x}^t\right\|_2^2\right.$$
$$\left. + \lambda R(\mathbf{x}, \mathbf{c}^t)\right\} \tag{16}$$

Note, in the third equality we remove the terms that do not depend on $\mathbf{x}$. Similarly, for $\mathbf{c}^{t+1}$ we have:

$$\mathbf{c}^{t+1} = \underset{\mathbf{c}\in\mathbb{R}^m}{\arg\min}\left\{\frac{1}{2\eta_2}\left\|\mathbf{c} - \mathbf{c}^t + \eta_2\nabla_{\mathbf{c}^t}f(\widetilde{Q}_{\mathbf{c}^t}(\mathbf{x}^{t+1}))\right\|_2^2 + \lambda R(\mathbf{x}^{t+1}, \mathbf{c})\right\}$$
$$= \underset{\mathbf{c}\in\mathbb{R}^m}{\arg\min}\left\{\left\langle\mathbf{c} - \mathbf{c}^t, \nabla_{\mathbf{c}^t}f(\widetilde{Q}_{\mathbf{c}^t}(\mathbf{x}^{t+1}))\right\rangle + \frac{1}{2\eta_2}\left\|\mathbf{c} - \mathbf{c}^t\right\|_2^2 + \frac{\eta_2}{2}\|\nabla_{\mathbf{c}^t}f(\widetilde{Q}_{\mathbf{c}^t}(\mathbf{x}^{t+1}))\|^2\right.$$
$$\left. + \lambda R(\mathbf{x}^{t+1}, \mathbf{c})\right\}$$
$$= \underset{\mathbf{c}\in\mathbb{R}^m}{\arg\min}\left\{\left\langle\mathbf{c} - \mathbf{c}^t, \nabla_{\mathbf{c}^t}f(\widetilde{Q}_{\mathbf{c}^t}(\mathbf{x}^{t+1}))\right\rangle + \frac{1}{2\eta_2}\left\|\mathbf{c} - \mathbf{c}^t\right\|_2^2 + \lambda R(\mathbf{x}^{t+1}, \mathbf{c})\right\} \tag{17}$$

Minimization problems in (16) and (17) are the main problems to characterize the update rules and we use them in multiple places throughout the section.

### B.3.1 Proof of the Claims

**Claim** (Restating Claim 5). $f(\mathbf{x}) + f(\widetilde{Q}_{\mathbf{c}}(\mathbf{x}))$ *is* $(L + GL_{Q_1} + G_{Q_1}LL_{Q_1})$*-smooth with respect to* $\mathbf{x}$.

*Proof.* From our assumptions, we have $f$ is $L$-smooth. And from Claim 3 we have $f(\widetilde{Q}_{\mathbf{c}}(\mathbf{x}))$ is $(GL_{Q_1} + G_{Q_1}LL_{Q_1})$-smooth. Using the fact that if two functions $g_1$ and $g_2$ are $L_1$ and $L_2$ smooth respectively, then $g_1 + g_2$ is $(L_1 + L_2)$-smooth concludes the proof. $\qquad\square$

**Claim** (Restating Claim 6). *Let*

$$A(\mathbf{x}^{t+1}) := \lambda R(\mathbf{x}^{t+1}, \mathbf{c}^t) + \left\langle \nabla f(\mathbf{x}^t), \mathbf{x}^{t+1} - \mathbf{x}^t \right\rangle + \left\langle \nabla_{\mathbf{x}^t} f(\widetilde{Q}_{\mathbf{c}^t}(\mathbf{x}^t)), \mathbf{x}^{t+1} - \mathbf{x}^t \right\rangle$$

$$+ \frac{1}{2\eta_1} \|\mathbf{x}^{t+1} - \mathbf{x}^t\|^2$$

$$A(\mathbf{x}^t) := \lambda R(\mathbf{x}^t, \mathbf{c}^t).$$

*Then* $A(\mathbf{x}^{t+1}) \leq A(\mathbf{x}^t)$.

*Proof.* Let $A(\mathbf{x})$ denote the expression inside the $\arg\min$ in (16) and we know that (16) is minimized when $\mathbf{x} = \mathbf{x}^{t+1}$. So we have $A(\mathbf{x}^{t+1}) \leq A(\mathbf{x}^t)$. This proves the claim. $\qquad\square$

**Claim** (Restating Claim 7). *Let*

$$B(\mathbf{c}^{t+1}) := \lambda R(\mathbf{x}^{t+1}, \mathbf{c}^{t+1}) + \left\langle \nabla_{\mathbf{c}^t} f(\widetilde{Q}_{\mathbf{c}^t}(\mathbf{x}^{t+1})), \mathbf{c}^{t+1} - \mathbf{c}^t \right\rangle + \frac{1}{2\eta_1} \|\mathbf{c}^{t+1} - \mathbf{c}^t\|^2$$

$$B(\mathbf{c}^t) := \lambda R(\mathbf{x}^{t+1}, \mathbf{c}^t).$$

*Then* $B(\mathbf{c}^{t+1}) \leq B(\mathbf{c}^t)$.

*Proof.* Let $B(\mathbf{c})$ denote the expression inside the $\arg\min$ in (17) and we know that (17) is minimized when $\mathbf{c} = \mathbf{c}^{t+1}$. So we have $B(\mathbf{c}^{t+1}) \leq B(\mathbf{c}^t)$. This proves the claim. $\qquad\square$

## C  Proof of Theorem 2

In this part, different than Section 3, we have an additional update due to local iterations. The key is to integrate the local iterations into our alternating update scheme. To do this, we utilize Assumptions **A.6** and **A.7**. This proof consists of two parts. First, we show the sufficient decrease property by sequentially using and combining Lipschitz properties for each update step in Algorithm 2. Then, we bound the norm of the gradient using optimality conditions of the proximal updates in Algorithm 2. Then, by combining the sufficient decrease results and bounds on partial gradients we will derive our result. We defer proofs of the claims and some derivation details to the end of this section. In this analysis we take $\mathbf{w}^t = \frac{1}{n} \sum_{i=1}^n \mathbf{w}_i^t$, so that $\mathbf{w}^t$ is defined for every time point.

**Alternating updates.** Let us first restate the alternating updates for $\mathbf{x}_i$ and $\mathbf{c}_i$:

$$\mathbf{x}_i^{t+1} = \text{prox}_{\eta_1 \lambda R_{\mathbf{c}_i^t}} \Big( \mathbf{x}_i^t - (1 - \lambda_p)\eta_1 \nabla f_i(\mathbf{x}_i^t) - (1 - \lambda_p)\eta_1 \nabla_{\mathbf{x}_i^t} f_i(\widetilde{Q}_{\mathbf{c}_i^t}(\mathbf{x}_i^t))$$

$$- \eta_1 \lambda_p \nabla_{\mathbf{x}_i^t} f_i^{KD}(\mathbf{x}_i^t, \mathbf{w}_i^t) - \eta_1 \lambda_p \nabla_{\mathbf{x}_i^t} f_i^{KD}(\widetilde{Q}_{\mathbf{c}_i^t}(\mathbf{x}_i^t), \mathbf{w}_i^t) \Big)$$

$$\mathbf{c}_i^{t+1} = \text{prox}_{\eta_2 \lambda R_{\mathbf{x}_i^{t+1}}} \Big( \mathbf{c}_i^t - (1 - \lambda_p)\eta_2 \nabla_{\mathbf{c}_i^t} f_i(\widetilde{Q}_{\mathbf{c}_i^t}(\mathbf{x}_i^{t+1})) - \eta_2 \lambda_p \nabla_{\mathbf{c}_i^t} f_i^{KD}(\widetilde{Q}_{\mathbf{c}_i^t}(\mathbf{x}_i^{t+1}), \mathbf{w}_i^t) \Big)$$

The alternating updates are equivalent to solving the following two optimization problems.

$$\mathbf{x}_i^{t+1} = \arg\min_{\mathbf{x} \in \mathbb{R}^d} \Big\{ (1 - \lambda_p) \left\langle \mathbf{x} - \mathbf{x}_i^t, \nabla f_i(\mathbf{x}_i^t) \right\rangle + (1 - \lambda_p) \left\langle \mathbf{x} - \mathbf{x}_i^t, \nabla_{\mathbf{x}_i^t} f_i(\widetilde{Q}_{\mathbf{c}_i^t}(\mathbf{x}_i^t)) \right\rangle$$

$$+ \left\langle \mathbf{x} - \mathbf{x}_i^t, \lambda_p \nabla_{\mathbf{x}_i^t} f_i^{KD}(\mathbf{x}_i^t, \mathbf{w}_i^t) \right\rangle + \left\langle \mathbf{x} - \mathbf{x}_i^t, \lambda_p \nabla_{\mathbf{x}_i^t} f_i^{KD}(\widetilde{Q}_{\mathbf{c}_i^t}(\mathbf{x}_i^t), \mathbf{w}_i^t) \right\rangle$$

$$+ \frac{1}{2\eta_1} \|\mathbf{x} - \mathbf{x}_i^t\|_2^2 + \lambda R(\mathbf{x}, \mathbf{c}_i^t) \Big\} \tag{18}$$

$$\mathbf{c}_i^{t+1} = \arg\min_{\mathbf{c}\in\mathbb{R}^m}\Big\{ \Big\langle \mathbf{c} - \mathbf{c}_i^t, (1-\lambda_p)\nabla_{\mathbf{c}_i^t}f_i(\widetilde{Q}_{\mathbf{c}_i^t}(\mathbf{x}_i^{t+1}))\Big\rangle + \Big\langle \mathbf{c} - \mathbf{c}_i^t, \lambda_p\nabla_{\mathbf{c}_i^t}f_i^{KD}(\widetilde{Q}_{\mathbf{c}_i^t}(\mathbf{x}_i^{t+1}), \mathbf{w}_i^t)\Big\rangle$$
$$+ \frac{1}{2\eta_2}\left\|\mathbf{c} - \mathbf{c}_i^t\right\|_2^2 + \lambda R(\mathbf{x}_i^{t+1}, \mathbf{c})\Big\} \tag{19}$$

Note that the update on $\mathbf{w}^t$ from Algorithm 2 can be written as:

$$\mathbf{w}^{t+1} = \mathbf{w}^t - \eta_3\mathbf{g}^t, \quad \text{where} \quad \mathbf{g}^t = \frac{1}{n}\sum_{i=1}^n \nabla_{\mathbf{w}_i^t}F_i(\mathbf{x}_i^{t+1}, \mathbf{c}_i^{t+1}, \mathbf{w}_i^t).$$

In the convergence analysis we require smoothness of the local functions $F_i$ w.r.t. the global parameter $\mathbf{w}$. Recall the definition of $F_i(\mathbf{x}_i, \mathbf{c}_i, \mathbf{w}) = (1-\lambda_p)\Big(f_i(\mathbf{x}_i) + f_i(\widetilde{Q}_{\mathbf{c}_i}(\mathbf{x}_i))\Big) + \lambda R(\mathbf{x}_i, \mathbf{c}_i) + \lambda_p\Big(f_i^{KD}(\mathbf{x}_i, \mathbf{w}) + f_i^{KD}(\widetilde{Q}_{\mathbf{c}_i}(\mathbf{x}_i), \mathbf{w})\Big)$ from (2). It follows that from Assumption **A.7** that $F_i$ is $(\lambda_p(L_{D_2} + L_{DQ_3}))$-smooth with respect to $\mathbf{w}$: Now let us move on with the proof.

## C.1 Sufficient Decrease

We will divide this part into three and obtain sufficient decrease properties for each variable: $\mathbf{x}_i, \mathbf{c}_i, \mathbf{w}$.

### C.1.1 Sufficient Decrease Due to $\mathbf{x}_i$

We begin with a useful claim.

**Claim 8.** $(1-\lambda_p)(f_i(\mathbf{x}) + f_i(\widetilde{Q}_{\mathbf{c}}(\mathbf{x}))) + \lambda_p(f_i^{KD}(\mathbf{x}, \mathbf{w}) + f_i^{KD}(\widetilde{Q}_{\mathbf{c}}(\mathbf{x}), \mathbf{w}))$ *is* $(\lambda_p(L_{D_1} + L_{DQ_1}) + (1-\lambda_p)(L + G^{(i)}L_{Q_1} + G_{Q_1}^{(i)}Ll_{Q_1}))$*-smooth with respect to* $\mathbf{x}$.

*Proof.* From our assumptions, we have $f_i$ is $L$-smooth, $f_i^{KD}(\mathbf{x}, \mathbf{w})$ is $L_{D_1}$-smooth and $f_i^{KD}(\widetilde{Q}_{\mathbf{c}}(\mathbf{x}), \mathbf{w})$ is $L_{DQ_1}$-smooth with respect to $\mathbf{x}$. And applying the Claim 3 to each client separately gives that $f_i(\widetilde{Q}_{\mathbf{c}}(\mathbf{x}))$ is $(G^{(i)}L_{Q_1} + G_{Q_1}^{(i)}Ll_{Q_1})$-smooth. Using the fact that if two functions $g_1$ and $g_2$ (defined over the same space) are $L_1$ and $L_2$-smooth respectively, then $g_1 + g_2$ is $(L_1 + L_2)$-smooth, and the fact that $\alpha g_1$ is $\alpha L_1$-smooth for a given constant $\alpha$ concludes the proof. $\qquad\square$

From Claim 8 we have:

$$F_i(\mathbf{x}_i^{t+1}, \mathbf{c}_i^t, \mathbf{w}^t) + \Big(\frac{1}{2\eta_1} - \frac{\lambda_p(L_{D_1} + L_{DQ_1}) + (1-\lambda_p)(L + G^{(i)}L_{Q_1} + G_{Q_1}^{(i)}Ll_{Q_1})}{2}\Big)\|\mathbf{x}_i^{t+1} - \mathbf{x}_i^t\|^2$$

$$= (1-\lambda_p)\Big(f_i(\mathbf{x}_i^{t+1}) + f_i(\widetilde{Q}_{\mathbf{c}_i^t}(\mathbf{x}_i^{t+1}))\Big) + \lambda_p\Big(f_i^{KD}(\mathbf{x}_i^{t+1}, \mathbf{w}^t) + f_i^{KD}(\widetilde{Q}_{\mathbf{c}_i^t}(\mathbf{x}_i^{t+1}), \mathbf{w}^t)\Big)$$

$$+ \Big(\frac{1}{2\eta_1} - \frac{\lambda_p(L_{D_1} + L_{DQ_1}) + (1-\lambda_p)(L + G^{(i)}L_{Q_1} + G_{Q_1}^{(i)}Ll_{Q_1})}{2}\Big)\|\mathbf{x}_i^{t+1} - \mathbf{x}_i^t\|^2$$

$$+ \lambda R(\mathbf{x}_i^{t+1}, \mathbf{c}_i^t)$$

$$\leq (1-\lambda_p)\Big(f_i(\mathbf{x}_i^t) + f_i(\widetilde{Q}_{\mathbf{c}_i^t}(\mathbf{x}_i^t))\Big) + \lambda_p\Big(f_i^{KD}(\mathbf{x}_i^t, \mathbf{w}^t) + f_i^{KD}(\widetilde{Q}_{\mathbf{c}_i^t}(\mathbf{x}_i^t), \mathbf{w}^t)\Big)$$

$$+ (1-\lambda_p)\Big\langle\nabla f_i(\mathbf{x}_i^t), \mathbf{x}_i^{t+1} - \mathbf{x}_i^t\Big\rangle + (1-\lambda_p)\Big\langle\nabla_{\mathbf{x}_i^t}f_i(\widetilde{Q}_{\mathbf{c}_i^t}(\mathbf{x}_i^t)), \mathbf{x}_i^{t+1} - \mathbf{x}_i^t\Big\rangle$$

$$+ \lambda_p\Big\langle\nabla_{\mathbf{x}_i^t}f_i^{KD}(\mathbf{x}_i^t, \mathbf{w}^t), \mathbf{x}_i^{t+1} - \mathbf{x}_i^t\Big\rangle + \lambda_p\Big\langle\nabla_{\mathbf{x}_i^t}f_i^{KD}(\widetilde{Q}_{\mathbf{c}_i^t}(\mathbf{x}_i^t), \mathbf{w}^t), \mathbf{x}_i^{t+1} - \mathbf{x}_i^t\Big\rangle$$

$$+ \lambda R(\mathbf{x}_i^{t+1}, \mathbf{c}_i^t) + \frac{1}{2\eta_1}\|\mathbf{x}_i^{t+1} - \mathbf{x}_i^t\|^2$$

$$= (1-\lambda_p)\Big(f_i(\mathbf{x}_i^t) + f_i(\widetilde{Q}_{\mathbf{c}_i^t}(\mathbf{x}_i^t))\Big) + \lambda_p\Big(f_i^{KD}(\mathbf{x}_i^t, \mathbf{w}^t) + f_i^{KD}(\widetilde{Q}_{\mathbf{c}_i^t}(\mathbf{x}_i^t), \mathbf{w}^t)\Big)$$

$$+ (1-\lambda_p)\Big\langle\nabla f_i(\mathbf{x}_i^t), \mathbf{x}_i^{t+1} - \mathbf{x}_i^t\Big\rangle + (1-\lambda_p)\Big\langle\nabla_{\mathbf{x}_i^t}f_i(\widetilde{Q}_{\mathbf{c}_i^t}(\mathbf{x}_i^t)), \mathbf{x}_i^{t+1} - \mathbf{x}_i^t\Big\rangle$$

$$+ \lambda_p \left\langle \nabla_{\mathbf{x}_i^t} f_i^{KD}(\mathbf{x}_i^t, \mathbf{w}^t) - \nabla_{\mathbf{x}_i^t} f_i^{KD}(\mathbf{x}_i^t, \mathbf{w}_i^t), \mathbf{x}_i^{t+1} - \mathbf{x}_i^t \right\rangle$$

$$+ \lambda_p \left\langle \nabla_{\mathbf{x}_i^t} f_i^{KD}(\widetilde{Q}_{\mathbf{c}_i^t}(\mathbf{x}_i^t), \mathbf{w}^t) - \nabla_{\mathbf{x}_i^t} f_i^{KD}(\widetilde{Q}_{\mathbf{c}_i^t}(\mathbf{x}_i^t), \mathbf{w}_i^t), \mathbf{x}_i^{t+1} - \mathbf{x}_i^t \right\rangle$$

$$+ \lambda_p \left\langle \nabla_{\mathbf{x}_i^t} f_i^{KD}(\mathbf{x}_i^t, \mathbf{w}_i^t), \mathbf{x}_i^{t+1} - \mathbf{x}_i^t \right\rangle + \lambda_p \left\langle \nabla_{\mathbf{x}_i^t} f_i^{KD}((\widetilde{Q}_{\mathbf{c}_i^t}(\mathbf{x}_i^t), \mathbf{w}_i^t), \mathbf{x}_i^{t+1} - \mathbf{x}_i^t \right\rangle$$

$$+ \frac{1}{2\eta_1} \|\mathbf{x}_i^{t+1} - \mathbf{x}_i^t\|^2 + \lambda R(\mathbf{x}_i^{t+1}, \mathbf{c}_i^t)$$

$$\leq (1 - \lambda_p) \left( f_i(\mathbf{x}_i^t) + f_i(\widetilde{Q}_{\mathbf{c}_i^t}(\mathbf{x}_i^t)) \right) + \lambda_p \left( f_i^{KD}(\mathbf{x}_i^t, \mathbf{w}^t) + f_i^{KD}(\widetilde{Q}_{\mathbf{c}_i^t}(\mathbf{x}_i^t), \mathbf{w}^t) \right)$$

$$+ (1 - \lambda_p) \left\langle \nabla f_i(\mathbf{x}_i^t), \mathbf{x}_i^{t+1} - \mathbf{x}_i^t \right\rangle + (1 - \lambda_p) \left\langle \nabla_{\mathbf{x}_i^t} f_i(\widetilde{Q}_{\mathbf{c}_i^t}(\mathbf{x}_i^t)), \mathbf{x}_i^{t+1} - \mathbf{x}_i^t \right\rangle$$

$$+ \lambda_p \left\langle \nabla_{\mathbf{x}_i^t} f_i^{KD}(\mathbf{x}_i^t, \mathbf{w}_i^t), \mathbf{x}_i^{t+1} - \mathbf{x}_i^t \right\rangle + \lambda_p \left\langle \nabla_{\mathbf{x}_i^t} f_i^{KD}((\widetilde{Q}_{\mathbf{c}_i^t}(\mathbf{x}_i^t), \mathbf{w}_i^t), \mathbf{x}_i^{t+1} - \mathbf{x}_i^t \right\rangle$$

$$+ \frac{\lambda_p}{2} \|\nabla_{\mathbf{x}_i^t} f_i^{KD}(\widetilde{Q}_{\mathbf{c}_i^t}(\mathbf{x}_i^t), \mathbf{w}^t) - \nabla_{\mathbf{x}_i^t} f_i^{KD}(\widetilde{Q}_{\mathbf{c}_i^t}(\mathbf{x}_i^t), \mathbf{w}_i^t)\|^2 + \lambda_p \|\mathbf{x}_i^{t+1} - \mathbf{x}_i^t\|^2$$

$$+ \frac{\lambda_p}{2} \|\nabla_{\mathbf{x}_i^t} f_i^{KD}(\mathbf{x}_i^t, \mathbf{w}^t) - \nabla_{\mathbf{x}_i^t} f_i^{KD}(\mathbf{x}_i^t, \mathbf{w}_i^t)\|^2 + \frac{1}{2\eta_1} \|\mathbf{x}_i^{t+1} - \mathbf{x}_i^t\|^2 + \lambda R(\mathbf{x}_i^{t+1}, \mathbf{c}_i^t).$$

(20)

In the last inequality we used:

$$\left\langle \lambda_p(\nabla_{\mathbf{x}_i^t} f_i^{KD}(\mathbf{x}_i^t, \mathbf{w}^t) - \nabla_{\mathbf{x}_i^t} f_i^{KD}(\mathbf{x}_i^t, \mathbf{w}_i^t)), \mathbf{x}_i^{t+1} - \mathbf{x}_i^t \right\rangle$$

$$= \left\langle \sqrt{\lambda_p}(\nabla_{\mathbf{x}_i^t} f_i^{KD}(\mathbf{x}_i^t, \mathbf{w}^t) - \nabla_{\mathbf{x}_i^t} f_i^{KD}(\mathbf{x}_i^t, \mathbf{w}_i^t)), \sqrt{\lambda_p}(\mathbf{x}_i^{t+1} - \mathbf{x}_i^t) \right\rangle$$

$$\leq \frac{\lambda_p}{2} \|\nabla_{\mathbf{x}_i^t} f_i^{KD}(\mathbf{x}_i^t, \mathbf{w}^t) - \nabla_{\mathbf{x}_i^t} f_i^{KD}(\mathbf{x}_i^t, \mathbf{w}_i^t)\|^2 + \frac{\lambda_p}{2} \|\mathbf{x}_i^{t+1} - \mathbf{x}_i^t\|^2$$

and similarly,

$$\left\langle \lambda_p(\nabla_{\mathbf{x}_i^t} f_i^{KD}(\widetilde{Q}_{\mathbf{c}_i^t}(\mathbf{x}_i^t), \mathbf{w}^t) - \nabla_{\mathbf{x}_i^t} f_i^{KD}(\widetilde{Q}_{\mathbf{c}_i^t}(\mathbf{x}_i^t), \mathbf{w}_i^t)), \mathbf{x}_i^{t+1} - \mathbf{x}_i^t \right\rangle$$

$$\leq \frac{\lambda_p}{2} \|\nabla_{\mathbf{x}_i^t} f_i^{KD}(\widetilde{Q}_{\mathbf{c}_i^t}(\mathbf{x}_i^t), \mathbf{w}^t) - \nabla_{\mathbf{x}_i^t} f_i^{KD}(\widetilde{Q}_{\mathbf{c}_i^t}(\mathbf{x}_i^t), \mathbf{w}_i^t)\|^2$$

$$+ \frac{\lambda_p}{2} \|\mathbf{x}_i^{t+1} - \mathbf{x}_i^t\|^2$$

**Claim 9.** *Let*

$$A(\mathbf{x}_i^{t+1}) := (1 - \lambda_p) \left\langle \nabla f_i(\mathbf{x}_i^t), \mathbf{x}_i^{t+1} - \mathbf{x}_i^t \right\rangle + (1 - \lambda_p) \left\langle \nabla_{\mathbf{x}_i^t} f_i(\widetilde{Q}_{\mathbf{c}_i^t}(\mathbf{x}_i^t)), \mathbf{x}_i^{t+1} - \mathbf{x}_i^t \right\rangle$$

$$+ \left\langle \lambda_p(\nabla_{\mathbf{x}_i^t} f_i^{KD}(\mathbf{x}_i^t, \mathbf{w}_i^t)), \mathbf{x}_i^{t+1} - \mathbf{x}_i^t \right\rangle + \left\langle \lambda_p(\nabla_{\mathbf{x}_i^t} f_i^{KD}(\widetilde{Q}_{\mathbf{c}_i^t}(\mathbf{x}_i^t), \mathbf{w}_i^t)), \mathbf{x}_i^{t+1} - \mathbf{x}_i^t \right\rangle$$

$$+ \lambda R(\mathbf{x}_i^{t+1}, \mathbf{c}_i^t) + \frac{1}{2\eta_1} \|\mathbf{x}_i^{t+1} - \mathbf{x}_i^t\|^2$$

$$A(\mathbf{x}_i^t) := \lambda R(\mathbf{x}_i^t, \mathbf{c}_i^t).$$

*Then* $A(\mathbf{x}_i^{t+1}) \leq A(\mathbf{x}_i^t)$.

*Proof.* Let $A(\mathbf{x})$ denote the expression inside the $\arg\min$ in (18) and we know that (18) is minimized when $\mathbf{x} = \mathbf{x}_i^{t+1}$. So we have $A(\mathbf{x}_i^{t+1}) \leq A(\mathbf{x}_i^t)$. This proves the claim. $\square$

Using the inequality from Claim 9 in (20) gives

$$F_i(\mathbf{x}_i^{t+1}, \mathbf{c}_i^t, \mathbf{w}^t) + \left( \frac{1}{2\eta_1} - \frac{\lambda_p(L_{D_1} + L_{DQ_1}) + (1 - \lambda_p)(L + G^{(i)} L_{Q_1} + G_{Q_1}^{(i)} L l_{Q_1})}{2} \right) \|\mathbf{x}_i^{t+1} - \mathbf{x}_i^t\|^2$$

$$\overset{(a)}{\leq} (1 - \lambda_p) \left( f_i(\mathbf{x}_i^t) + f_i(\widetilde{Q}_{\mathbf{c}_i^t}(\mathbf{x}_i^t)) \right) + \lambda_p \left( f_i^{KD}(\mathbf{x}_i^t, \mathbf{w}^t) + f_i^{KD}(\widetilde{Q}_{\mathbf{c}_i^t}(\mathbf{x}_i^t), \mathbf{w}^t) \right) + A(\mathbf{x}_i^{t+1})$$

$$+ \frac{\lambda_p}{2} \|\nabla_{\mathbf{x}_i^t} f_i^{KD}(\mathbf{x}_i^t, \mathbf{w}^t) - \nabla_{\mathbf{x}_i^t} f_i^{KD}(\mathbf{x}_i^t, \mathbf{w}_i^t)\|^2 + \frac{\lambda_p}{2} \|\nabla_{\mathbf{x}_i^t} f_i^{KD}(\widetilde{Q}_{\mathbf{c}_i^t}(\mathbf{x}_i^t), \mathbf{w}^t)$$

$$- \nabla_{\mathbf{x}_i^t} f_i^{KD}(\widetilde{Q}_{\mathbf{c}_i^t}(\mathbf{x}_i^t), \mathbf{w}_i^t)\|^2 + \lambda_p \|\mathbf{x}_i^{t+1} - \mathbf{x}_i^t\|^2$$

$$\overset{(b)}{\leq} (1 - \lambda_p) \left( f_i(\mathbf{x}_i^t) + f_i(\widetilde{Q}_{\mathbf{c}_i^t}(\mathbf{x}_i^t)) \right) + \lambda_p \left( f_i^{KD}(\mathbf{x}_i^t, \mathbf{w}^t) + f_i^{KD}(\widetilde{Q}_{\mathbf{c}_i^t}(\mathbf{x}_i^t), \mathbf{w}^t) \right) + \lambda R(\mathbf{x}_i^t, \mathbf{c}_i^t)$$

$$+ \frac{\lambda_p}{2} \|\nabla f_i^{KD}(\mathbf{x}_i^t, \mathbf{w}^t) - \nabla f_i^{KD}(\mathbf{x}_i^t, \mathbf{w}_i^t)\|^2 + \frac{\lambda_p}{2} \|\nabla f_i^{KD}(\widetilde{Q}_{\mathbf{c}_i^t}(\mathbf{x}_i^t), \mathbf{w}^t)$$

$$- \nabla f_i^{KD}(\widetilde{Q}_{\mathbf{c}_i^t}(\mathbf{x}_i^t), \mathbf{w}_i^t)\|^2 + \lambda_p \|\mathbf{x}_i^{t+1} - \mathbf{x}_i^t\|^2$$

$$\overset{(c)}{\leq} (1 - \lambda_p) \left( f_i(\mathbf{x}_i^t) + f_i(\widetilde{Q}_{\mathbf{c}_i^t}(\mathbf{x}_i^t)) \right) + \lambda_p \left( f_i^{KD}(\mathbf{x}_i^t, \mathbf{w}^t) + f_i^{KD}(\widetilde{Q}_{\mathbf{c}_i^t}(\mathbf{x}_i^t), \mathbf{w}^t) \right)$$

$$+ \lambda R(\mathbf{x}_i^t, \mathbf{c}_i^t) + \frac{\lambda_p(L_D^2 + L_{DQ}^2)}{2} \|\mathbf{w}^t - \mathbf{w}_i^t\|^2 + \lambda_p \|\mathbf{x}_i^{t+1} - \mathbf{x}_i^t\|^2$$

$$= F_i(\mathbf{x}_i^t, \mathbf{c}_i^t, \mathbf{w}^t) + \frac{\lambda_p(L_D^2 + L_{DQ}^2)}{2} \|\mathbf{w}_i^t - \mathbf{w}^t\|^2 + \lambda_p \|\mathbf{x}_i^{t+1} - \mathbf{x}_i^t\|^2. \tag{21}$$

To obtain (a), we substituted the value of $A(\mathbf{x}_i^{t+1})$ from Claim 9 into (20). In (b), we used $A(\mathbf{x}_i^{t+1}) \leq \lambda R(\mathbf{x}_i^t, \mathbf{c}_i^t)$, $\|\nabla_{\mathbf{x}_i^t} f_i^{KD}(\mathbf{x}_i^t, \mathbf{w}^t) - \nabla_{\mathbf{x}_i^t} f_i^{KD}(\mathbf{x}_i^t, \mathbf{w}_i^t)\|^2 \leq \|\nabla f_i^{KD}(\mathbf{x}_i^t, \mathbf{w}^t) - \nabla f_i^{KD}(\mathbf{x}_i^t, \mathbf{w}_i^t)\|^2$ and the fact that $\|\nabla_{\mathbf{x}_i^t} f_i^{KD}(\widetilde{Q}_{\mathbf{c}_i^t}(\mathbf{x}_i^t)), \mathbf{w}^t) - \nabla_{\mathbf{x}_i^t} f_i^{KD}(\widetilde{Q}_{\mathbf{c}_i^t}(\mathbf{x}_i^t)), \mathbf{w}_i^t)\|^2 \leq \|\nabla f_i^{KD}(\mathbf{x}_i^t, \mathbf{w}^t) - \nabla f_i^{KD}(\mathbf{x}_i^t, \mathbf{w}_i^t)\|^2$. And in (c) we used the assumption that $f_i^{KD}(\mathbf{x}, \mathbf{w})$ is $L_D$-smooth and $f_i^{KD}(\widetilde{Q}_{\mathbf{c}}(\mathbf{x}), \mathbf{w})$ is $L_{DQ}$-smooth.

Substituting $\eta_1 = \frac{1}{2(\lambda_p(2 + L_{D_1} + L_{DQ_1}) + (1 - \lambda_p)(L + G^{(i)} L_{Q_1} + G_{Q_1}^{(i)} Ll_{Q_1}))}$ in (21) gives:

$$F_i(\mathbf{x}_i^{t+1}, \mathbf{c}_i^t, \mathbf{w}^t) + \left( \frac{\lambda_p(2 + L_{D_1} + L_{DQ_1}) + (1 - \lambda_p)(L + G^{(i)} L_{Q_1} + G_{Q_1}^{(i)} Ll_{Q_1})}{2} \right) \|\mathbf{x}_i^{t+1} - \mathbf{x}_i^t\|^2$$

$$\leq F_i(\mathbf{x}_i^t, \mathbf{c}_i^t, \mathbf{w}^t) + \frac{\lambda_p(L_D^2 + L_{DQ}^2)}{2} \|\mathbf{w}_i^t - \mathbf{w}^t\|^2. \tag{22}$$

### C.1.2 Sufficient Decrease Due to $\mathbf{c}_i$

In parallel with Claim 8, we have following smoothness result for $\mathbf{c}$:

**Claim 10.** $(1 - \lambda_p) f_i(\widetilde{Q}_{\mathbf{c}}(\mathbf{x})) + \lambda_p f_i^{KD}(\widetilde{Q}_{\mathbf{c}}(\mathbf{x}), \mathbf{w}))$ is $(\lambda_p L_{DQ_2} + (1 - \lambda_p)(G^{(i)} L_{Q_2} + G_{Q_2}^{(i)} Ll_{Q_2}))$-smooth with respect to $\mathbf{c}$.

From Claim 10 we have:

$$F_i(\mathbf{x}_i^{t+1}, \mathbf{c}_i^{t+1}, \mathbf{w}^t) + \left( \frac{1}{2\eta_2} - \frac{\lambda_p L_{DQ_2} + (1 - \lambda_p)(G^{(i)} L_{Q_2} + G_{Q_2}^{(i)} Ll_{Q_2})}{2} \right) \|\mathbf{c}_i^{t+1} - \mathbf{c}_i^t\|^2$$

$$= (1 - \lambda_p) \left( f_i(\mathbf{x}_i^{t+1}) + f_i(\widetilde{Q}_{\mathbf{c}_i^{t+1}}(\mathbf{x}_i^{t+1})) \right)$$

$$+ \lambda R(\mathbf{x}_i^{t+1}, \mathbf{c}_i^{t+1}) + \lambda_p \left( f_i^{KD}(\mathbf{x}_i^{t+1}, \mathbf{w}^t) + f_i^{KD}(\widetilde{Q}_{\mathbf{c}_i^{t+1}}(\mathbf{x}_i^{t+1})), \mathbf{w}^t) \right)$$

$$+ \left( \frac{1}{2\eta_2} - \frac{\lambda_p L_{DQ_2} + (1 - \lambda_p)(G^{(i)} L_{Q_2} + G_{Q_2}^{(i)} Ll_{Q_2})}{2} \right) \|\mathbf{c}_i^{t+1} - \mathbf{c}_i^t\|^2$$

$$\leq (1 - \lambda_p) \left( f_i(\mathbf{x}_i^{t+1}) + f_i(\widetilde{Q}_{\mathbf{c}_i^t}(\mathbf{x}_i^{t+1})) \right) + \lambda R(\mathbf{x}_i^{t+1}, \mathbf{c}_i^{t+1})$$

$$+ \lambda_p \left( f_i^{KD}(\mathbf{x}_i^{t+1}, \mathbf{w}^t) + f_i^{KD}(\widetilde{Q}_{\mathbf{c}_i^t}(\mathbf{x}_i^{t+1})), \mathbf{w}^t) \right) + \frac{1}{2\eta_2} \|\mathbf{c}_i^{t+1} - \mathbf{c}_i^t\|^2$$

$$+ (1 - \lambda_p) \left\langle \nabla_{\mathbf{c}_i^t} f_i(\widetilde{Q}_{\mathbf{c}_i^t}(\mathbf{x}_i^{t+1})), \mathbf{c}_i^{t+1} - \mathbf{c}_i^t \right\rangle + \lambda_p \left\langle \nabla_{\mathbf{c}_i^t} f_i^{KD}(\widetilde{Q}_{\mathbf{c}_i^t}(\mathbf{x}_i^{t+1}), \mathbf{w}^t), \mathbf{c}_i^{t+1} - \mathbf{c}_i^t \right\rangle$$

$$= (1 - \lambda_p) \left( f_i(\mathbf{x}_i^{t+1}) + f_i(\widetilde{Q}_{\mathbf{c}_i^t}(\mathbf{x}_i^{t+1})) \right) + \lambda R(\mathbf{x}_i^{t+1}, \mathbf{c}_i^{t+1})$$

$$+ \lambda_p \left( f_i^{KD}(\mathbf{x}_i^{t+1}, \mathbf{w}^t) + f_i^{KD}(\widetilde{Q}_{\mathbf{c}_i^t}(\mathbf{x}_i^{t+1})), \mathbf{w}^t) \right) + \frac{1}{2\eta_2} \|\mathbf{c}_i^{t+1} - \mathbf{c}_i^t\|^2$$

$$+ (1 - \lambda_p)\left\langle \nabla_{\mathbf{c}_i^t} f_i(\widetilde{Q}_{\mathbf{c}_i^t}(\mathbf{x}_i^{t+1})), \mathbf{c}_i^{t+1} - \mathbf{c}_i^t \right\rangle + \lambda_p \left\langle \nabla_{\mathbf{c}_i^t} f_i^{KD}(\widetilde{Q}_{\mathbf{c}_i^t}(\mathbf{x}_i^{t+1}), \mathbf{w}_i^t), \mathbf{c}_i^{t+1} - \mathbf{c}_i^t \right\rangle$$

$$+ \lambda_p \left\langle \nabla_{\mathbf{c}_i^t} f_i^{KD}(\widetilde{Q}_{\mathbf{c}_i^t}(\mathbf{x}_i^{t+1}), \mathbf{w}^t) - \nabla_{\mathbf{c}_i^t} f_i^{KD}(\widetilde{Q}_{\mathbf{c}_i^t}(\mathbf{x}_i^{t+1}), \mathbf{w}_i^t), \mathbf{c}_i^{t+1} - \mathbf{c}_i^t \right\rangle$$

$$\leq (1 - \lambda_p)\left( f_i(\mathbf{x}_i^{t+1}) + f_i(\widetilde{Q}_{\mathbf{c}_i^t}(\mathbf{x}_i^{t+1})) \right) + \lambda R(\mathbf{x}_i^{t+1}, \mathbf{c}_i^{t+1})$$

$$+ \lambda_p \left( f_i^{KD}(\mathbf{x}_i^{t+1}, \mathbf{w}^t) + f_i^{KD}(\widetilde{Q}_{\mathbf{c}_i^t}(\mathbf{x}_i^{t+1}), \mathbf{w}^t) \right) + \frac{1}{2\eta_2}\|\mathbf{c}_i^{t+1} - \mathbf{c}_i^t\|^2$$

$$+ (1 - \lambda_p)\left\langle \nabla_{\mathbf{c}_i^t} f_i(\widetilde{Q}_{\mathbf{c}_i^t}(\mathbf{x}_i^{t+1})), \mathbf{c}_i^{t+1} - \mathbf{c}_i^t \right\rangle + \lambda_p \left\langle \nabla_{\mathbf{c}_i^t} f_i^{KD}(\widetilde{Q}_{\mathbf{c}_i^t}(\mathbf{x}_i^{t+1}), \mathbf{w}_i^t), \mathbf{c}_i^{t+1} - \mathbf{c}_i^t \right\rangle$$

$$+ \frac{\lambda_p}{2}\|\nabla_{\mathbf{c}_i^t} f_i^{KD}(\widetilde{Q}_{\mathbf{c}_i^t}(\mathbf{x}_i^{t+1}), \mathbf{w}^t) - \nabla_{\mathbf{c}_i^t} f_i^{KD}(\widetilde{Q}_{\mathbf{c}_i^t}(\mathbf{x}_i^{t+1}), \mathbf{w}_i^t)\|^2 + \frac{\lambda_p}{2}\|\mathbf{c}_i^{t+1} - \mathbf{c}_i^t\|^2. \quad (23)$$

in the last inequality we used:

$$\left\langle \lambda_p(\nabla_{\mathbf{c}_i^t} f_i^{KD}(\widetilde{Q}_{\mathbf{c}_i^t}(\mathbf{x}_i^{t+1}), \mathbf{w}^t) - \nabla_{\mathbf{c}_i^t} f_i^{KD}(\widetilde{Q}_{\mathbf{c}_i^t}(\mathbf{x}_i^{t+1}), \mathbf{w}_i^t)), \mathbf{c}_i^{t+1} - \mathbf{c}_i^t \right\rangle$$

$$\leq \frac{\lambda_p}{2}\|\nabla_{\mathbf{c}_i^t} f_i^{KD}(\widetilde{Q}_{\mathbf{c}_i^t}(\mathbf{x}_i^{t+1}), \mathbf{w}^t) - \nabla_{\mathbf{c}_i^t} f_i^{KD}(\widetilde{Q}_{\mathbf{c}_i^t}(\mathbf{x}_i^{t+1}), \mathbf{w}_i^t)\|^2$$

$$+ \frac{\lambda_p}{2}\|\mathbf{c}_i^{t+1} - \mathbf{c}_i^t\|^2$$

**Claim 11.** *Let*

$$B(\mathbf{c}_i^{t+1}) := \lambda R(\mathbf{x}_i^{t+1}, \mathbf{c}_i^{t+1}) + (1 - \lambda_p)\left\langle \nabla_{\mathbf{c}_i^t} f_i(\widetilde{Q}_{\mathbf{c}_i^t}(\mathbf{x}_i^{t+1})), \mathbf{c}_i^{t+1} - \mathbf{c}_i^t \right\rangle$$

$$+ \lambda_p \left\langle \nabla_{\mathbf{c}_i^t} f_i^{KD}(\widetilde{Q}_{\mathbf{c}_i^t}(\mathbf{x}_i^{t+1}), \mathbf{w}_i^t), \mathbf{c}_i^{t+1} - \mathbf{c}_i^t \right\rangle + \frac{1}{2\eta_2}\|\mathbf{c}_i^{t+1} - \mathbf{c}_i^t\|^2$$

$$B(\mathbf{c}_i^t) := \lambda R(\mathbf{x}_i^{t+1}, \mathbf{c}_i^t).$$

*Then* $B(\mathbf{c}_i^{t+1}) \leq B(\mathbf{c}_i^t)$.

*Proof.* Let $B(\mathbf{c})$ denote the expression inside the $\arg\min$ in (19) and we know that (19) is minimized when $\mathbf{c} = \mathbf{c}_i^{t+1}$. So we have $B(\mathbf{c}_i^{t+1}) \leq B(\mathbf{c}_i^t)$. This proves the claim. $\square$

Substituting the bound from Claim 11 in (23),

$$F_i(\mathbf{x}_i^{t+1}, \mathbf{c}_i^{t+1}, \mathbf{w}^t) + \left(\frac{1}{2\eta_2} - \frac{\lambda_p L_{DQ_2} + (1 - \lambda_p)(G^{(i)} L_{Q_2} + G_{Q_2}^{(i)} L l_{Q_2})}{2}\right)\|\mathbf{c}_i^{t+1} - \mathbf{c}_i^t\|^2$$

$$\leq (1 - \lambda_p)\left( f_i(\mathbf{x}_i^{t+1}) + f_i(\widetilde{Q}_{\mathbf{c}_i^t}(\mathbf{x}_i^{t+1})) \right) + \lambda_p \left( f_i^{KD}(\mathbf{x}_i^{t+1}, \mathbf{w}^t) + f_i^{KD}(\widetilde{Q}_{\mathbf{c}_i^t}(\mathbf{x}_i^{t+1}), \mathbf{w}^t) \right)$$

$$+ \frac{\lambda_p}{2}\|\nabla_{\mathbf{c}_i^t} f_i^{KD}(\widetilde{Q}_{\mathbf{c}_i^t}(\mathbf{x}_i^{t+1}), \mathbf{w}^t) - \nabla_{\mathbf{c}_i^t} f_i^{KD}(\widetilde{Q}_{\mathbf{c}_i^t}(\mathbf{x}_i^{t+1}), \mathbf{w}_i^t)\|^2$$

$$+ B(\mathbf{c}_i^{t+1}) + \frac{\lambda_p}{2}\|\mathbf{c}_i^{t+1} - \mathbf{c}_i^t\|^2$$

$$\leq (1 - \lambda_p)\left( f_i(\mathbf{x}_i^{t+1}) + f_i(\widetilde{Q}_{\mathbf{c}_i^t}(\mathbf{x}_i^{t+1})) \right) + \lambda_p \left( f_i^{KD}(\mathbf{x}_i^{t+1}, \mathbf{w}^t) + f_i^{KD}(\widetilde{Q}_{\mathbf{c}_i^t}(\mathbf{x}_i^{t+1}), \mathbf{w}^t) \right)$$

$$+ \frac{\lambda_p}{2}\|\nabla f_i^{KD}(\widetilde{Q}_{\mathbf{c}_i^t}(\mathbf{x}_i^{t+1}), \mathbf{w}^t) - \nabla f_i^{KD}(\widetilde{Q}_{\mathbf{c}_i^t}(\mathbf{x}_i^{t+1}), \mathbf{w}_i^t)\|^2 + R(\mathbf{x}_i^{t+1}, \mathbf{c}_i^t)$$

$$+ \frac{\lambda_p}{2}\|\mathbf{c}_i^{t+1} - \mathbf{c}_i^t\|^2$$

$$\leq (1 - \lambda_p)\left( f_i(\mathbf{x}_i^{t+1}) + f_i(\widetilde{Q}_{\mathbf{c}_i^t}(\mathbf{x}_i^{t+1})) \right) + \lambda_p \left( f_i^{KD}(\mathbf{x}_i^{t+1}, \mathbf{w}^t) + f_i^{KD}(\widetilde{Q}_{\mathbf{c}_i^t}(\mathbf{x}_i^{t+1}), \mathbf{w}^t) \right)$$

$$+ \frac{\lambda_p L_{DQ}^2}{2}\|\mathbf{w}^t - \mathbf{w}_i^t\|^2 + R(\mathbf{x}_i^{t+1}, \mathbf{c}_i^t) + \frac{\lambda_p}{2}\|\mathbf{c}_i^{t+1} - \mathbf{c}_i^t\|^2$$

$$= F_i(\mathbf{x}_i^{t+1}, \mathbf{c}_i^t, \mathbf{w}^t) + \frac{\lambda_p L_{DQ}^2}{2}\|\mathbf{w}^t - \mathbf{w}_i^t\|^2 + \frac{\lambda_p}{2}\|\mathbf{c}_i^{t+1} - \mathbf{c}_i^t\|^2$$

Substituting $\eta_2 = \frac{1}{2(\lambda_p(1+L_{DQ_2})+(1-\lambda_p)(G^{(i)}L_{Q_2}+G^{(i)}_{Q_2}Ll_{Q_2}))}$ gives us:

$$F_i(\mathbf{x}_i^{t+1}, \mathbf{c}_i^{t+1}, \mathbf{w}^t) + \frac{\lambda_p(1+L_{DQ_2})+(1-\lambda_p)(G^{(i)}L_{Q_2}+G^{(i)}_{Q_2}Ll_{Q_2})}{2}\|\mathbf{c}_i^{t+1}-\mathbf{c}_i^t\|^2$$
$$\leq F_i(\mathbf{x}_i^{t+1}, \mathbf{c}_i^t, \mathbf{w}^t) + \frac{\lambda_p L_{DQ}^2}{2}\|\mathbf{w}^t - \mathbf{w}_i^t\|^2 \qquad (24)$$

### C.1.3 Sufficient Decrease Due to w

Now, we use $(\lambda_p(L_{D_2}+L_{DQ_3}))$-smoothness of $F_i(\mathbf{x}, \mathbf{c}, \mathbf{w})$ with respect to $\mathbf{w}$:

$$F_i(\mathbf{x}_i^{t+1}, \mathbf{c}_i^{t+1}, \mathbf{w}^{t+1}) \leq F_i(\mathbf{x}_i^{t+1}, \mathbf{c}_i^{t+1}, \mathbf{w}^t) + \langle \nabla_{\mathbf{w}^t}F_i(\mathbf{x}_i^{t+1}, \mathbf{c}_i^{t+1}, \mathbf{w}^t), \mathbf{w}^{t+1} - \mathbf{w}^t \rangle$$
$$+ \frac{\lambda_p(L_{D_2}+L_{DQ_3})}{2}\|\mathbf{w}^{t+1}-\mathbf{w}^t\|^2$$

After some algebraic manipulations (see Appendix C.3) we have:

$$F_i(\mathbf{x}_i^{t+1}, \mathbf{c}_i^{t+1}, \mathbf{w}^{t+1}) + (\frac{\eta_3}{2} - \lambda_p(L_{D_2}+L_{DQ_3})\eta_3^2)\left\|\nabla_{\mathbf{w}^t}F_i(\mathbf{x}_i^{t+1}, \mathbf{c}_i^{t+1}, \mathbf{w}^t)\right\|^2$$
$$\leq F_i(\mathbf{x}_i^{t+1}, \mathbf{c}_i^{t+1}, \mathbf{w}^t)$$
$$+ (\eta_3 + 2\lambda_p(L_{D_2}+L_{DQ_3})\eta_3^2)\left\|\mathbf{g}^t - \nabla_{\mathbf{w}_i^t}F_i(\mathbf{x}_i^{t+1}, \mathbf{c}_i^{t+1}, \mathbf{w}_i^t)\right\|^2$$
$$+ (\eta_3 + 2\lambda_p(L_{D_2}+L_{DQ_3})\eta_3^2)(\lambda_p(L_{D_2}+L_{DQ_3}))^2\left\|\mathbf{w}_i^t - \mathbf{w}^t\right\|^2$$
$$(25)$$

### C.1.4 Overall Decrease

Let $L_x^{(i)}, L_c^{(i)}$ for any $i \in [n]$ and $L_w$ are defined as follows:

$$L_x^{(i)} = (1-\lambda_p)(L + G^{(i)}L_{Q_1} + G^{(i)}_{Q_1}Ll_{Q_1}) + \lambda_p(2 + L_{D_1} + L_{DQ_1}) \qquad (26)$$
$$L_c^{(i)} = (1-\lambda_p)(G^{(i)}L_{Q_2} + G^{(i)}_{Q_2}Ll_{Q_2}) + \lambda_p(1 + L_{DQ_2}) \qquad (27)$$
$$L_w = L_{D_2} + L_{DQ_3}. \qquad (28)$$

Summing (22), (24), (25) we get the overall decrease property:

$$F_i(\mathbf{x}_i^{t+1}, \mathbf{c}_i^{t+1}, \mathbf{w}^{t+1}) + (\frac{\eta_3}{2} - \lambda_p L_w \eta_3^2)\left\|\nabla_{\mathbf{w}^t}F_i(\mathbf{x}_i^{t+1}, \mathbf{c}_i^{t+1}, \mathbf{w}^t)\right\|^2 + \frac{L_x}{2}\|\mathbf{x}_i^{t+1} - \mathbf{x}_i^t\|^2$$
$$+ \frac{L_c}{2}\|\mathbf{c}_i^{t+1} - \mathbf{c}_i^t\|^2 \leq (\eta_3 + 2\lambda_p L_w \eta_3^2)\left\|\mathbf{g}^t - \nabla_{\mathbf{w}_i^t}F_i(\mathbf{x}_i^{t+1}, \mathbf{c}_i^{t+1}, \mathbf{w}_i^t)\right\|^2$$
$$+ (L_{DQ}^2 + \frac{L_D^2}{2} + \eta_3\lambda_p L_w^2 + 2\lambda_p^2 L_w^3 \eta_3^2)\lambda_p\|\mathbf{w}_i^t - \mathbf{w}^t\|^2 + F_i(\mathbf{x}_i^t, \mathbf{c}_i^t, \mathbf{w}^t)$$
$$(29)$$

Let $L_{\min}^{(i)}$ for any $i \in [n]$ and $L_{\min}$ are defined as follows:

$$L_{\min}^{(i)} = \min\{L_x^{(i)}, L_c^{(i)}, (\eta_3 - 2\lambda_p L_w \eta_3^2)\} \qquad (30)$$
$$L_{\min} = \min\{L_{\min}^{(i)} : i \in [n]\}. \qquad (31)$$

Then,

$$F_i(\mathbf{x}_i^{t+1}, \mathbf{c}_i^{t+1}, \mathbf{w}^{t+1}) + \frac{L_{\min}}{2}\left(\left\|\nabla_{\mathbf{w}^t}F_i(\mathbf{x}_i^{t+1}, \mathbf{c}_i^{t+1}, \mathbf{w}^t)\right\|^2 + \|\mathbf{x}_i^{t+1} - \mathbf{x}_i^t\|^2 + \|\mathbf{c}_i^{t+1} - \mathbf{c}_i^t\|^2\right)$$
$$\leq (\eta_3 + 2\lambda_p L_w \eta_3^2)\left\|\mathbf{g}^t - \nabla_{\mathbf{w}_i^t}F_i(\mathbf{x}_i^{t+1}, \mathbf{c}_i^{t+1}, \mathbf{w}_i^t)\right\|^2$$
$$+ (L_{DQ}^2 + \frac{L_D^2}{2} + \eta_3\lambda_p L_w^2 + 2\lambda_p^2 L_w^3 \eta_3^2)\lambda_p\|\mathbf{w}_i^t - \mathbf{w}^t\|^2 + F_i(\mathbf{x}_i^t, \mathbf{c}_i^t, \mathbf{w}^t) \qquad (32)$$

We have obtained the sufficient decrease property for the alternating steps; now, we need to arrive at the first order stationarity of the gradient of general loss function. To do this we move on with bounding the gradients with respect to each type of variables.

## C.2 Bound on the Gradient

Now, we will use the first order optimality conditions due to proximal updates and bound the partial gradients with respect to variables $\mathbf{x}$ and $\mathbf{c}$. After obtaining bounds for partial gradients we will bound the overall gradient and use our results from Section C.1 to arrive at the final bound.

### C.2.1 Bound on the Gradient w.r.t. $\mathbf{x}_i$

Taking the derivative inside the minimization problem (18) with respect to $\mathbf{x}$ at $\mathbf{x} = \mathbf{x}_i^{t+1}$ and setting it to 0 gives the following optimality condition:

$$(1 - \lambda_p)\left(\nabla_{\mathbf{x}_i^t} f_i(\mathbf{x}_i^t) + \nabla_{\mathbf{x}_i^t} f_i(\widetilde{Q}_{\mathbf{c}_i^t}(\mathbf{x}_i^t))\right) + \lambda_p\left(\nabla_{\mathbf{x}_i^t} f_i^{KD}(\mathbf{x}_i^t, \mathbf{w}_i^t) + \nabla_{\mathbf{x}_i^t} f_i^{KD}(\widetilde{Q}_{\mathbf{c}_i^t}(\mathbf{x}_i^t), \mathbf{w}_i^t)\right)$$
$$+ \frac{1}{\eta_1}(\mathbf{x}_i^{t+1} - \mathbf{x}_i^t) + \lambda\nabla_{\mathbf{x}_i^{t+1}} R(\mathbf{x}_i^{t+1}, \mathbf{c}_i^t) = 0 \quad (33)$$

Then we have,

$$\left\|\nabla_{\mathbf{x}_i^{t+1}} F_i(\mathbf{x}_i^{t+1}, \mathbf{c}_i^t, \mathbf{w}^t)\right\| = \left\|(1 - \lambda_p)\left(\nabla_{\mathbf{x}_i^{t+1}} f_i(\mathbf{x}_i^{t+1}) + \nabla_{\mathbf{x}_i^{t+1}} f_i(\widetilde{Q}_{\mathbf{c}_i^t}(\mathbf{x}_i^{t+1}))\right)\right.$$
$$+ \lambda_p\left(\nabla_{\mathbf{x}_i^{t+1}} f_i^{KD}(\mathbf{x}_i^{t+1}, \mathbf{w}^t) + \nabla_{\mathbf{x}_i^{t+1}} f_i^{KD}(\widetilde{Q}_{\mathbf{c}_i^t}(\mathbf{x}_i^{t+1}), \mathbf{w}_i^t)\right)$$
$$\left. + \lambda\nabla_{\mathbf{x}_i^{t+1}} R(\mathbf{x}_i^{t+1}, \mathbf{c}_i^t)\right\|$$
$$\overset{(a)}{=} \left\|(1 - \lambda_p)\left(\nabla_{\mathbf{x}_i^{t+1}} f_i(\mathbf{x}_i^{t+1}) - \nabla_{\mathbf{x}_i^t} f_i(\mathbf{x}_i^t) + \nabla_{\mathbf{x}_i^{t+1}} f_i(\widetilde{Q}_{\mathbf{c}_i^t}(\mathbf{x}_i^{t+1}))\right.\right.$$
$$\left. - \nabla_{\mathbf{x}_i^t} f_i(\widetilde{Q}_{\mathbf{c}_i^t}(\mathbf{x}_i^t))\right) - \frac{1}{\eta_1}(\mathbf{x}_i^{t+1} - \mathbf{x}_i^t)$$
$$+ \lambda_p\left(\nabla_{\mathbf{x}_i^{t+1}} f_i^{KD}(\mathbf{x}_i^{t+1}, \mathbf{w}^t) - \nabla_{\mathbf{x}_i^t} f_i^{KD}(\mathbf{x}_i^t, \mathbf{w}_i^t)\right.$$
$$\left.\left. + \nabla_{\mathbf{x}_i^{t+1}} f_i^{KD}(\widetilde{Q}_{\mathbf{c}_i^t}(\mathbf{x}_i^{t+1}), \mathbf{w}^t) - \nabla_{\mathbf{x}_i^t} f_i^{KD}(\widetilde{Q}_{\mathbf{c}_i^t}(\mathbf{x}_i^t), \mathbf{w}_i^t)\right)\right\|$$
$$\overset{(b)}{\leq} \left(\frac{1}{\eta_1} + (1 - \lambda_p)(L + G^{(i)}L_{Q_1} + G_{Q_1}^{(i)}Ll_{Q_1})\right)\|\mathbf{x}_i^{t+1} - \mathbf{x}_i^t\|$$
$$+ \lambda_p\|\nabla_{\mathbf{x}^{t+1}} f_i^{KD}(\mathbf{x}_i^{t+1}, \mathbf{w}^t) - \nabla_{\mathbf{x}_i^t} f_i^{KD}(\mathbf{x}_i^t, \mathbf{w}_i^t)\|$$
$$+ \lambda_p\|\nabla_{\mathbf{x}_i^{t+1}} f_i^{KD}(\widetilde{Q}_{\mathbf{c}_i^t}(\mathbf{x}_i^{t+1}), \mathbf{w}^t) - \nabla_{\mathbf{x}_i^t} f_i^{KD}(\widetilde{Q}_{\mathbf{c}_i^t}(\mathbf{x}_i^t), \mathbf{w}^t)\|$$
$$\leq \left(\frac{1}{\eta_1} + (1 - \lambda_p)(L + G^{(i)}L_{Q_1} + G_{Q_1}^{(i)}Ll_{Q_1})\right)\|\mathbf{x}_i^{t+1} - \mathbf{x}_i^t\|$$
$$+ \lambda_p\|\nabla f_i^{KD}(\mathbf{x}_i^{t+1}, \mathbf{w}^t) - \nabla f_i^{KD}(\mathbf{x}_i^t, \mathbf{w}_i^t)\|$$
$$+ \lambda_p\|\nabla f_i^{KD}(\widetilde{Q}_{\mathbf{c}_i^t}(\mathbf{x}_i^{t+1}), \mathbf{w}^t) - \nabla f_i^{KD}(\widetilde{Q}_{\mathbf{c}_i^t}(\mathbf{x}_i^t), \mathbf{w}^t)\|$$
$$\overset{(c)}{\leq} \left(\frac{1}{\eta_1} + (1 - \lambda_p)(L + G^{(i)}L_{Q_1} + G_{Q_1}^{(i)}Ll_{Q_1})\right)\|\mathbf{x}_i^{t+1} - \mathbf{x}_i^t\|$$
$$+ \lambda_p(L_D + L_{DQ})(\|\mathbf{x}_i^{t+1} - \mathbf{x}_i^t\| + \|\mathbf{w}_i^t - \mathbf{w}^t\|)$$
$$= \left(\frac{1}{\eta_1} + \lambda_p(L_D + L_{DQ}) + (1 - \lambda_p)(L + G^{(i)}L_{Q_1} + G_{Q_1}^{(i)}Ll_{Q_1})\right)\|\mathbf{x}_i^{t+1} - \mathbf{x}_i^t\|$$
$$+ \lambda_p(L_D + L_{DQ})\|\mathbf{w}_i^t - \mathbf{w}^t\|$$

where (a) is from (33) by substituting the value of $\lambda\nabla_{\mathbf{x}_i^{t+1}} R(\mathbf{x}_i^{t+1}, \mathbf{c}_i^t)$, (b) is due to Claim 3 and **A.1**, and (c) is due to **A.7**. This implies:

$$\left\|\nabla_{\mathbf{x}_i^{t+1}} F_i(\mathbf{x}_i^{t+1}, \mathbf{c}_i^t, \mathbf{w}^t)\right\|^2 \leq 2(\lambda_p(L_D + L_{DQ}))^2\|\mathbf{w}_i^t - \mathbf{w}^t\|^2$$

$$+ 2\Big(\frac{1}{\eta_1} + \lambda_p(L_D + L_{DQ}) + (1 - \lambda_p)(L + G^{(i)}L_{Q_1} + G_{Q_1}^{(i)}Ll_{Q_1})\Big)^2 \|\mathbf{x}_i^{t+1} - \mathbf{x}_i^t\|^2$$

Substituting $\eta_1 = \frac{1}{2(\lambda_p(2 + L_{D_1} + L_{DQ_1}) + (1 - \lambda_p)(L + G^{(i)}L_{Q_1} + G_{Q_1}^{(i)}Ll_{Q_1}))}$ we have:

$$\Big\|\nabla_{\mathbf{x}_i^{t+1}} F_i(\mathbf{x}_i^{t+1}, \mathbf{c}_i^t, \mathbf{w}^t)\Big\|^2 \leq 2(\lambda_p(L_D + L_{DQ}))^2 \|\mathbf{w}_i^t - \mathbf{w}^t\|^2$$
$$+ 2\Big(\lambda_p(4 + 2L_{D_1} + 2L_{DQ_1} + L_D + L_{DQ})$$
$$+ 3(1 - \lambda_p)(L + G^{(i)}L_{Q_1} + G_{Q_1}^{(i)}Ll_{Q_1})\Big)^2 \|\mathbf{x}_i^{t+1} - \mathbf{x}_i^t\|^2$$
$$= 18\Big(\frac{\lambda_p}{3}(4 + 2L_{D_1} + 2L_{DQ_1} + L_D + L_{DQ})$$
$$+ (1 - \lambda_p)(L + G^{(i)}L_{Q_1} + G_{Q_1}^{(i)}Ll_{Q_1})\Big)^2 \|\mathbf{x}_i^{t+1} - \mathbf{x}_i^t\|^2$$
$$+ 2(\lambda_p(L_D + L_{DQ}))^2 \|\mathbf{w}_i^t - \mathbf{w}^t\|^2 \qquad (34)$$

### C.2.2 Bound on the Gradient w.r.t. $\mathbf{c}_i$

Similarly, taking the derivative inside the minimization problem (19) with respect to $\mathbf{c}$ at $\mathbf{c} = \mathbf{c}_i^{t+1}$ and setting it to 0 gives the following optimality condition:

$$(1 - \lambda_p)\nabla_{\mathbf{c}_i^t} f_i(\widetilde{Q}_{\mathbf{c}_i^t}(\mathbf{x}_i^{t+1})) + \lambda_p\nabla_{\mathbf{c}_i^t} f_i^{KD}(\widetilde{Q}_{\mathbf{c}_i^t}(\mathbf{x}_i^{t+1}), \mathbf{w}_i^t)$$
$$+ \frac{1}{\eta_2}(\mathbf{c}_i^{t+1} - \mathbf{c}_i^t) + \lambda\nabla_{\mathbf{c}_i^{t+1}} R(\mathbf{x}_i^{t+1}, \mathbf{c}_i^{t+1}) = 0 \quad (35)$$

Then we have

$$\Big\|\nabla_{\mathbf{c}_i^{t+1}} F_i(\mathbf{x}_i^{t+1}, \mathbf{c}_i^{t+1}, \mathbf{w}^t)\Big\| = \Big\|(1 - \lambda_p)\nabla_{\mathbf{c}_i^{t+1}} f_i(\widetilde{Q}_{\mathbf{c}_i^{t+1}}(\mathbf{x}_i^{t+1}))$$
$$+ \lambda_p\nabla_{\mathbf{c}_i^{t+1}} f_i^{KD}(\widetilde{Q}_{\mathbf{c}_i^{t+1}}(\mathbf{x}_i^{t+1}), \mathbf{w}^t) + \lambda\nabla_{\mathbf{c}_i^{t+1}} R(\mathbf{x}_i^{t+1}, \mathbf{c}_i^{t+1})\Big\|$$
$$\overset{(a)}{=} \Big\|(1 - \lambda_p)\Big(\nabla_{\mathbf{c}_i^{t+1}} f_i(\widetilde{Q}_{\mathbf{c}_i^{t+1}}(\mathbf{x}_i^{t+1})) - \nabla_{\mathbf{c}_i^t} f_i(\widetilde{Q}_{\mathbf{c}_i^t}(\mathbf{x}_i^{t+1}))\Big)$$
$$+ \lambda_p\Big(\nabla_{\mathbf{c}_i^{t+1}} f_i^{KD}(\widetilde{Q}_{\mathbf{c}_i^{t+1}}(\mathbf{x}_i^{t+1}), \mathbf{w}^t) - \nabla_{\mathbf{c}_i^t} f_i^{KD}(\widetilde{Q}_{\mathbf{c}_i^t}(\mathbf{x}_i^{t+1}), \mathbf{w}_i^t)\Big)$$
$$+ \frac{1}{\eta_2}(\mathbf{c}_i^t - \mathbf{c}_i^{t+1})\Big\|$$
$$\leq (1 - \lambda_p)\Big\|\nabla_{\mathbf{c}_i^{t+1}} f_i(\widetilde{Q}_{\mathbf{c}_i^{t+1}}(\mathbf{x}_i^{t+1})) - \nabla_{\mathbf{c}_i^t} f_i(\widetilde{Q}_{\mathbf{c}_i^t}(\mathbf{x}_i^{t+1}))\Big\|$$
$$+ \lambda_p\Big\|\nabla_{\mathbf{c}_i^{t+1}} f_i^{KD}(\widetilde{Q}_{\mathbf{c}_i^{t+1}}(\mathbf{x}_i^{t+1}), \mathbf{w}^t) - \nabla_{\mathbf{c}_i^t} f_i^{KD}(\widetilde{Q}_{\mathbf{c}_i^t}(\mathbf{x}_i^{t+1}), \mathbf{w}_i^t)\Big\|$$
$$+ \frac{1}{\eta_2}\|\mathbf{c}_i^t - \mathbf{c}_i^{t+1}\|$$
$$\leq (1 - \lambda_p)\Big\|\nabla_{\mathbf{c}_i^{t+1}} f_i(\widetilde{Q}_{\mathbf{c}_i^{t+1}}(\mathbf{x}_i^{t+1})) - \nabla_{\mathbf{c}_i^t} f_i(\widetilde{Q}_{\mathbf{c}_i^t}(\mathbf{x}_i^{t+1}))\Big\|$$
$$+ \lambda_p\Big\|\nabla f_i^{KD}(\widetilde{Q}_{\mathbf{c}_i^{t+1}}(\mathbf{x}_i^{t+1}), \mathbf{w}^t) - \nabla f_i^{KD}(\widetilde{Q}_{\mathbf{c}_i^t}(\mathbf{x}_i^{t+1}), \mathbf{w}_i^t)\Big\|$$
$$+ \frac{1}{\eta_2}\|\mathbf{c}_i^t - \mathbf{c}_i^{t+1}\|$$
$$\leq \Big(\frac{1}{\eta_2} + \lambda_p L_{DQ} + (1 - \lambda_p)(G^{(i)}L_{Q_2} + G_{Q_2}^{(i)}Ll_{Q_2})\Big)\|\mathbf{c}_i^{t+1} - \mathbf{c}_i^t\|$$
$$+ \lambda_p L_{DQ}\|\mathbf{w}^t - \mathbf{w}_i^t\|$$

where in (a) we substituted the value of $\lambda\nabla_{\mathbf{c}_i^{t+1}} R(\mathbf{x}_i^{t+1}, \mathbf{c}_i^{t+1})$ from (35) and the last inequality is due to Claim 4 and Assumption **A.7**. As a result we have,

$$\Big\|\nabla_{\mathbf{c}_i^{t+1}} F_i(\mathbf{x}_i^{t+1}, \mathbf{c}_i^{t+1}, \mathbf{w}^t)\Big\|^2 \leq 2\Big(\frac{1}{\eta_2} + \lambda_p L_{DQ} + (1 - \lambda_p)(G^{(i)}L_{Q_2} + G_{Q_2}^{(i)}Ll_{Q_2})\Big)^2 \|\mathbf{c}_i^{t+1} - \mathbf{c}_i^t\|^2$$

$$+ 2(\lambda_p L_{DQ})^2 \|\mathbf{w}^t - \mathbf{w}_i^t\|^2$$

substituting $\eta_2 = \frac{1}{2(\lambda_p(1+L_{DQ_2})+(1-\lambda_p)(G^{(i)}L_{Q_2}+G^{(i)}_{Q_2}Ll_{Q_2}))}$ we have:

$$
\begin{aligned}
\left\|\nabla_{\mathbf{c}_i^{t+1}} F_i(\mathbf{x}_i^{t+1}, \mathbf{c}_i^{t+1}, \mathbf{w}^t)\right\|^2 \leq 18\Big(&\frac{\lambda_p}{3}(2 + 2L_{DQ_2} + L_{DQ}) \\
&+ (1-\lambda_p)(G^{(i)}L_{Q_2} + G^{(i)}_{Q_2}Ll_{Q_2})\Big)^2 \|\mathbf{c}_i^{t+1} - \mathbf{c}_i^t\|^2 \\
&+ 2(\lambda_p L_{DQ})^2 \|\mathbf{w}^t - \mathbf{w}_i^t\|^2
\end{aligned}
\tag{36}
$$

### C.2.3 Overall Bound

Let $\|\mathbf{G}_i^t\|^2 = \left\|[\nabla_{\mathbf{x}_i^{t+1}} F_i(\mathbf{x}_i^{t+1}, \mathbf{c}_i^t, \mathbf{w}^t)^T, \nabla_{\mathbf{c}_i^{t+1}} F_i(\mathbf{x}_i^{t+1}, \mathbf{c}_i^{t+1}, \mathbf{w}^t)^T, \nabla_{\mathbf{w}^t} F_i(\mathbf{x}_i^{t+1}, \mathbf{c}_i^{t+1}, \mathbf{w}^t)^T]^T\right\|^2$.
Then,

$$
\begin{aligned}
\|\mathbf{G}_i^t\|^2 &= \left\|[\nabla_{\mathbf{x}_i^{t+1}} F_i(\mathbf{x}_i^{t+1}, \mathbf{c}_i^t, \mathbf{w}^t)^T, \nabla_{\mathbf{c}_i^{t+1}} F_i(\mathbf{x}_i^{t+1}, \mathbf{c}_i^{t+1}, \mathbf{w}^t)^T, \nabla_{\mathbf{w}^t} F_i(\mathbf{x}_i^{t+1}, \mathbf{c}_i^{t+1}, \mathbf{w}^t)^T]^T\right\|^2 \\
&= \left\|\nabla_{\mathbf{x}_i^{t+1}} F_i(\mathbf{x}_i^{t+1}, \mathbf{c}_i^t, \mathbf{w}^t)\right\|^2 + \left\|\nabla_{\mathbf{c}_i^{t+1}} F_i(\mathbf{x}_i^{t+1}, \mathbf{c}_i^{t+1}, \mathbf{w}^t)\right\|^2 + \left\|\nabla_{\mathbf{w}^t} F_i(\mathbf{x}_i^{t+1}, \mathbf{c}_i^{t+1}, \mathbf{w}^t)\right\|^2 \\
&\leq 18\Big(\frac{\lambda_p}{3}(4 + 2L_{D_1} + 2L_{DQ_1} + L_D + L_{DQ}) \\
&\quad + (1-\lambda_p)(L + G^{(i)}L_{Q_1} + G^{(i)}_{Q_1}Ll_{Q_1})\Big)^2 \|\mathbf{x}_i^{t+1} - \mathbf{x}_i^t\|^2 \\
&\quad + 18\Big(\frac{\lambda_p}{3}(2 + 2L_{DQ_2} + L_{DQ}) + (1-\lambda_p)(G^{(i)}L_{Q_2} + G^{(i)}_{Q_2}Ll_{Q_2})\Big)^2 \|\mathbf{c}_i^{t+1} - \mathbf{c}_i^t\|^2 \\
&\quad + 2(\lambda_p(L_D + 2L_{DQ}))^2 \|\mathbf{w}_i^t - \mathbf{w}^t\|^2 + \left\|\nabla_{\mathbf{w}^t} F_i(\mathbf{x}_i^{t+1}, \mathbf{c}_i^{t+1}, \mathbf{w}^t)\right\|^2
\end{aligned}
$$

where the last inequality is due to (34), (36) and using that $2(\lambda_p L_{DQ})^2 + 2(\lambda_p(L_D + L_{DQ}))^2 \leq 2(\lambda_p(L_D + 2L_{DQ}))^2$. Let

$$
\begin{aligned}
L_{\max}^{(i)} = \max\Big\{&\sqrt{\frac{1}{18}}, \Big(\frac{\lambda_p}{3}(2 + 2L_{DQ_2} + L_{DQ}) + (1-\lambda_p)(G^{(i)}L_{Q_2} + G^{(i)}_{Q_2}Ll_{Q_2})\Big), \\
&\Big(\frac{\lambda_p}{3}(4 + 2L_{D_1} + 2L_{DQ_1} + L_D + L_{DQ}) + (1-\lambda_p)(L + G^{(i)}L_{Q_1} + G^{(i)}_{Q_1}Ll_{Q_1})\Big)\Big\}
\end{aligned}
\tag{37}
$$

Then,

$$
\begin{aligned}
&\left\|[\nabla_{\mathbf{x}_i^{t+1}} F_i(\mathbf{x}_i^{t+1}, \mathbf{c}_i^t, \mathbf{w}^t)^T, \nabla_{\mathbf{c}_i^{t+1}} F_i(\mathbf{x}_i^{t+1}, \mathbf{c}_i^{t+1}, \mathbf{w}^t)^T, \nabla_{\mathbf{w}^t} F_i(\mathbf{x}_i^{t+1}, \mathbf{c}_i^{t+1}, \mathbf{w}^t)^T]^T\right\|^2 \\
&\leq 18(L_{\max}^{(i)})^2 \Big(\|\mathbf{x}_i^{t+1} - \mathbf{x}_i^t\|^2 + \|\mathbf{c}_i^{t+1} - \mathbf{c}_i^t\|^2 + \left\|\nabla_{\mathbf{w}^t} F_i(\mathbf{x}_i^{t+1}, \mathbf{c}_i^{t+1}, \mathbf{w}^t)\right\|^2\Big) \\
&\quad + 2(\lambda_p(L_D + 2L_{DQ}))^2 \|\mathbf{w}_i^t - \mathbf{w}^t\|^2 \\
&\overset{(a)}{\leq} 36\frac{(L_{\max}^{(i)})^2}{L_{\min}} \Big[(\eta_3 + 2\lambda_p L_w \eta_3^2)\left\|\mathbf{g}^t - \nabla_{\mathbf{w}_i^t} F_i(\mathbf{x}_i^{t+1}, \mathbf{c}_i^{t+1}, \mathbf{w}_i^t)\right\|^2 \\
&\quad + (L_{DQ}^2 + \frac{L_D^2}{2} + \eta_3\lambda_p L_w^2 + 2\lambda_p^2 L_w^3 \eta_3^2)\lambda_p \|\mathbf{w}_i^t - \mathbf{w}^t\|^2 \\
&\quad + F_i(\mathbf{x}_i^t, \mathbf{c}_i^t, \mathbf{w}^t) - F_i(\mathbf{x}_i^{t+1}, \mathbf{c}_i^{t+1}, \mathbf{w}^{t+1})\Big] + 2(\lambda_p(L_D + 2L_{DQ}))^2 \|\mathbf{w}_i^t - \mathbf{w}^t\|^2, \quad (38)
\end{aligned}
$$

where in (a) we use the bound from (32), and $L_{\min}$ is defined in (31).

Now we state a useful lemma that bounds the average deviation between the local versions of the global model at all clients, and the global model itself. See Appendix C.3 for a proof.

**Lemma 1.** *Let $\eta_3$ be chosen such that $\eta_3 \le \sqrt{\dfrac{1}{6\tau^2(\lambda_p L_w)^2\left(1+\frac{\overline{L}_{\max}^2}{(L_{\max}^{(\min)})^2}\right)}}$ where $L_{\max}^{(\min)} = \min\{L_{\max}^{(i)} :$*

*$i \in [n]\}$ and $\overline{L}_{\max} = \sqrt{\frac{1}{n}\sum_{i=1}^n (L_{\max}^{(i)})^2}$ (where $L_{\max}^{(i)}$ is defined in (37)), then we have,*

$$\frac{1}{T}\sum_{t=0}^{T-1}\frac{1}{n}\sum_{i=1}^n (L_{\max}^{(i)})^2\|\mathbf{w}^t - \mathbf{w}_i^t\|^2 \le \frac{1}{T}\sum_{t=0}^{T-1}\gamma_t \le 6\tau^2\eta_3^2\frac{1}{n}\sum_{i=1}^n(L_{\max}^{(i)})^2\kappa_i$$

As a corollary:

**Corollary 1.** *Recall, $\mathbf{g}^t = \frac{1}{n}\sum_{i=1}^n \nabla_{\mathbf{w}^t}F_i(\mathbf{x}_i^{t+1}, \mathbf{c}_i^{t+1}, \mathbf{w}_i^t)$. Then, we have:*

$$\frac{1}{T}\sum_{t=0}^{T-1}\frac{1}{n}\sum_{i=1}^n (L_{\max}^{(i)})^2\left\|\mathbf{g}^t - \nabla_{\mathbf{w}_i^t}F_i(\mathbf{x}_i^{t+1},\mathbf{c}_i^{t+1},\mathbf{w}_i^t)\right\|^2 \le 3\frac{1}{n}\sum_{i=1}^n(L_{\max}^{(i)})^2\kappa_i$$

$$+ 3(\lambda_p L_w)^2\left(1 + \frac{\overline{L}_{\max}^2}{(L_{\max}^{(\min)})^2}\right)6\tau^2\eta_3^2\frac{1}{n}\sum_{i=1}^n(L_{\max}^{(i)})^2\kappa_i,$$

*where $L_{\max}^{(i)}$ is defined in (37), and $\overline{L}_{\max}, L_{\max}^{(\min)}$ are defined in Lemma 1.*

Let $\overline{\kappa} := \frac{1}{n}\sum_{i=1}^n (L_{\max}^{(i)})^2\kappa_i$ and $C_L := 1 + \frac{\frac{1}{n}\sum_{i=1}^n (L_{\max}^{(i)})^2}{(\min_i\{L_{\max}^{(i)}\})^2}$ , using Lemma 1 and Corollary 1, summing the bound in (38) over time and clients, dividing by $T$ and $n$:

$$\frac{1}{T}\sum_{t=0}^{T-1}\frac{1}{n}\sum_{i=1}^n \|\mathbf{G}_i^t\|^2 \le \frac{36}{L_{\min}}\Bigg[(\eta_3 + 2\lambda_p L_w\eta_3^2)\times\left(3\overline{\kappa} + 3(\lambda_p L_w)^2 C_L 6\tau^2\eta_3^2\overline{\kappa}\right)$$

$$+ (L_{DQ}^2 + \frac{L_D^2}{2} + \eta_3\lambda_p L_w^2 + 2\lambda_p^2 L_w^3\eta_3^2)\lambda_p \times 6\tau^2\eta_3^2\overline{\kappa}$$

$$+ \frac{1}{T}\sum_{t=0}^{T-1}\frac{1}{n}\sum_{i=1}^n (L_{\max}^{(i)})^2\left(F_i(\mathbf{x}_i^t,\mathbf{c}_i^t,\mathbf{w}^t) - F_i(\mathbf{x}_i^{t+1},\mathbf{c}_i^{t+1},\mathbf{w}^{t+1})\right)\Bigg]$$

$$+ 2(\lambda_p(L_D + 2L_{DQ}))^2\times 6\tau^2\eta_3^2\overline{\kappa}$$

$$= \frac{36}{L_{\min}}\Big[3\tau^2\eta_3^2\overline{\kappa}(2\lambda_p L_{DQ}^2 + \lambda_p L_D^2 + \eta_3\lambda_p^2 L_w^2(2+6C_L) + \eta_3^2\lambda_p^3 L_w^3(4+12C_L))$$

$$+ 3\eta_3\overline{\kappa} + 6\lambda_p L_w\eta_3^2\overline{\kappa}\Big] + \frac{36}{L_{\min}}\frac{1}{n}\sum_{i=1}^n\frac{(L_{\max}^{(i)})^2\Delta_F^{(i)}}{T}$$

$$+ 12\lambda_p^2(L_D + 2L_{DQ})^2\tau^2\eta_3^2\overline{\kappa} \tag{39}$$

where $\Delta_F^{(i)} = F_i(\mathbf{x}_i^0,\mathbf{c}_i^0,\mathbf{w}_i^0) - F_i(\mathbf{x}_i^T,\mathbf{c}_i^T,\mathbf{w}_i^T)$.

**Choice of $\eta_3$.** Note that in Lemma 1 we chose $\eta_3$ such that $\eta_3 \le \sqrt{\frac{1}{6\tau^2\lambda_p^2 L_w^2 C_L}}$. Now, we further introduce upper bounds on $\eta_3$.

- We can choose $\eta_3$ small enough so that $L_{\min} = \eta_3 - 2\lambda_p L_w\eta_3^2$; see the definition of $L_{\min}$ in (31).
- We can choose $\eta_3$ small enough so that $\eta_3 - 2\lambda_p L_w\eta_3^2 \ge \frac{\eta_3}{2}$. This is equivalent to choosing $\eta_3 \le \frac{1}{4\lambda_p L_w}$.

These two choices imply $L_{\min} \ge \frac{\eta_3}{2}$.

In the end, we have 2 critical constraints on $\eta_3$, $\{\eta_3 : \eta_3 \le \sqrt{\frac{1}{6\tau^2\lambda_p^2 L_w^2 C_L}}, \eta_3 \le \frac{1}{4\lambda_p L_w}\}$ . Then, let $\{\eta_3 : \eta_3 \le \frac{1}{4\lambda_p L_w\tau\sqrt{C_L}}\}$. Moreover, choosing $\tau \le \sqrt{T}$ we can take $\eta_3 = \frac{1}{4\lambda_p L_w\sqrt{C_L}\sqrt{T}}$ this choice clearly satisfies the above constraints.

From (39) we have,

$$\frac{1}{T}\sum_{t=0}^{T-1}\frac{1}{n}\sum_{i=1}^{n}\left\|\mathbf{G}_i^t\right\|^2 \overset{(a)}{\leq} \frac{72}{\eta_3}\Big[3\tau^2\eta_3^2\overline{\kappa}(2\lambda_p L_{DQ}^2 + \lambda_p L_D^2 + \eta_3\lambda_p^2 L_w^2(2+6C_L) + \eta_3^2\lambda_p^3 L_w^3(4+12C_L))$$

$$+ 3\eta_3\overline{\kappa} + 6\lambda_p L_w \eta_3^2\overline{\kappa}\Big] + \frac{72}{\eta_3}\frac{1}{n}\sum_{i=1}^{n}\frac{(L_{\max}^{(i)})^2\Delta_F^{(i)}}{T}$$

$$+ 12\lambda_p^2(L_D + 2L_{DQ})^2\tau^2\eta_3^2\overline{\kappa}$$

$$= 72\Big[3\tau^2\eta_3\overline{\kappa}(2\lambda_p L_{DQ}^2 + \lambda_p L_D^2 + \eta_3\lambda_p^2 L_w^2(2+6C_L) + \eta_3^2\lambda_p^3 L_w^3(4+12C_L))$$

$$+ 3\overline{\kappa} + 6\lambda_p L_w \eta_3\overline{\kappa}\Big] + \frac{72}{\eta_3}\frac{1}{n}\sum_{i=1}^{n}\frac{(L_{\max}^{(i)})^2\Delta_F^{(i)}}{T}$$

$$+ 12\lambda_p^2(L_D + 2L_{DQ})^2\tau^2\eta_3^2\overline{\kappa}$$

In (a) we used $L_{\min} \geq \frac{\eta_3}{2}$. Now, we plug in $\eta_3 = \frac{1}{4\lambda_p L_w\sqrt{C_L}\sqrt{T}}$ then:

$$\frac{1}{T}\sum_{t=0}^{T-1}\frac{1}{n}\sum_{i=1}^{n}\left\|\mathbf{G}_i^t\right\|^2 \leq 72\Big[\frac{3}{4}\tau^2\overline{\kappa}\frac{L_D^2 + 2L_{DQ}^2}{\sqrt{C_L}L_w}\frac{1}{\sqrt{T}} + \frac{3(2+6C_L)}{16C_L}\tau^2\overline{\kappa}\frac{1}{T} + \frac{3(4+12C_L)}{64C_L^{\frac{3}{2}}}\tau^2\overline{\kappa}\frac{1}{T^{\frac{3}{2}}}$$

$$+ 3\overline{\kappa} + \frac{3}{2}\frac{\overline{\kappa}}{\sqrt{C_L}\sqrt{T}}\Big] + 288\lambda_p L_w\sqrt{C_L}\frac{1}{n}\sum_{i=1}^{n}\frac{(L_{\max}^{(i)})^2\Delta_F^{(i)}}{\sqrt{T}}$$

$$+ \frac{3}{4}\tau^2\overline{\kappa}\frac{(L_D + 2L_{DQ})^2}{C_L L_w^2}\frac{1}{T}$$

$$= \frac{54\frac{L_D^2 + 2L_{DQ}^2}{\sqrt{C_L}L_w}\tau^2\overline{\kappa} + \frac{108\overline{\kappa}}{\sqrt{C_L}} + 288\sqrt{C_L}\lambda_p L_w\frac{1}{n}\sum_{i=1}^{n}(L_{\max}^{(i)})^2\Delta_F^{(i)}}{\sqrt{T}}$$

$$+ \frac{\frac{27}{2}\frac{2+C_L}{C_L}\tau^2\overline{\kappa} + \frac{3(L_D+2L_{DQ})^2}{4C_L L_w^2}\tau^2\overline{\kappa}}{T} + \frac{\frac{27}{2}\frac{2+C_L}{C_L^{\frac{3}{2}}}\tau^2\overline{\kappa}}{T^{\frac{3}{2}}} + 216\overline{\kappa}$$

## C.3  Omitted Details in Proof of Theorem 2

*Obtaining (25),*

$$F_i(\mathbf{x}_i^{t+1}, \mathbf{c}_i^{t+1}, \mathbf{w}^{t+1}) \leq F_i(\mathbf{x}_i^{t+1}, \mathbf{c}_i^{t+1}, \mathbf{w}^t) + \big\langle\nabla_{\mathbf{w}^t}F_i(\mathbf{x}_i^{t+1}, \mathbf{c}_i^{t+1}, \mathbf{w}^t), \mathbf{w}^{t+1} - \mathbf{w}^t\big\rangle$$

$$+ \frac{\lambda_p(L_{D_2} + L_{DQ_3})}{2}\|\mathbf{w}^{t+1} - \mathbf{w}^t\|^2$$

$$= F_i(\mathbf{x}_i^{t+1}, \mathbf{c}_i^{t+1}, \mathbf{w}^t) - \big\langle\nabla_{\mathbf{w}^t}F_i(\mathbf{x}_i^{t+1}, \mathbf{c}_i^{t+1}, \mathbf{w}^t), \eta_3\mathbf{g}^t\big\rangle + \frac{\lambda_p(L_{D_2} + L_{DQ_3})}{2}\|\eta_3\mathbf{g}^t\|^2$$

$$= F_i(\mathbf{x}_i^{t+1}, \mathbf{c}_i^{t+1}, \mathbf{w}^t)$$

$$- \eta_3\big\langle\nabla_{\mathbf{w}^t}F_i(\mathbf{x}_i^{t+1}, \mathbf{c}_i^{t+1}, \mathbf{w}^t), \mathbf{g}^t - \nabla_{\mathbf{w}^t}F_i(\mathbf{x}_i^{t+1}, \mathbf{c}_i^{t+1}, \mathbf{w}^t) + \nabla_{\mathbf{w}^t}F_i(\mathbf{x}_i^{t+1}, \mathbf{c}_i^{t+1}, \mathbf{w}^t)\big\rangle$$

$$+ \frac{\lambda_p(L_{D_2} + L_{DQ_3})}{2}\eta_3^2\|\mathbf{g}^t - \nabla_{\mathbf{w}^t}F_i(\mathbf{x}_i^{t+1}, \mathbf{c}_i^{t+1}, \mathbf{w}^t) + \nabla_{\mathbf{w}^t}F_i(\mathbf{x}_i^{t+1}, \mathbf{c}_i^{t+1}, \mathbf{w}^t)\|^2$$

$$= F_i(\mathbf{x}_i^{t+1}, \mathbf{c}_i^{t+1}, \mathbf{w}^t) - \eta_3\left\|\nabla_{\mathbf{w}^t}F_i(\mathbf{x}_i^{t+1}, \mathbf{c}_i^{t+1}, \mathbf{w}^t)\right\|^2$$

$$- \eta_3\big\langle\nabla_{\mathbf{w}^t}F_i(\mathbf{x}_i^{t+1}, \mathbf{c}_i^{t+1}, \mathbf{w}^t), \mathbf{g}^t - \nabla_{\mathbf{w}^t}F_i(\mathbf{x}_i^{t+1}, \mathbf{c}_i^{t+1}, \mathbf{w}^t)\big\rangle$$

$$+ \frac{\lambda_p(L_{D_2} + L_{DQ_3})}{2}\eta_3^2\|\mathbf{g}^t - \nabla_{\mathbf{w}^t}F_i(\mathbf{x}_i^{t+1}, \mathbf{c}_i^{t+1}, \mathbf{w}^t) + \nabla_{\mathbf{w}^t}F_i(\mathbf{x}_i^{t+1}, \mathbf{c}_i^{t+1}, \mathbf{w}^t)\|^2$$

$$\leq F_i(\mathbf{x}_i^{t+1}, \mathbf{c}_i^{t+1}, \mathbf{w}^t) - \eta_3\left\|\nabla_{\mathbf{w}^t}F_i(\mathbf{x}_i^{t+1}, \mathbf{c}_i^{t+1}, \mathbf{w}^t)\right\|^2 + \frac{\eta_3}{2}\left\|\nabla_{\mathbf{w}^t}F_i(\mathbf{x}_i^{t+1}, \mathbf{c}_i^{t+1}, \mathbf{w}^t)\right\|^2$$

$$+ \frac{\eta_3}{2}\left\|\mathbf{g}^t - \nabla_{\mathbf{w}^t}F_i(\mathbf{x}_i^{t+1}, \mathbf{c}_i^{t+1}, \mathbf{w}^t)\right\|^2 + \lambda_p(L_{D_2} + L_{DQ_3})\eta_3^2\left\|\mathbf{g}^t - \nabla_{\mathbf{w}^t}F_i(\mathbf{x}_i^{t+1}, \mathbf{c}_i^{t+1}, \mathbf{w}^t)\right\|^2$$

$$+ \lambda_p(L_{D_2} + L_{DQ_3})\eta_3^2 \left\| \nabla_{\mathbf{w}^t} F_i(\mathbf{x}_i^{t+1}, \mathbf{c}_i^{t+1}, \mathbf{w}^t) \right\|^2$$

$$= F_i(\mathbf{x}_i^{t+1}, \mathbf{c}_i^{t+1}, \mathbf{w}^t) - (\frac{\eta_3}{2} - \lambda_p(L_{D_2} + L_{DQ_3})\eta_3^2) \left\| \nabla_{\mathbf{w}^t} F_i(\mathbf{x}_i^{t+1}, \mathbf{c}_i^{t+1}, \mathbf{w}^t) \right\|^2$$

$$+ (\frac{\eta_3}{2} + \lambda_p(L_{D_2} + L_{DQ_3})\eta_3^2) \left\| \mathbf{g}^t - \nabla_{\mathbf{w}_i^t} F_i(\mathbf{x}_i^{t+1}, \mathbf{c}_i^{t+1}, \mathbf{w}_i^t) + \nabla_{\mathbf{w}_i^t} F_i(\mathbf{x}_i^{t+1}, \mathbf{c}_i^{t+1}, \mathbf{w}_i^t) \right.$$

$$\left. - \nabla_{\mathbf{w}^t} F_i(\mathbf{x}_i^{t+1}, \mathbf{c}_i^{t+1}, \mathbf{w}^t) \right\|^2$$

$$\leq F_i(\mathbf{x}_i^{t+1}, \mathbf{c}_i^{t+1}, \mathbf{w}^t) - (\frac{\eta_3}{2} - \lambda_p(L_{D_2} + L_{DQ_3})\eta_3^2) \left\| \nabla_{\mathbf{w}^t} F_i(\mathbf{x}_i^{t+1}, \mathbf{c}_i^{t+1}, \mathbf{w}^t) \right\|^2$$

$$+ (\eta_3 + 2\lambda_p(L_{D_2} + L_{DQ_3})\eta_3^2) \left\| \mathbf{g}^t - \nabla_{\mathbf{w}_i^t} F_i(\mathbf{x}_i^{t+1}, \mathbf{c}_i^{t+1}, \mathbf{w}_i^t) \right\|^2$$

$$+ (\eta_3 + 2\lambda_p(L_{D_2} + L_{DQ_3})\eta_3^2) \left\| \nabla_{\mathbf{w}_i^t} F_i(\mathbf{x}_i^{t+1}, \mathbf{c}_i^{t+1}, \mathbf{w}_i^t) - \nabla_{\mathbf{w}^t} F_i(\mathbf{x}_i^{t+1}, \mathbf{c}_i^{t+1}, \mathbf{w}^t) \right\|^2$$

$$\leq F_i(\mathbf{x}_i^{t+1}, \mathbf{c}_i^{t+1}, \mathbf{w}^t) - (\frac{\eta_3}{2} - \lambda_p(L_{D_2} + L_{DQ_3})\eta_3^2) \left\| \nabla_{\mathbf{w}^t} F_i(\mathbf{x}_i^{t+1}, \mathbf{c}_i^{t+1}, \mathbf{w}^t) \right\|^2$$

$$+ (\eta_3 + 2\lambda_p(L_{D_2} + L_{DQ_3})\eta_3^2) \left\| \mathbf{g}^t - \nabla_{\mathbf{w}_i^t} F_i(\mathbf{x}_i^{t+1}, \mathbf{c}_i^{t+1}, \mathbf{w}_i^t) \right\|^2$$

$$+ (\eta_3 + 2\lambda_p(L_{D_2} + L_{DQ_3})\eta_3^2)(\lambda_p(L_{D_2} + L_{DQ_3}))^2 \left\| \mathbf{w}_i^t - \mathbf{w}^t \right\|^2.$$

Rearranging the terms gives (25),

*Proof Lemma 1.* Let $t_c$ be the latest synchronization time before $t$. Define $\gamma_t = \frac{1}{n}\sum_{i=1}^n (L_{\max}^{(i)})^2 \|\mathbf{w}^t - \mathbf{w}_i^t\|^2$. Then:

$$\gamma_t = \frac{1}{n}\sum_{i=1}^n (L_{\max}^{(i)})^2 \left\| \mathbf{w}^{t_c} - \frac{\eta_3}{n}\sum_{j=t_c}^t \sum_{k=1}^n \nabla_{\mathbf{w}_k^j} F_k(\mathbf{x}_k^{j+1}, \mathbf{c}_k^{j+1}, \mathbf{w}_k^j) \right.$$

$$\left. - (\mathbf{w}^{t_c} - \eta_3 \sum_{j=t_c}^t \nabla_{\mathbf{w}_i^j} F_i(\mathbf{x}_i^{j+1}, \mathbf{c}_i^{j+1}, \mathbf{w}_i^j)) \right\|^2$$

$$\overset{(a)}{\leq} \tau \sum_{j=t_c}^t \frac{\eta_3^2}{n}\sum_{i=1}^n (L_{\max}^{(i)})^2 \left\| \frac{1}{n}\sum_{k=1}^n \nabla_{\mathbf{w}_k^j} F_k(\mathbf{x}_k^{j+1}, \mathbf{c}_k^{j+1}, \mathbf{w}_k^j) - \nabla_{\mathbf{w}_i^j} F_i(\mathbf{x}_i^{j+1}, \mathbf{c}_i^{j+1}, \mathbf{w}_i^j) \right\|^2 \quad (\star 1)$$

$$= \tau \sum_{j=t_c}^t \frac{\eta_3^2}{n}\sum_{i=1}^n (L_{\max}^{(i)})^2 \left[ \left\| \frac{1}{n}\sum_{k=1}^n \left( \nabla_{\mathbf{w}_k^j} F_k(\mathbf{x}_k^{j+1}, \mathbf{c}_k^{j+1}, \mathbf{w}_k^j) - \nabla_{\mathbf{w}^j} F_k(\mathbf{x}_k^{j+1}, \mathbf{c}_k^{j+1}, \mathbf{w}^j) \right. \right. \right.$$

$$\left. + \nabla_{\mathbf{w}^j} F_k(\mathbf{x}_k^{j+1}, \mathbf{c}_k^{j+1}, \mathbf{w}^j) \right) - \nabla_{\mathbf{w}^j} F_i(\mathbf{x}_i^{j+1}, \mathbf{c}_i^{j+1}, \mathbf{w}^j) + \nabla_{\mathbf{w}^j} F_i(\mathbf{x}_i^{j+1}, \mathbf{c}_i^{j+1}, \mathbf{w}^j)$$

$$\left. \left. - \nabla_{\mathbf{w}_i^j} F_i(\mathbf{x}_i^{j+1}, \mathbf{c}_i^{j+1}, \mathbf{w}_i^j) \right\|^2 \right]$$

$$\leq \tau \sum_{j=t_c}^{t_c+\tau} 3\frac{\eta_3^2}{n}\sum_{i=1}^n (L_{\max}^{(i)})^2 \left[ \left\| \frac{1}{n}\sum_{k=1}^n \left( \nabla_{\mathbf{w}_k^j} F_k(\mathbf{x}_k^{j+1}, \mathbf{c}_k^{j+1}, \mathbf{w}_k^j) - \nabla_{\mathbf{w}^j} F_k(\mathbf{x}_k^{j+1}, \mathbf{c}_k^{j+1}, \mathbf{w}^j) \right) \right\|^2 \right.$$

$$+ \left\| \frac{1}{n}\sum_{k=1}^n \nabla_{\mathbf{w}^j} F_k(\mathbf{x}_k^{j+1}, \mathbf{c}_k^{j+1}, \mathbf{w}^j) - \nabla_{\mathbf{w}^j} F_i(\mathbf{x}_i^{j+1}, \mathbf{c}_i^{j+1}, \mathbf{w}^j) \right\|^2$$

$$\left. + \left\| \nabla_{\mathbf{w}^j} F_i(\mathbf{x}_i^{j+1}, \mathbf{c}_i^{j+1}, \mathbf{w}^j) - \nabla_{\mathbf{w}_i^j} F_i(\mathbf{x}_i^{j+1}, \mathbf{c}_i^{j+1}, \mathbf{w}_i^j) \right\|^2 \right]$$

$$\leq \tau \sum_{j=t_c}^{t_c+\tau} 3\eta_3^2 \left[ \frac{(\lambda_p L_w)^2 n}{n^2}\sum_{k=1}^n \frac{1}{n}\sum_{i=1}^n (L_{\max}^{(i)})^2 \|\mathbf{w}_k^j - \mathbf{w}^j\|^2 + \frac{1}{n}\sum_{i=1}^n (L_{\max}^{(i)})^2 \kappa_i \right.$$

$$\left. + (\lambda_p L_w)^2 \frac{1}{n}\sum_{i=1}^n (L_{\max}^{(i)})^2 \|\mathbf{w}^j - \mathbf{w}_i^j\|^2 \right]$$

$$\leq \tau \sum_{j=t_c}^{t_c+\tau} 3\eta_3^2 \Big[ \frac{(\lambda_p L_w)^2 n}{n^2} \frac{1}{n} \sum_{i=1}^{n} (L_{\max}^{(i)})^2 \frac{1}{(L_{\max}^{(\min)})^2} \sum_{k=1}^{n} (L_{\max}^{(k)})^2 \|\mathbf{w}_k^j - \mathbf{w}^j\|^2$$

$$+ \frac{1}{n} \sum_{i=1}^{n} (L_{\max}^{(i)})^2 \kappa_i + (\lambda_p L_w)^2 \gamma_j \|^2 \Big]$$

$$= \tau \sum_{j=t_c}^{t_c+\tau} 3\eta_3^2 \Big( (\lambda_p L_w)^2 \frac{1}{n} \sum_{i=1}^{n} (L_{\max}^{(i)})^2 \frac{1}{(L_{\max}^{(\min)})^2} \gamma_j + \frac{1}{n} \sum_{i=1}^{n} (L_{\max}^{(i)})^2 \kappa_i + (\lambda_p L_w)^2 \gamma_j \Big)$$

$$= \tau \sum_{j=t_c}^{t_c+\tau} 3\eta_3^2 \Big( (\lambda_p L_w)^2 \frac{\overline{L}_{\max}^2}{(L_{\max}^{(\min)})^2} \gamma_j + \frac{1}{n} \sum_{i=1}^{n} (L_{\max}^{(i)})^2 \kappa_i + (\lambda_p L_w)^2 \gamma_j \Big) \qquad (\star 2)$$

in (a) we use the facts that $\|\sum_{i=1}^{K} a_i\|^2 \leq K \sum_{i=1}^{K} \|a_i\|^2$, $t \leq \tau + t_c$ and that we are summing over non-negative terms. As a result, we have:

$$\gamma_t \leq \tau \sum_{j=t_c}^{t_c+\tau} 3\eta_3^2 \Big( (\lambda_p L_w)^2 \Big(1 + \frac{\overline{L}_{\max}^2}{(L_{\max}^{(\min)})^2}\Big) \gamma_j + \frac{1}{n} \sum_{i=1}^{n} (L_{\max}^{(i)})^2 \kappa_i \Big)$$

$$\implies \sum_{t=t_c}^{t_c+\tau} \gamma_t \leq \sum_{t=t_c}^{t_c+\tau} \sum_{j=t_c}^{t_c+\tau} 3\tau \eta_3^2 \Big( (\lambda_p L_w)^2 \Big(1 + \frac{\overline{L}_{\max}^2}{(L_{\max}^{(\min)})^2}\Big) \gamma_j + \frac{1}{n} \sum_{i=1}^{n} (L_{\max}^{(i)})^2 \kappa_i \Big)$$

$$= 3\tau^2 \eta_3^2 (\lambda_p L_w)^2 \Big(1 + \frac{\overline{L}_{\max}^2}{(L_{\max}^{(\min)})^2}\Big) \sum_{j=t_c}^{t_c+\tau} \gamma_j + 3\tau^3 \eta_3^2 \frac{1}{n} \sum_{i=1}^{n} (L_{\max}^{(i)})^2 \kappa_i$$

Let us choose $\eta_3$ such that $3\tau^2 \eta_3^2 (\lambda_p L_w)^2 \Big(1 + \frac{\overline{L}_{\max}^2}{(L_{\max}^{(\min)})^2}\Big) \leq \frac{1}{2} \Leftrightarrow \eta_3 \leq \sqrt{\frac{1}{6\tau^2(\lambda_p L_w)^2 \Big(1 + \frac{\overline{L}_{\max}^2}{(L_{\max}^{(\min)})^2}\Big)}}$, sum over all syncronization times, and divide both sides by $T$:

$$\frac{1}{T} \sum_{t=0}^{T-1} \gamma_t \leq \frac{1}{2} \sum_{j=0}^{T-1} \gamma_j + 3\tau^2 \eta_3^2 \frac{1}{n} \sum_{i=1}^{n} (L_{\max}^{(i)})^2 \kappa_i$$

$$\implies \frac{1}{T} \sum_{t=0}^{T-1} \gamma_t \leq 6\tau^2 \eta_3^2 \frac{1}{n} \sum_{i=1}^{n} (L_{\max}^{(i)})^2 \kappa_i$$

$\square$

*Proof of Corollary 1.* From $(\star 1) \leq (\star 2)$ in the proof of Lemma 1, we have:

$$\sum_{t=t_c}^{t_c+\tau} \frac{1}{n} \sum_{i=1}^{n} (L_{\max}^{(i)})^2 \Big\| \mathbf{g}^t - \nabla_{\mathbf{w}_i^t} F_i(\mathbf{x}_i^{t+1}, \mathbf{c}_i^{t+1}, \mathbf{w}_i^t) \Big\|^2$$

$$\leq \sum_{t=t_c}^{t_c+\tau} 3 \Big( (\lambda_p L_w)^2 \Big(1 + \frac{\overline{L}_{\max}^2}{(L_{\max}^{(\min)})^2}\Big) \gamma_t + \frac{1}{n} \sum_{i=1}^{n} (L_{\max}^{(i)})^2 \kappa_i \Big)$$

Summing over all $t_c$ and dividing by $T$:

$$\frac{1}{T} \sum_{t=0}^{T-1} \frac{1}{n} \sum_{i=1}^{n} (L_{\max}^{(i)})^2 \Big\| \mathbf{g}^t - \nabla_{\mathbf{w}_i^t} F_i(\mathbf{x}_i^{t+1}, \mathbf{c}_i^{t+1}, \mathbf{w}_i^t) \Big\|^2$$

$$\leq 3(\lambda_p L_w)^2 \Big(1 + \frac{\overline{L}_{\max}^2}{(L_{\max}^{(\min)})^2}\Big) \frac{1}{T} \sum_{t=0}^{T-1} \gamma_t + 3 \frac{1}{n} \sum_{i=1}^{n} (L_{\max}^{(i)})^2 \kappa_i$$

$$\overset{(a)}{\leq} 3(\lambda_p L_w)^2 \Big(1 + \frac{\overline{L}_{\max}^2}{(L_{\max}^{(\min)})^2}\Big) \times 6\tau^2 \eta_3^2 \frac{1}{n} \sum_{i=1}^{n} (L_{\max}^{(i)})^2 \kappa_i + 3 \frac{1}{n} \sum_{i=1}^{n} (L_{\max}^{(i)})^2 \kappa_i,$$

where (a) is from Lemma 1. $\square$

## C.4 Proof Outline with Client Sampling

Incorporating partial client participation and analyzing the resulting algorithm is fairly simple. Essentially only changes are in Lemma 1 and Corollary 1, as everything before that is for local updates only. Now we give a summary of what changes:

Let $\mathcal{K}_t$ denote the set of clients that participates at time $t$, where $|\mathcal{K}_t| = K$, i.e., $K$ clients participate in the training process at any time. In this case, we define the average parameter $\mathbf{w}^t$ and the gradient $\mathbf{g}^t$ as the average over the respective parameters of only the active clients at time $t$; we also define $\gamma_t$ similarly.

- **Change in the proof of Lemma 1:** In the proof of Lemma 1, the second term on the RHS of the second inequality, with the above modification will be equal to $\left\| \frac{1}{K} \sum_{k \in \mathcal{K}_t} \nabla_{\mathbf{w}^j} F_k(\mathbf{x}_k^{j+1}, \mathbf{c}_k^{j+1}, \mathbf{w}^j) - \nabla_{\mathbf{w}^j} F_i(\mathbf{x}_i^{j+1}, \mathbf{c}_i^{j+1}, \mathbf{w}^j) \right\|^2$. Earlier, the average was over all clients from 1 to $n$ and this term was bounded by $\kappa_i$ using Assumption A.6. Now, we can use the Jensen's inequality (iteratively) and Assumption A.6 and bound this by $2\kappa_i + \frac{2}{K} \sum_{j \in \mathcal{K}_t} \kappa_j$. This change will propagate over until the end.
- **Change in the proof of Corollary 1:** Since this is a corollary to Lemma 1, this will also see a similar change.
- **Remaining convergence proof:** Now, continuing the exact same convergence proof and using the modified bounds of Lemma 1 and Corollary 1 will give the bound of our algorithm with partial client participation.

This is the modification in the entire proof.

# D  Problem Setup and Convergence Result of QuPeL

For QuPeL, we define the following augmented loss function at client $i$:

$$F_i(\mathbf{x}_i, \mathbf{c}_i, \mathbf{w}) := f_i(\mathbf{x}_i) + f_i(\widetilde{Q}_{\mathbf{c}_i}(\mathbf{x}_i)) + \lambda R(\mathbf{x}_i, \mathbf{c}_i) + \frac{\lambda_p}{2}(\|\mathbf{x}_i - \mathbf{w}\|^2 + \|\widetilde{Q}_{\mathbf{c}_i}(\mathbf{x}_i) - \mathbf{w}\|^2). \tag{40}$$

Here, $\mathbf{w} \in \mathbb{R}^d$ denotes the global model, $\mathbf{x}_i \in \mathbb{R}^d$ denotes the personalized model at client $i$, and $\mathbf{c}_i \in \mathbb{R}^{m_i}$ denotes the model quantization centers at client $i$, where $m_i$ is the number of centers at client $i$, with $\log m_i$ representing the number of bits per parameter, which could be different for each client – larger the $m_i$, higher the precision. Note that different than QuPeD, this time all the model parameters $\mathbf{x}_i$ are of the same dimension with $\mathbf{w}$.

From (40) we get the following global loss function:

$$\arg\min \left( F(\{\mathbf{x}_i\}, \{\mathbf{c}_i\}, \mathbf{w}) := \frac{1}{n} \sum_{i=1}^n F_i(\mathbf{x}_i, \mathbf{c}_i, \mathbf{w}) \right), \tag{41}$$

where minimization is taken over $\mathbf{x}_1, \ldots, \mathbf{x}_n \in \mathbb{R}^d$, $\mathbf{c}_i \in \mathbb{R}^{m_i}$ for $i \in [n]$, and $\mathbf{w} \in \mathbb{R}^d$.

We present our proposed algorithm QuPeL for minimizing (41) in Algorithm 3.

## D.1 Convergence Analysis

First we state our convergence result for QuPeL,

**Theorem 3.** *Consider running Algorithm 3 for $T$ iterations with $\tau \leq \sqrt{T}$, $\eta_1 = \frac{1}{2(3\lambda_p + \lambda_p L_{Q_1} + L + GL_{Q_1} + G_{Q_1} Ll_{Q_1})}$, $\eta_2 = \frac{1}{2(\lambda_p + \lambda_p L_{Q_2} + GL_{Q_2} + G_{Q_2} LL_{Q_2})}$, and $\eta_3 = \frac{1}{8\lambda_p \sqrt{T}}$. For any $t \in [T]$, define $\mathbf{G}_i^t := [\nabla_{\mathbf{x}_i^{t+1}} F_i(\mathbf{x}_i^{t+1}, \mathbf{c}_i^t, \mathbf{w}^t)^T, \nabla_{\mathbf{c}_i^{t+1}} F_i(\mathbf{x}_i^{t+1}, \mathbf{c}_i^{t+1}, \mathbf{w}^t)^T, \nabla_{\mathbf{w}^t} F_i(\mathbf{x}_i^{t+1}, \mathbf{c}_i^{t+1}, \mathbf{w}^t)^T]^T$. Then, under assumptions **A.1-A.6**, we have:*

$$\frac{1}{T} \sum_{t=0}^{T-1} \frac{1}{n} \sum_{i=1}^n \left\| \mathbf{G}_i^t \right\|^2 = \mathcal{O}\left( \frac{\Lambda}{\sqrt{T}} + \frac{L_{\max}^2 \tau^2 \kappa + \tau^2 \kappa^2}{T} + \frac{L_{\max}^2 \tau^2 \kappa}{T^{\frac{3}{2}}} + L_{\max}^2 \kappa \right),$$

---

**Algorithm 3** QuPeL: Quantized Personalization

---

**Input:** Regularization parameters $\lambda, \lambda_p$; synchronization gap $\tau$; for each client $i \in [n]$, initialize full precision personalized model $\mathbf{x}_i^0$, quantization centers $\mathbf{c}_i^0$, and local model $\mathbf{w}_i^0$; a penalty function enforcing quantization $R(\mathbf{x}, \mathbf{c})$; a soft quantizer $\widetilde{Q}_{\mathbf{c}}(\mathbf{x})$; and learning rates $\eta_1, \eta_2, \eta_3$.

1: **for** $t = 0$ **to** $T - 1$ **do**
2:    **On Clients** $i = 1$ **to** $n$ (in parallel) **do**:
3:    **if** $\tau$ does not divide $t$ **then**
4:       Compute $\mathbf{g}_i^t := \nabla_{\mathbf{x}_i^t} f_i(\mathbf{x}_i^t) + \nabla_{\mathbf{x}_i^t} f_i(\widetilde{Q}_{\mathbf{c}_i^t}(\mathbf{x}_i^t)) + \lambda_p(\mathbf{x}_i^t - \mathbf{w}_i^t) + \lambda_p \nabla_{\mathbf{x}_i^t} \widetilde{Q}_{\mathbf{c}_i^t}(\mathbf{x}_i^t)(\widetilde{Q}_{\mathbf{c}_i^t}(\mathbf{x}_i^t) - \mathbf{w}_i^t)$
5:       $\mathbf{x}_i^{t+1} = \text{prox}_{\eta_1 \lambda R_{\mathbf{c}_i^t}}(\mathbf{x}_i^t - \eta_1 \mathbf{g}_i^t)$
6:       Compute $\mathbf{h}_i^t := \nabla_{\mathbf{c}_i^t} f_i(\widetilde{Q}_{\mathbf{c}_i^t}(\mathbf{x}_i^{t+1})) + \lambda_p \nabla_{\mathbf{c}_i^{t+1}} \widetilde{Q}_{\mathbf{c}_i^t}(\mathbf{x}_i^{t+1})(\widetilde{Q}_{\mathbf{c}_i^t}(\mathbf{x}_i^{t+1}) - \mathbf{w}_i^t)$
7:       $\mathbf{c}_i^{t+1} = \text{prox}_{\eta_2 \lambda R_{\mathbf{x}_i^{t+1}}}(\mathbf{c}_i^t - \eta_2 \mathbf{h}_i^t)$
8:       $\mathbf{w}_i^{t+1} = \mathbf{w}_i^t - \eta_3 \lambda_p((\mathbf{w}_i^t - \mathbf{x}_i^{t+1}) + (\mathbf{w}_i^t - \widetilde{Q}_{\mathbf{c}_i^{t+1}}(\mathbf{x}_i^{t+1}))$
9:    **else**
10:      Send $\mathbf{w}_i^t$ to **Server**
11:      Receive $\mathbf{w}^t$ from **Server** and set $\mathbf{w}_i^{t+1} = \mathbf{w}^t$
12:    **end if**
13:    **On Server do**:
14:    **if** $\tau$ divides $t$ **then**
15:      Receive $\{\mathbf{w}_i^t\}_{i=1}^n$ and compute $\mathbf{w}^t := \frac{1}{n} \sum_{i=1}^n \mathbf{w}_i^t$
16:      Broadcast $\mathbf{w}^t$ to all **Clients**
17:    **end if**
18: **end for**
19: $\hat{\mathbf{x}}_i^T = Q_{\mathbf{c}_i^T}(\mathbf{x}_i^T)$ for all $i \in [n]$

**Output:** Quantized personalized models $\hat{\mathbf{x}}_i^T$ for $i \in [n]$

---

*where* $L_{\max} = \max\{\sqrt{\frac{1}{18}}, \frac{2\lambda_p}{3} + \frac{2\lambda_p L_{Q_2}}{3} + GL_{Q_2} + G_{Q_2}LL_{Q_2}, \frac{7}{3}\lambda_p + \frac{2\lambda_p L_{Q_1}}{3} + \frac{\lambda_p G_{Q_1} l_{Q_1}}{3} + L + Gl_{Q_1} + G_{Q_1}Ll_{Q_1}\}$, *and* $\Lambda = L_{\max}^2 \tau \kappa(1 + G_{Q_1}^2 + G_{Q_2}^2) + L_{\max}^2 + L_{\max}^2 \lambda_p \Delta_F$ *with* $\Delta_F = \frac{1}{n} \sum_{i=1}^n \left(F_i(\mathbf{x}_i^0, \mathbf{c}_i^0, \mathbf{w}_i^0) - F_i(\mathbf{x}_i^T, \mathbf{c}_i^T, \mathbf{w}_i^T)\right)$.

The rest of this section is devoted to proving Theorem 3.

The analysis is similar to the one in Appendix C, however, this time we can assume same gradient bounds for each client since all the clients have the same model structure. As a result the analysis requires slightly less algebraic manipulations. Naturally, we don't need **A.7** in this analysis. We defer proofs of the claims and some derivation details to the end of section. We take $\mathbf{w}^t = \frac{1}{n} \sum_{i=1}^n \mathbf{w}_i^t$, so that $\mathbf{w}^t$ is defined for every time point.

**Alternating updates.** Let us first restate the alternating updates for $\mathbf{x}_i$ and $\mathbf{c}_i$:

$$\mathbf{x}_i^{t+1} = \text{prox}_{\eta_1 \lambda R_{\mathbf{c}_i^t}}(\mathbf{x}_i^t - \eta_1 \nabla f_i(\mathbf{x}_i^t) - \eta_1 \nabla_{\mathbf{x}_i^t} f_i(\widetilde{Q}_{\mathbf{c}_i^t}(\mathbf{x}_i^t)) - \eta_1 \lambda_p(\mathbf{x}_i^t - \mathbf{w}_i^t)$$
$$+ \nabla_{\mathbf{x}_i^t} \widetilde{Q}_{\mathbf{c}_i^t}(\mathbf{x}_i^t)(\widetilde{Q}_{\mathbf{c}_i^t}(\mathbf{x}_i^t) - \mathbf{w}_i^t))$$
$$\mathbf{c}_i^{t+1} = \text{prox}_{\eta_2 \lambda R_{\mathbf{x}_i^{t+1}}}(\mathbf{c}_i^t - \eta_2 \nabla_{\mathbf{c}_i^t} f_i(\widetilde{Q}_{\mathbf{c}_i^t}(\mathbf{x}_i^t)) - \eta_2 \lambda_p \nabla_{\mathbf{c}_i^{t+1}} \widetilde{Q}_{\mathbf{c}_i^t}(\mathbf{x}_i^{t+1})(\widetilde{Q}_{\mathbf{c}_i^t}(\mathbf{x}_i^{t+1}) - \mathbf{w}_i^t))$$

The alternating updates are equivalent to solving the following two optimization problems.

$$\mathbf{x}_i^{t+1} = \underset{\mathbf{x} \in \mathbb{R}^d}{\arg\min} \left\{ \left\langle \mathbf{x} - \mathbf{x}_i^t, \nabla f_i(\mathbf{x}_i^t) \right\rangle + \left\langle \mathbf{x} - \mathbf{x}_i^t, \nabla_{\mathbf{x}_i^t} f_i(\widetilde{Q}_{\mathbf{c}_i^t}(\mathbf{x}_i^t)) \right\rangle + \left\langle \mathbf{x} - \mathbf{x}_i^t, \lambda_p(\mathbf{x}_i^t - \mathbf{w}_i^t) \right\rangle \right.$$
$$\left. + \left\langle \mathbf{x} - \mathbf{x}_i^t, \lambda_p \nabla_{\mathbf{x}_i^t} \widetilde{Q}_{\mathbf{c}_i^t}(\mathbf{x}_i^t)(\widetilde{Q}_{\mathbf{c}_i^t}(\mathbf{x}_i^t) - \mathbf{w}_i^t) \right\rangle + \frac{1}{2\eta_1} \|\mathbf{x} - \mathbf{x}_i^t\|_2^2 + \lambda R(\mathbf{x}, \mathbf{c}_i^t) \right\}$$
$$(42)$$

$$\mathbf{c}_i^{t+1} = \underset{\mathbf{c} \in \mathbb{R}^m}{\arg\min} \left\{ \left\langle \mathbf{c} - \mathbf{c}_i^t, \nabla_{\mathbf{c}_i^t} f_i(\widetilde{Q}_{\mathbf{c}_i^t}(\mathbf{x}_i^{t+1})) \right\rangle + \left\langle \mathbf{c} - \mathbf{c}_i^t, \lambda_p \nabla_{\mathbf{c}_i^{t+1}} \widetilde{Q}_{\mathbf{c}_i^t}(\mathbf{x}_i^{t+1})(\widetilde{Q}_{\mathbf{c}_i^t}(\mathbf{x}_i^{t+1}) - \mathbf{w}_i^t) \right\rangle \right.$$

$$+\frac{1}{2\eta_2}\left\|\mathbf{c}-\mathbf{c}_i^t\right\|_2^2+\lambda R(\mathbf{x}_i^{t+1},\mathbf{c})\bigg\} \tag{43}$$

Note that the update on $\mathbf{w}^t$ from Algorithm 3 can be written as:

$$\mathbf{w}^{t+1}=\mathbf{w}^t-\eta_3\mathbf{g}^t, \quad \text{where} \quad \mathbf{g}^t=\frac{1}{n}\sum_{i=1}^n\nabla_{\mathbf{w}_i^t}F_i(\mathbf{x}_i^{t+1},\mathbf{c}_i^{t+1},\mathbf{w}_i^t).$$

In the convergence analysis we will require smoothness of the local functions $F_i$ w.r.t. the global parameter $\mathbf{w}$. Recall the definition of $F_i(\mathbf{x}_i,\mathbf{c}_i,\mathbf{w})=f_i(\mathbf{x}_i)+f_i(\widetilde{Q}_{\mathbf{c}_i}(\mathbf{x}_i))+\lambda R(\mathbf{x}_i,\mathbf{c}_i)+\frac{\lambda_p}{2}(\|\mathbf{x}_i-\mathbf{w}\|^2+\|\widetilde{Q}_{\mathbf{c}_i}(\mathbf{x}_i)-\mathbf{w}\|^2)$ from (40). It follows that $F_i$ is $2\lambda_p$-smooth with respect to $\mathbf{w}$:

$$\left\|\nabla_\mathbf{w}F_i(\mathbf{x},\mathbf{c},\mathbf{w})-\nabla_{\mathbf{w}'}F_i(\mathbf{x},\mathbf{c},\mathbf{w}')\right\|=\left\|\lambda_p(2\mathbf{w}-\mathbf{x}-\widetilde{Q}_\mathbf{c}(\mathbf{x}))-\lambda_p(2\mathbf{w}'-\mathbf{x}-\widetilde{Q}_\mathbf{c}(\mathbf{x}))\right\|$$

$$\leq 2\lambda_p\|\mathbf{w}-\mathbf{w}'\|, \quad \forall \mathbf{w},\mathbf{w}'\in\mathbb{R}^d. \tag{44}$$

Now let us move on with the proof.

### D.2 Sufficient Decrease

We will divide this part into three and obtain sufficient decrease properties for each variable: $\mathbf{x}_i,\mathbf{c}_i,\mathbf{w}$.

#### D.2.1 Sufficient Decrease Due to $\mathbf{x}_i$

We begin with a useful claim.

**Claim 12.** $f_i(\mathbf{x})+f_i(\widetilde{Q}_\mathbf{c}(\mathbf{x}))+\frac{\lambda_p}{2}(\|\mathbf{x}-\mathbf{w}\|^2+\|\widetilde{Q}_\mathbf{c}(\mathbf{x})-\mathbf{w}\|^2)$ is $(\lambda_p+\lambda_pL_{Q_1}+L+GL_{Q_1}+G_{Q_1}Ll_{Q_1})$-smooth with respect to $\mathbf{x}$.

From Claim 12 we have:

$$F_i(\mathbf{x}_i^{t+1},\mathbf{c}_i^t,\mathbf{w}^t)+(\frac{1}{2\eta_1}-\frac{\lambda_p+\lambda_pL_{Q_1}+L+GL_{Q_1}+G_{Q_1}Ll_{Q_1}}{2})\|\mathbf{x}_i^{t+1}-\mathbf{x}_i^t\|^2$$

$$=f_i(\mathbf{x}_i^{t+1})+f_i(\widetilde{Q}_{\mathbf{c}_i^t}(\mathbf{x}_i^{t+1}))+\lambda R(\mathbf{x}_i^{t+1},\mathbf{c}_i^t)+\frac{\lambda_p}{2}\|\mathbf{x}_i^{t+1}-\mathbf{w}^t\|^2$$

$$+\frac{\lambda_p}{2}\|\widetilde{Q}_{\mathbf{c}_i^t}(\mathbf{x}_i^{t+1})-\mathbf{w}^t\|^2+(\frac{1}{2\eta_1}-\frac{\lambda_p+\lambda_pL_{Q_1}+L+GL_{Q_1}+G_{Q_1}Ll_{Q_1}}{2})\|\mathbf{x}_i^{t+1}-\mathbf{x}_i^t\|^2$$

$$\leq f_i(\mathbf{x}_i^t)+f_i(\widetilde{Q}_{\mathbf{c}_i^t}(\mathbf{x}_i^t))+\frac{\lambda_p}{2}\|\mathbf{x}_i^t-\mathbf{w}^t\|^2+\frac{\lambda_p}{2}\|\widetilde{Q}_{\mathbf{c}_i^t}(\mathbf{x}_i^t)-\mathbf{w}^t\|^2+\lambda R(\mathbf{x}_i^{t+1},\mathbf{c}_i^t)$$

$$+\left\langle\nabla f_i(\mathbf{x}_i^t),\mathbf{x}_i^{t+1}-\mathbf{x}_i^t\right\rangle+\left\langle\nabla_{\mathbf{x}_i^t}f_i(\widetilde{Q}_{\mathbf{c}_i^t}(\mathbf{x}_i^t)),\mathbf{x}_i^{t+1}-\mathbf{x}_i^t\right\rangle+\left\langle\lambda_p(\mathbf{x}_i^t-\mathbf{w}_i^t),\mathbf{x}_i^{t+1}-\mathbf{x}_i^t\right\rangle$$

$$+\left\langle\lambda_p\nabla_{\mathbf{x}_i^t}\widetilde{Q}_{\mathbf{c}_i^t}(\mathbf{x}_i^t)(\widetilde{Q}_{\mathbf{c}_i^t}(\mathbf{x}_i^t)-\mathbf{w}_i^t),\mathbf{x}_i^{t+1}-\mathbf{x}_i^t\right\rangle+\left\langle\lambda_p(\mathbf{w}_i^t-\mathbf{w}^t),\mathbf{x}_i^{t+1}-\mathbf{x}_i^t\right\rangle$$

$$+\left\langle\lambda_p\nabla_{\mathbf{x}_i^t}\widetilde{Q}_{\mathbf{c}_i^t}(\mathbf{x}_i^t)(\mathbf{w}_i^t-\mathbf{w}^t),\mathbf{x}_i^{t+1}-\mathbf{x}_i^t\right\rangle+\frac{1}{2\eta_1}\|\mathbf{x}_i^{t+1}-\mathbf{x}_i^t\|^2 \tag{45}$$

**Claim 13.** Let

$$A(\mathbf{x}_i^{t+1}):=\lambda R(\mathbf{x}_i^{t+1},\mathbf{c}_i^t)+\left\langle\nabla f_i(\mathbf{x}_i^t),\mathbf{x}_i^{t+1}-\mathbf{x}_i^t\right\rangle+\left\langle\nabla_{\mathbf{x}_i^t}f_i(\widetilde{Q}_{\mathbf{c}_i^t}(\mathbf{x}_i^t)),\mathbf{x}_i^{t+1}-\mathbf{x}_i^t\right\rangle$$

$$+\left\langle\lambda_p\nabla_{\mathbf{x}_i^t}\widetilde{Q}_{\mathbf{c}_i^t}(\mathbf{x}_i^t)(\widetilde{Q}_{\mathbf{c}_i^t}(\mathbf{x}_i^t)-\mathbf{w}_i^t),\mathbf{x}_i^{t+1}-\mathbf{x}_i^t\right\rangle+\left\langle\lambda_p(\mathbf{x}_i^t-\mathbf{w}_i^t),\mathbf{x}_i^{t+1}-\mathbf{x}_i^t\right\rangle$$

$$+\frac{1}{2\eta_1}\|\mathbf{x}_i^{t+1}-\mathbf{x}_i^t\|^2$$

$$A(\mathbf{x}_i^t):=\lambda R(\mathbf{x}_i^t,\mathbf{c}_i^t).$$

Then $A(\mathbf{x}_i^{t+1})\leq A(\mathbf{x}_i^t)$.

Using the inequality from Claim 13 in (45) gives

$$F_i(\mathbf{x}_i^{t+1}, \mathbf{c}_i^t, \mathbf{w}^t) + (\frac{1}{2\eta_1} - \frac{\lambda_p + \lambda_p L_{Q_1} + L + G L_{Q_1} + G_{Q_1} L l_{Q_1}}{2})\|\mathbf{x}_i^{t+1} - \mathbf{x}_i^t\|^2$$

$$\overset{(a)}{\leq} f_i(\mathbf{x}_i^t) + f_i(\widetilde{Q}_{\mathbf{c}_i^t}(\mathbf{x}_i^t)) + \frac{\lambda_p}{2}\|\mathbf{x}_i^t - \mathbf{w}^t\|^2 + \frac{\lambda_p}{2}\|\widetilde{Q}_{\mathbf{c}_i^t}(\mathbf{x}_i^t) - \mathbf{w}^t\|^2 + A(\mathbf{x}_i^{t+1})$$

$$+ \left\langle \lambda_p \nabla_{\mathbf{x}_i^t} \widetilde{Q}_{\mathbf{c}_i^t}(\mathbf{x}_i^t)(\mathbf{w}_i^t - \mathbf{w}^t), \mathbf{x}_i^{t+1} - \mathbf{x}_i^t \right\rangle + \left\langle \lambda_p(\mathbf{w}_i^t - \mathbf{w}^t), \mathbf{x}_i^{t+1} - \mathbf{x}_i^t \right\rangle$$

$$\overset{(b)}{\leq} f_i(\mathbf{x}_i^t) + f_i(\widetilde{Q}_{\mathbf{c}_i^t}(\mathbf{x}_i^t)) + \frac{\lambda_p}{2}\|\mathbf{x}_i^t - \mathbf{w}^t\|^2 + \frac{\lambda_p}{2}\|\widetilde{Q}_{\mathbf{c}_i^t}(\mathbf{x}_i^t) - \mathbf{w}^t\|^2 + \lambda R(\mathbf{x}_i^t, \mathbf{c}_i^t)$$

$$+ \frac{\lambda_p(1 + G_{Q_1}^2)}{2}\|\mathbf{w}_i^t - \mathbf{w}^t\|^2 + \lambda_p \|\mathbf{x}_i^{t+1} - \mathbf{x}_i^t\|^2$$

$$= F_i(\mathbf{x}_i^t, \mathbf{c}_i^t, \mathbf{w}^t) + \frac{\lambda_p(1 + G_{Q_1}^2)}{2}\|\mathbf{w}_i^t - \mathbf{w}^t\|^2 + \lambda_p \|\mathbf{x}_i^{t+1} - \mathbf{x}_i^t\|^2 \qquad (46)$$

To obtain (a), we substituted the value of $A(\mathbf{x}_i^{t+1})$ from Claim 13 into (45). In (b), we used $A(\mathbf{x}_i^{t+1}) \leq \lambda R(\mathbf{x}_i^t, \mathbf{c}_i^t)$ and $\langle \lambda_p(\mathbf{w}_i^t - \mathbf{w}^t), \mathbf{x}_i^{t+1} - \mathbf{x}_i^t \rangle = \langle \sqrt{\lambda_p}(\mathbf{w}_i^t - \mathbf{w}^t), \sqrt{\lambda_p}(\mathbf{x}_i^{t+1} - \mathbf{x}_i^t) \rangle \leq \frac{\lambda_p}{2}\|\mathbf{w}_i^t - \mathbf{w}^t\|^2 + \frac{\lambda_p}{2}\|\mathbf{x}_i^{t+1} - \mathbf{x}_i^t\|^2$, and

$$\left\langle \lambda_p \nabla_{\mathbf{x}_i^t} \widetilde{Q}_{\mathbf{c}_i^t}(\mathbf{x}_i^t)(\mathbf{w}_i^t - \mathbf{w}^t), \mathbf{x}_i^{t+1} - \mathbf{x}_i^t \right\rangle \leq \frac{\lambda_p}{2}\left\| \widetilde{Q}_{\mathbf{c}_i^t}(\mathbf{x}_i^t)(\mathbf{w}_i^t - \mathbf{w}^t) \right\|_2^2 + \frac{\lambda_p}{2}\|\mathbf{x}_i^{t+1} - \mathbf{x}_i^t\|^2$$

$$\leq \frac{\lambda_p}{2}\left\| \widetilde{Q}_{\mathbf{c}_i^t}(\mathbf{x}_i^t) \right\|_F^2 \|\mathbf{w}_i^t - \mathbf{w}^t\|_2^2 + \frac{\lambda_p}{2}\|\mathbf{x}_i^{t+1} - \mathbf{x}_i^t\|^2$$

$$\leq \frac{\lambda_p G_{Q_1}^2}{2}\|\mathbf{w}_i^t - \mathbf{w}^t\|^2 + \frac{\lambda_p}{2}\|\mathbf{x}_i^{t+1} - \mathbf{x}_i^t\|^2,$$

where the second inequality follows from $\|A\mathbf{x}\|_2 \leq \|A\|_F \|\mathbf{x}\|_2$, which holds for any matrix $A$ and vector $\mathbf{x}$ (dimension compatible); and the third inequality uses Assumption **A.5**.

Substituting $\eta_1 = \frac{1}{2(3\lambda_p + \lambda_p L_{Q_1} + L + G L_{Q_1} + G_{Q_1} L l_{Q_1})}$ in (46) gives:

$$F_i(\mathbf{x}_i^{t+1}, \mathbf{c}_i^t, \mathbf{w}^t) + (\frac{3\lambda_p + \lambda_p L_{Q_1} + L + G L_{Q_1} + G_{Q_1} L l_{Q_1}}{2})\|\mathbf{x}_i^{t+1} - \mathbf{x}_i^t\|^2 \leq F_i(\mathbf{x}_i^t, \mathbf{c}_i^t, \mathbf{w}^t)$$

$$+ \frac{\lambda_p(1 + G_{Q_1}^2)}{2}\|\mathbf{w}_i^t - \mathbf{w}^t\|^2$$
$$(47)$$

### D.2.2 Sufficient Decrease Due to $\mathbf{c}_i$

In parallel with Claim 12, we have following smoothness result for $\mathbf{c}$:

**Claim 14.** $f_i(\widetilde{Q}_{\mathbf{c}}(\mathbf{x})) + \frac{\lambda_p}{2}(\|\widetilde{Q}_{\mathbf{c}}(\mathbf{x}) - \mathbf{w}\|^2)$ is $(\lambda_p L_{Q_2} + G L_{Q_2} + G_{Q_2} L l_{Q_2})$-smooth with respect to $\mathbf{c}$.

From Claim 14 we have:

$$F_i(\mathbf{x}_i^{t+1}, \mathbf{c}_i^{t+1}, \mathbf{w}^t) + (\frac{1}{2\eta_2} - \frac{\lambda_p L_{Q_2} + G L_{Q_2} + G_{Q_2} L l_{Q_2}}{2})\|\mathbf{c}_i^{t+1} - \mathbf{c}_i^t\|^2$$

$$= f_i(\mathbf{x}_i^{t+1}) + f_i(\widetilde{Q}_{\mathbf{c}_i^{t+1}}(\mathbf{x}_i^{t+1})) + \lambda R(\mathbf{x}_i^{t+1}, \mathbf{c}_i^{t+1}) + \frac{\lambda_p}{2}\|\widetilde{Q}_{\mathbf{c}_i^t}(\mathbf{x}_i^{t+1}) - \mathbf{w}^t\|^2$$

$$+ \frac{\lambda_p}{2}\|\mathbf{x}_i^{t+1} - \mathbf{w}^t\|^2 + (\frac{1}{2\eta_2} - \frac{\lambda_p L_{Q_2} + G L_{Q_2} + G_{Q_2} L l_{Q_2}}{2})\|\mathbf{c}_i^{t+1} - \mathbf{c}_i^t\|^2$$

$$\leq f_i(\mathbf{x}_i^{t+1}) + f_i(\widetilde{Q}_{\mathbf{c}_i^t}(\mathbf{x}_i^{t+1})) + \lambda R(\mathbf{x}_i^{t+1}, \mathbf{c}_i^{t+1}) + \frac{\lambda_p}{2}\|\widetilde{Q}_{\mathbf{c}_i^t}(\mathbf{x}_i^{t+1}) - \mathbf{w}^t\|^2$$

$$+ \frac{\lambda_p}{2}\|\mathbf{x}_i^{t+1} - \mathbf{w}^t\|^2 + \left\langle \nabla_{\mathbf{c}_i^t} f_i(\widetilde{Q}_{\mathbf{c}_i^t}(\mathbf{x}_i^{t+1})), \mathbf{c}_i^{t+1} - \mathbf{c}_i^t \right\rangle + \frac{1}{2\eta_2}\|\mathbf{c}_i^{t+1} - \mathbf{c}_i^t\|^2$$

$$+ \left\langle \lambda_p \nabla_{\mathbf{c}_i^t} \widetilde{Q}_{\mathbf{c}_i^t}(\mathbf{x}_i^{t+1})(\widetilde{Q}_{\mathbf{c}_i^t}(\mathbf{x}_i^{t+1}) - \mathbf{w}_i^t), \mathbf{c}_i^{t+1} - \mathbf{c}_i^t \right\rangle \qquad (48)$$

$$+\left\langle \lambda_p \nabla_{\mathbf{c}_i^t}\widetilde{Q}_{\mathbf{c}_i^t}(\mathbf{x}_i^{t+1})(\mathbf{w}_i^t - \mathbf{w}^t), \mathbf{c}_i^{t+1} - \mathbf{c}_i^t \right\rangle \tag{49}$$

**Claim 15.** *Let*

$$B(\mathbf{c}_i^{t+1}) := \lambda R(\mathbf{x}_i^{t+1}, \mathbf{c}_i^{t+1}) + \left\langle \nabla_{\mathbf{c}_i^t} f_i(\widetilde{Q}_{\mathbf{c}_i^t}(\mathbf{x}_i^{t+1})), \mathbf{c}_i^{t+1} - \mathbf{c}_i^t \right\rangle + \frac{1}{2\eta_2}\|\mathbf{c}_i^{t+1} - \mathbf{c}_i^t\|^2$$

$$+ \left\langle \lambda_p \nabla_{\mathbf{c}_i^{t+1}}\widetilde{Q}_{\mathbf{c}_i^t}(\mathbf{x}_i^{t+1})(\widetilde{Q}_{\mathbf{c}_i^t}(\mathbf{x}_i^{t+1}) - \mathbf{w}_i^t), \mathbf{c}_i^{t+1} - \mathbf{c}_i^t \right\rangle$$

$$B(\mathbf{c}_i^t) := \lambda R(\mathbf{x}_i^{t+1}, \mathbf{c}_i^t).$$

*Then* $B(\mathbf{c}_i^{t+1}) \leq B(\mathbf{c}_i^t)$.

Substituting the bound from Claim 15 in (49),

$$F_i(\mathbf{x}_i^{t+1}, \mathbf{c}_i^{t+1}, \mathbf{w}^t) + \left(\frac{1}{2\eta_2} - \frac{\lambda_p L_{Q_2} + G L_{Q_2} + G_{Q_2} L L_{Q_2}}{2}\right)\|\mathbf{c}_i^{t+1} - \mathbf{c}_i^t\|^2$$

$$\leq f_i(\mathbf{x}_i^{t+1}) + f_i(\widetilde{Q}_{\mathbf{c}_i^t}(\mathbf{x}_i^{t+1})) + \lambda R(\mathbf{x}_i^{t+1}, \mathbf{c}_i^t) + \frac{\lambda_p}{2}\|\mathbf{x}_i^{t+1} - \mathbf{w}^t\|^2$$

$$+ \frac{\lambda_p}{2}\|\widetilde{Q}_{\mathbf{c}_i^t}(\mathbf{x}_i^{t+1}) - \mathbf{w}^t\|^2 + \left\langle \lambda_p \nabla_{\mathbf{c}_i^t}\widetilde{Q}_{\mathbf{c}_i^t}(\mathbf{x}_i^{t+1})(\mathbf{w}_i^t - \mathbf{w}^t), \mathbf{c}_i^{t+1} - \mathbf{c}_i^t \right\rangle$$

$$= F_i(\mathbf{x}_i^{t+1}, \mathbf{c}_i^t, \mathbf{w}^t) + \frac{\lambda_p}{2}\|\mathbf{c}_i^{t+1} - \mathbf{c}_i^t\|^2 + \frac{\lambda_p G_{Q_2}^2}{2}\|\mathbf{w}_i^t - \mathbf{w}^t\|^2$$

Substituting $\eta_2 = \frac{1}{2(\lambda_p + \lambda_p L_{Q_2} + G L_{Q_2} + G_{Q_2} L L_{Q_2})}$ gives us:

$$F_i(\mathbf{x}_i^{t+1}, \mathbf{c}_i^{t+1}, \mathbf{w}^t) + \frac{\lambda_p + \lambda_p L_{Q_2} + G L_{Q_2} + G_{Q_2} L L_{Q_2}}{2}\|\mathbf{c}_i^{t+1} - \mathbf{c}_i^t\|^2 \leq F_i(\mathbf{x}_i^{t+1}, \mathbf{c}_i^t, \mathbf{w}^t)$$

$$+ \frac{\lambda_p G_{Q_2}^2}{2}\|\mathbf{w}_i^t - \mathbf{w}^t\|^2. \tag{50}$$

### D.2.3  Sufficient Decrease Due to w

Now, we use $2\lambda_p$-smoothness of $F_i(\mathbf{x}, \mathbf{c}, \mathbf{w})$ with respect to $\mathbf{w}$:

$$F_i(\mathbf{x}_i^{t+1}, \mathbf{c}_i^{t+1}, \mathbf{w}^{t+1}) \leq F_i(\mathbf{x}_i^{t+1}, \mathbf{c}_i^{t+1}, \mathbf{w}^t) + \left\langle \nabla_{\mathbf{w}^t} F_i(\mathbf{x}_i^{t+1}, \mathbf{c}_i^{t+1}, \mathbf{w}^t), \mathbf{w}^{t+1} - \mathbf{w}^t \right\rangle$$

$$+ \lambda_p\|\mathbf{w}^{t+1} - \mathbf{w}^t\|^2$$

Then we have,

$$F_i(\mathbf{x}_i^{t+1}, \mathbf{c}_i^{t+1}, \mathbf{w}^{t+1}) \leq F_i(\mathbf{x}_i^{t+1}, \mathbf{c}_i^{t+1}, \mathbf{w}^t) + \left\langle \nabla_{\mathbf{w}^t} F_i(\mathbf{x}_i^{t+1}, \mathbf{c}_i^{t+1}, \mathbf{w}^t), \mathbf{w}^{t+1} - \mathbf{w}^t \right\rangle$$

$$+ \lambda_p\|\mathbf{w}^{t+1} - \mathbf{w}^t\|^2$$

$$= F_i(\mathbf{x}_i^{t+1}, \mathbf{c}_i^{t+1}, \mathbf{w}^t) + \left\langle \nabla_{\mathbf{w}^t} F_i(\mathbf{x}_i^{t+1}, \mathbf{c}_i^{t+1}, \mathbf{w}^t), \eta_3 \mathbf{g}^t \right\rangle + \lambda_p\|\eta_3 \mathbf{g}^t\|^2$$

$$= F_i(\mathbf{x}_i^{t+1}, \mathbf{c}_i^{t+1}, \mathbf{w}^t) - \eta_3 \left\langle \nabla_{\mathbf{w}^t} F_i(\mathbf{x}_i^{t+1}, \mathbf{c}_i^{t+1}, \mathbf{w}^t), \mathbf{g}^t - \nabla_{\mathbf{w}^t} F_i(\mathbf{x}_i^{t+1}, \mathbf{c}_i^{t+1}, \mathbf{w}^t)\right.$$

$$\left. + \nabla_{\mathbf{w}^t} F_i(\mathbf{x}_i^{t+1}, \mathbf{c}_i^{t+1}, \mathbf{w}^t) \right\rangle$$

$$+ \lambda_p 2\eta_3^2\|\mathbf{g}^t - \nabla_{\mathbf{w}^t} F_i(\mathbf{x}_i^{t+1}, \mathbf{c}_i^{t+1}, \mathbf{w}^t) + \nabla_{\mathbf{w}^t} F_i(\mathbf{x}_i^{t+1}, \mathbf{c}_i^{t+1}, \mathbf{w}^t)\|^2$$

$$= F_i(\mathbf{x}_i^{t+1}, \mathbf{c}_i^{t+1}, \mathbf{w}^t) - \eta_3\left\|\nabla_{\mathbf{w}^t} F_i(\mathbf{x}_i^{t+1}, \mathbf{c}_i^{t+1}, \mathbf{w}^t)\right\|^2$$

$$- \eta_3 \left\langle \nabla_{\mathbf{w}^t} F_i(\mathbf{x}_i^{t+1}, \mathbf{c}_i^{t+1}, \mathbf{w}^t), \mathbf{g}^t - \nabla_{\mathbf{w}^t} F_i(\mathbf{x}_i^{t+1}, \mathbf{c}_i^{t+1}, \mathbf{w}^t) \right\rangle$$

$$+ \lambda_p \eta_3^2\|\mathbf{g}^t - \nabla_{\mathbf{w}^t} F_i(\mathbf{x}_i^{t+1}, \mathbf{c}_i^{t+1}, \mathbf{w}^t) + \nabla_{\mathbf{w}^t} F_i(\mathbf{x}_i^{t+1}, \mathbf{c}_i^{t+1}, \mathbf{w}^t)\|^2$$

$$\leq F_i(\mathbf{x}_i^{t+1}, \mathbf{c}_i^{t+1}, \mathbf{w}^t) - \eta_3\left\|\nabla_{\mathbf{w}^t} F_i(\mathbf{x}_i^{t+1}, \mathbf{c}_i^{t+1}, \mathbf{w}^t)\right\|^2$$

$$+ \frac{\eta_3}{2}\left\|\nabla_{\mathbf{w}^t} F_i(\mathbf{x}_i^{t+1}, \mathbf{c}_i^{t+1}, \mathbf{w}^t)\right\|^2 + \frac{\eta_3}{2}\left\|\mathbf{g}^t - \nabla_{\mathbf{w}^t} F_i(\mathbf{x}_i^{t+1}, \mathbf{c}_i^{t+1}, \mathbf{w}^t)\right\|^2$$

$$+ 2\lambda_p \eta_3^2 \left( \left\| \mathbf{g}^t - \nabla_{\mathbf{w}^t} F_i(\mathbf{x}_i^{t+1}, \mathbf{c}_i^{t+1}, \mathbf{w}^t) \right\|^2 + \left\| \nabla_{\mathbf{w}^t} F_i(\mathbf{x}_i^{t+1}, \mathbf{c}_i^{t+1}, \mathbf{w}^t) \right\|^2 \right)$$

$$= F_i(\mathbf{x}_i^{t+1}, \mathbf{c}_i^{t+1}, \mathbf{w}^t) - (\frac{\eta_3}{2} - 2\lambda_p \eta_3^2) \left\| \nabla_{\mathbf{w}^t} F_i(\mathbf{x}_i^{t+1}, \mathbf{c}_i^{t+1}, \mathbf{w}^t) \right\|^2$$
$$+ (\frac{\eta_3}{2} + 2\lambda_p \eta_3^2) \left\| \mathbf{g}^t - \nabla_{\mathbf{w}_i^t} F_i(\mathbf{x}_i^{t+1}, \mathbf{c}_i^{t+1}, \mathbf{w}_i^t) + \nabla_{\mathbf{w}_i^t} F_i(\mathbf{x}_i^{t+1}, \mathbf{c}_i^{t+1}, \mathbf{w}_i^t) \right.$$
$$\left. - \nabla_{\mathbf{w}^t} F_i(\mathbf{x}_i^{t+1}, \mathbf{c}_i^{t+1}, \mathbf{w}^t) \right\|^2$$

$$\leq F_i(\mathbf{x}_i^{t+1}, \mathbf{c}_i^{t+1}, \mathbf{w}^t) - (\frac{\eta_3}{2} - 2\lambda_p \eta_3^2) \left\| \nabla_{\mathbf{w}^t} F_i(\mathbf{x}_i^{t+1}, \mathbf{c}_i^{t+1}, \mathbf{w}^t) \right\|^2$$
$$` \quad + (\eta_3 + 4\lambda_p \eta_3^2) \left\| \mathbf{g}^t - \nabla_{\mathbf{w}_i^t} F_i(\mathbf{x}_i^{t+1}, \mathbf{c}_i^{t+1}, \mathbf{w}_i^t) \right\|^2$$
$$+ (\eta_3 + 4\lambda_p \eta_3^2) \left\| \nabla_{\mathbf{w}_i^t} F_i(\mathbf{x}_i^{t+1}, \mathbf{c}_i^{t+1}, \mathbf{w}_i^t) - \nabla_{\mathbf{w}^t} F_i(\mathbf{x}_i^{t+1}, \mathbf{c}_i^{t+1}, \mathbf{w}^t) \right\|^2$$

$$\leq F_i(\mathbf{x}_i^{t+1}, \mathbf{c}_i^{t+1}, \mathbf{w}^t) - (\frac{\eta_3}{2} - 2\lambda_p \eta_3^2) \left\| \nabla_{\mathbf{w}^t} F_i(\mathbf{x}_i^{t+1}, \mathbf{c}_i^{t+1}, \mathbf{w}^t) \right\|^2$$
$$+ (\eta_3 + 4\lambda_p \eta_3^2) \left\| \mathbf{g}^t - \nabla_{\mathbf{w}_i^t} F_i(\mathbf{x}_i^{t+1}, \mathbf{c}_i^{t+1}, \mathbf{w}_i^t) \right\|^2$$
$$+ (\eta_3 + 4\lambda_p \eta_3^2) 4\lambda_p^2 \left\| \mathbf{w}_i^t - \mathbf{w}^t \right\|^2$$

Rearranging the terms, we have:

$$F_i(\mathbf{x}_i^{t+1}, \mathbf{c}_i^{t+1}, \mathbf{w}^{t+1}) + (\frac{\eta_3}{2} - 2\lambda_p \eta_3^2) \left\| \nabla_{\mathbf{w}^t} F_i(\mathbf{x}_i^{t+1}, \mathbf{c}_i^{t+1}, \mathbf{w}^t) \right\|^2$$
$$\leq (\eta_3 + 4\lambda_p \eta_3^2) \left\| \mathbf{g}^t - \nabla_{\mathbf{w}_i^t} F_i(\mathbf{x}_i^{t+1}, \mathbf{c}_i^{t+1}, \mathbf{w}_i^t) \right\|^2 + (\eta_3 + 4\lambda_p \eta_3^2) 4\lambda_p^2 \left\| \mathbf{w}_i^t - \mathbf{w}^t \right\|^2$$
$$+ F_i(\mathbf{x}_i^{t+1}, \mathbf{c}_i^{t+1}, \mathbf{w}^t) \tag{51}$$

### D.2.4 Overall Decrease

Define
$$L_x = 3\lambda_p + \lambda_p L_{Q_1} + L + G L_{Q_1} + G_{Q_1} L l_{Q_1} \tag{52}$$
$$L_c = \lambda_p + \lambda_p L_{Q_2} + G L_{Q_2} + G_{Q_2} L l_{Q_2}. \tag{53}$$

Summing (47), (50), (51) we get the overall decrease property:

$$F_i(\mathbf{x}_i^{t+1}, \mathbf{c}_i^{t+1}, \mathbf{w}^{t+1}) + (\frac{\eta_3}{2} - 2\lambda_p \eta_3^2) \left\| \nabla_{\mathbf{w}^t} F_i(\mathbf{x}_i^{t+1}, \mathbf{c}_i^{t+1}, \mathbf{w}^t) \right\|^2 + \frac{L_x}{2} \|\mathbf{x}_i^{t+1} - \mathbf{x}_i^t\|^2$$
$$+ \frac{L_c}{2} \|\mathbf{c}_i^{t+1} - \mathbf{c}_i^t\|^2 \leq (\eta_3 + 4\lambda_p \eta_3^2) \left\| \mathbf{g}^t - \nabla_{\mathbf{w}_i^t} F_i(\mathbf{x}_i^{t+1}, \mathbf{c}_i^{t+1}, \mathbf{w}_i^t) \right\|^2$$
$$+ (1 + G_{Q_1}^2 + G_{Q_2}^2 + 4\eta_3 \lambda_p + 16\lambda_p^2 \eta_3^2) \lambda_p \|\mathbf{w}_i^t - \mathbf{w}^t\|^2 + F_i(\mathbf{x}_i^t, \mathbf{c}_i^t, \mathbf{w}^t) \tag{54}$$

Let
$$L_{\min} = \min\{L_x, L_c, (\eta_3 - 4\lambda_p \eta_3^2)\}. \tag{55}$$

Then,
$$F_i(\mathbf{x}_i^{t+1}, \mathbf{c}_i^{t+1}, \mathbf{w}^{t+1}) + \frac{L_{\min}}{2} \left( \left\| \nabla_{\mathbf{w}^t} F_i(\mathbf{x}_i^{t+1}, \mathbf{c}_i^{t+1}, \mathbf{w}^t) \right\|^2 + \|\mathbf{x}_i^{t+1} - \mathbf{x}_i^t\|^2 + \|\mathbf{c}_i^{t+1} - \mathbf{c}_i^t\|^2 \right)$$
$$\leq (\eta_3 + 4\lambda_p \eta_3^2) \left\| \mathbf{g}^t - \nabla_{\mathbf{w}_i^t} F_i(\mathbf{x}_i^{t+1}, \mathbf{c}_i^{t+1}, \mathbf{w}_i^t) \right\|^2 + F_i(\mathbf{x}_i^t, \mathbf{c}_i^t, \mathbf{w}^t)$$
$$+ (1 + G_{Q_1}^2 + G_{Q_2}^2 + 4\eta_3 \lambda_p + 16\lambda_p^2 \eta_3^2) \lambda_p \|\mathbf{w}_i^t - \mathbf{w}^t\|^2 \tag{56}$$

### D.3 Bound on the Gradient

Now, we will use the first order optimality conditions due to proximal updates and bound the partial gradients with respect to variables $\mathbf{x}$ and $\mathbf{c}$. After obtaining bounds for partial gradients we will bound the overall gradient and use our results from Section D.2 to arrive at the final bound.

### D.3.1 Bound on the Gradient w.r.t. $\mathbf{x}_i$

Taking the derivative inside the minimization problem (42) with respect to $\mathbf{x}$ at $\mathbf{x} = \mathbf{x}_i^{t+1}$ and setting it to 0 gives the following optimality condition:

$$\nabla f_i(\mathbf{x}_i^t) + \nabla_{\mathbf{x}_i^t} f_i(\widetilde{Q}_{\mathbf{c}_i^t}(\mathbf{x}_i^t)) + \lambda_p(\mathbf{x}_i^t - \mathbf{w}_i^t) + \lambda_p \nabla_{\mathbf{x}_i^t} \widetilde{Q}_{\mathbf{c}_i^t}(\mathbf{x}_i^t)(\widetilde{Q}_{\mathbf{c}_i^t}(\mathbf{x}_i^t) - \mathbf{w}_i^t)$$
$$+ \frac{1}{\eta_1}(\mathbf{x}_i^{t+1} - \mathbf{x}_i^t) + \lambda \nabla_{\mathbf{x}_i^{t+1}} R(\mathbf{x}_i^{t+1}, \mathbf{c}_i^t) = 0 \quad (57)$$

Then we have,

$$\left\| \nabla_{\mathbf{x}_i^{t+1}} F_i(\mathbf{x}_i^{t+1}, \mathbf{c}_i^t, \mathbf{w}^t) \right\| = \left\| \nabla f_i(\mathbf{x}_i^{t+1}) + \nabla_{\mathbf{x}_i^{t+1}} f_i(\widetilde{Q}_{\mathbf{c}_i^t}(\mathbf{x}_i^{t+1})) + \lambda_p(\mathbf{x}_i^{t+1} - \mathbf{w}^t) \right.$$
$$\left. + \lambda_p \nabla_{\mathbf{x}_i^{t+1}} \widetilde{Q}_{\mathbf{c}_i^t}(\mathbf{x}_i^{t+1})(\widetilde{Q}_{\mathbf{c}_i^t}(\mathbf{x}_i^{t+1}) - \mathbf{w}^t) + \lambda \nabla_{\mathbf{x}_i^{t+1}} R(\mathbf{x}_i^{t+1}, \mathbf{c}_i^t) \right\|$$
$$\overset{(a)}{=} \left\| \nabla f_i(\mathbf{x}_i^{t+1}) - \nabla f_i(\mathbf{x}_i^t) + \nabla_{\mathbf{x}_i^{t+1}} f_i(\widetilde{Q}_{\mathbf{c}_i^t}(\mathbf{x}_i^{t+1})) - \nabla_{\mathbf{x}_i^t} f_i(\widetilde{Q}_{\mathbf{c}_i^t}(\mathbf{x}_i^t)) \right.$$
$$+ \lambda_p \nabla_{\mathbf{x}_i^{t+1}} \widetilde{Q}_{\mathbf{c}_i^t}(\mathbf{x}_i^{t+1})(\widetilde{Q}_{\mathbf{c}_i^t}(\mathbf{x}_i^{t+1}) - \mathbf{w}^t)$$
$$- \lambda_p \nabla_{\mathbf{x}_i^t} \widetilde{Q}_{\mathbf{c}_i^t}(\mathbf{x}_i^t)(\widetilde{Q}_{\mathbf{c}_i^t}(\mathbf{x}_i^t) - \mathbf{w}_i^t)$$
$$\left. + (\lambda_p - \frac{1}{\eta_1})(\mathbf{x}_i^{t+1} - \mathbf{x}_i^t) + \lambda_p(\mathbf{w}_i^t - \mathbf{w}^t) \right\|$$
$$\leq (\frac{1}{\eta_1} + \lambda_p + \lambda_p G_{Q_1} l_{Q_1} + L + G L_{Q_1} + G_{Q_1} L l_{Q_1}) \| \mathbf{x}_i^{t+1} - \mathbf{x}_i^t \|$$
$$+ \lambda_p(1 + G_{Q_1}) \| \mathbf{w}_i^t - \mathbf{w}^t \|$$

where (a) is from (57) and the last inequality is due to Lipschitz continuous gradients and triangle inequality. This implies:

$$\left\| \nabla_{\mathbf{x}_i^{t+1}} F_i(\mathbf{x}_i^{t+1}, \mathbf{c}_i^t, \mathbf{w}^t) \right\|^2 \leq 2(\frac{1}{\eta_1} + \lambda_p + \lambda_p G_{Q_1} l_{Q_1} + L + G L_{Q_1} + G_{Q_1} L l_{Q_1})^2 \| \mathbf{x}_i^{t+1} - \mathbf{x}_i^t \|^2$$
$$+ 2\lambda_p^2(1 + G_{Q_1})^2 \| \mathbf{w}_i^t - \mathbf{w}^t \|^2$$

Substituting $\eta_1 = \frac{1}{2(3\lambda_p + \lambda_p L_{Q_1} + L + G L_{Q_1} + G_{Q_1} L l_{Q_1})}$ we have:

$$\left\| \nabla_{\mathbf{x}_i^{t+1}} F_i(\mathbf{x}_i^{t+1}, \mathbf{c}_i^t, \mathbf{w}^t) \right\|^2 \leq 2(7\lambda_p + 2\lambda_p L_{Q_1} + \lambda_p G_{Q_1} l_{Q_1}$$
$$+ 3(L + G L_{Q_1} + G_{Q_1} L l_{Q_1}))^2 \| \mathbf{x}_i^{t+1} - \mathbf{x}_i^t \|^2$$
$$+ 2\lambda_p^2(1 + G_{Q_1})^2 \| \mathbf{w}_i^t - \mathbf{w}^t \|^2$$
$$= 18(\frac{7}{3}\lambda_p + \frac{2\lambda_p L_{Q_1}}{3} + \frac{\lambda_p G_{Q_1} l_{Q_1}}{3} + L + G l_{Q_1}$$
$$+ G_{Q_1} L l_{Q_1})^2 \| \mathbf{x}_i^{t+1} - \mathbf{x}_i^t \|^2$$
$$+ 2\lambda_p^2(1 + G_{Q_1})^2 \| \mathbf{w}_i^t - \mathbf{w}^t \|^2 \quad (58)$$

### D.3.2 Bound on the Gradient w.r.t. $\mathbf{c}_i$

Similarly. taking the derivative inside the minimization problem (43) with respect to $\mathbf{c}$ at $\mathbf{c}_i^{t+1}$ and setting it to 0 gives the following optimality condition:

$$\nabla_{\mathbf{c}_i^t} f_i(\widetilde{Q}_{\mathbf{c}_i^t}(\mathbf{x}_i^{t+1})) + \frac{1}{\eta_2}(\mathbf{c}_i^{t+1} - \mathbf{c}_i^t) + \lambda_p \nabla_{\mathbf{c}_i^t} \widetilde{Q}_{\mathbf{c}_i^t}(\mathbf{x}_i^{t+1})(\widetilde{Q}_{\mathbf{c}_i^t}(\mathbf{x}_i^{t+1}) - \mathbf{w}_i^t)$$
$$+ \lambda \nabla_{\mathbf{c}_i^{t+1}} R(\mathbf{x}_i^{t+1}, \mathbf{c}_i^{t+1}) = 0 \quad (59)$$

Then we have

$$\left\| \nabla_{\mathbf{c}_i^{t+1}} F_i(\mathbf{x}_i^{t+1}, \mathbf{c}_i^{t+1}, \mathbf{w}^t) \right\| = \left\| \nabla_{\mathbf{c}_i^{t+1}} f_i(\widetilde{Q}_{\mathbf{c}_i^{t+1}}(\mathbf{x}_i^{t+1})) + \lambda_p \nabla_{\mathbf{c}_i^{t+1}} \widetilde{Q}_{\mathbf{c}_i^{t+1}}(\mathbf{x}_i^{t+1})(\widetilde{Q}_{\mathbf{c}_i^{t+1}}(\mathbf{x}_i^{t+1}) - \mathbf{w}_i^t) \right.$$

$$+ \lambda \nabla_{\mathbf{c}_i^{t+1}} R(\mathbf{x}_i^{t+1}, \mathbf{c}_i^{t+1}) \Big\|$$

$$= \Big\| \nabla_{\mathbf{c}_i^{t+1}} f_i(\widetilde{Q}_{\mathbf{c}_i^{t+1}}(\mathbf{x}_i^{t+1})) - \nabla_{\mathbf{c}_i^t} f_i(\widetilde{Q}_{\mathbf{c}_i^t}(\mathbf{x}_i^{t+1})) + \frac{1}{\eta_2}(\mathbf{c}_i^t - \mathbf{c}_i^{t+1})$$

$$+ \lambda_p \nabla_{\mathbf{c}_i^{t+1}} \widetilde{Q}_{\mathbf{c}_i^{t+1}}(\mathbf{x}_i^{t+1})(\widetilde{Q}_{\mathbf{c}_i^{t+1}}(\mathbf{x}_i^{t+1}) - \mathbf{w}_i^t)$$

$$- \lambda_p \nabla_{\mathbf{c}_i^t} \widetilde{Q}_{\mathbf{c}_i^t}(\mathbf{x}_i^{t+1})(\widetilde{Q}_{\mathbf{c}_i^t}(\mathbf{x}_i^{t+1}) - \mathbf{w}^t) \Big\|$$

$$\leq (\frac{1}{\eta_2} + \lambda_p G_{Q_2} l_{Q_2} + GL_{Q_2} + G_{Q_2} Ll_{Q_2}) \| \mathbf{c}_i^{t+1} - \mathbf{c}_i^t \|$$

$$+ \lambda_p G_{Q_2} \| \mathbf{w}_i^t - \mathbf{w}^t \|$$

the first equality is due to (59) and the last inequality is due to Lipschitz continuous gradient and triangle inequality. As a result we have,

$$\left\| \nabla_{\mathbf{c}_i^{t+1}} F_i(\mathbf{x}_i^{t+1}, \mathbf{c}_i^{t+1}, \mathbf{w}^t) \right\|^2 \leq 2(\frac{1}{\eta_2} + \lambda_p G_{Q_2} l_{Q_2} + GL_{Q_2} + G_{Q_2} Ll_{Q_2})^2 \| \mathbf{c}_i^{t+1} - \mathbf{c}_i^t \|^2$$

$$+ 2\lambda_p^2 G_{Q_2}^2 \| \mathbf{w}_i^t - \mathbf{w}^t \|^2 \tag{60}$$

substituting $\eta_2 = \frac{1}{2(\lambda_p + \lambda_p L_{Q_2} + GL_{Q_2} + G_{Q_2} LL_{Q_2})}$ we have:

$$\left\| \nabla_{\mathbf{c}_i^{t+1}} F_i(\mathbf{x}_i^{t+1}, \mathbf{c}_i^{t+1}, \mathbf{w}^t) \right\|^2 \leq 18(\frac{2\lambda_p}{3} + \frac{2\lambda_p L_{Q_2}}{3} + GL_{Q_2} + G_{Q_2} LL_{Q_2})^2 \| \mathbf{c}_i^{t+1} - \mathbf{c}_i^t \|^2$$

$$+ 2\lambda_p^2 G_{Q_2}^2 \| \mathbf{w}_i^t - \mathbf{w}^t \|^2$$

### D.3.3 Overall Bound

Then, let us write $\|\mathbf{G}_i^t\|^2$ :

$$\|\mathbf{G}_i^t\|^2 = \left\| [\nabla_{\mathbf{x}_i^{t+1}} F_i(\mathbf{x}_i^{t+1}, \mathbf{c}_i^t, \mathbf{w}^t)^T, \nabla_{\mathbf{c}_i^{t+1}} F_i(\mathbf{x}_i^{t+1}, \mathbf{c}_i^{t+1}, \mathbf{w}^t)^T, \nabla_{\mathbf{w}^t} F_i(\mathbf{x}_i^{t+1}, \mathbf{c}_i^{t+1}, \mathbf{w}^t)^T]^T \right\|^2$$

$$= \left\| \nabla_{\mathbf{x}_i^{t+1}} F_i(\mathbf{x}_i^{t+1}, \mathbf{c}_i^t, \mathbf{w}^t) \right\|^2 + \left\| \nabla_{\mathbf{c}_i^{t+1}} F_i(\mathbf{x}_i^{t+1}, \mathbf{c}_i^{t+1}, \mathbf{w}^t) \right\|^2 + \left\| \nabla_{\mathbf{w}^t} F_i(\mathbf{x}_i^{t+1}, \mathbf{c}_i^{t+1}, \mathbf{w}^t) \right\|^2$$

$$\leq 18(\frac{7}{3}\lambda_p + \frac{2\lambda_p L_{Q_1}}{3} + \frac{\lambda_p G_{Q_1} l_{Q_1}}{3} + L + Gl_{Q_1} + G_{Q_1} Ll_{Q_1})^2 \| \mathbf{x}_i^{t+1} - \mathbf{x}_i^t \|^2$$

$$+ 18(\frac{2\lambda_p}{3} + \frac{2\lambda_p L_{Q_2}}{3} + GL_{Q_2} + G_{Q_2} LL_{Q_2})^2 \| \mathbf{c}_i^{t+1} - \mathbf{c}_i^t \|^2 + \left\| \nabla_{\mathbf{w}^t} F_i(\mathbf{x}_i^{t+1}, \mathbf{c}_i^{t+1}, \mathbf{w}^t) \right\|^2$$

$$+ 2\lambda_p^2 ((1 + G_{Q_1})^2 + G_{Q_2}^2) \| \mathbf{w}_i^t - \mathbf{w}^t \|^2$$

where the last inequality is due to (58) and (60). Let

$$L_{\max} = \max \left\{ \sqrt{\frac{1}{18}}, \frac{2\lambda_p}{3} + \frac{2\lambda_p L_{Q_2}}{3} + GL_{Q_2} + G_{Q_2} LL_{Q_2}, \right.$$

$$\left. \frac{7}{3}\lambda_p + \frac{2\lambda_p L_{Q_1}}{3} + \frac{\lambda_p G_{Q_1} l_{Q_1}}{3} + L + Gl_{Q_1} + G_{Q_1} Ll_{Q_1} \right\}. \tag{61}$$

Then,

$$\left\| [\nabla_{\mathbf{x}_i^{t+1}} F_i(\mathbf{x}_i^{t+1}, \mathbf{c}_i^t, \mathbf{w}^t)^T, \nabla_{\mathbf{c}_i^{t+1}} F_i(\mathbf{x}_i^{t+1}, \mathbf{c}_i^{t+1}, \mathbf{w}^t)^T, \nabla_{\mathbf{w}^t} F_i(\mathbf{x}_i^{t+1}, \mathbf{c}_i^{t+1}, \mathbf{w}^t)^T]^T \right\|^2$$

$$\leq 18 L_{\max}^2 (\| \mathbf{x}_i^{t+1} - \mathbf{x}_i^t \|^2 + \| \mathbf{c}_i^{t+1} - \mathbf{c}_i^t \|^2 + \left\| \nabla_{\mathbf{w}^t} F_i(\mathbf{x}_i^{t+1}, \mathbf{c}_i^{t+1}, \mathbf{w}^t) \right\|^2)$$

$$+ 2\lambda_p^2 ((1 + G_{Q_1})^2 + G_{Q_2}^2) \| \mathbf{w}_i^t - \mathbf{w}^t \|^2$$

$$\overset{(a)}{\leq} 36 \frac{L_{\max}^2}{L_{\min}} \left[ (\eta_3 + 4\lambda_p \eta_3^2) \left\| \mathbf{g}^t - \nabla_{\mathbf{w}_i^t} F_i(\mathbf{x}_i^{t+1}, \mathbf{c}_i^{t+1}, \mathbf{w}_i^t) \right\|^2 \right.$$

$$+ (1 + G_{Q_1}^2 + G_{Q_2}^2 + 4\eta_3\lambda_p + 16\lambda_p^2\eta_3^2)\lambda_p\|\mathbf{w}_i^t - \mathbf{w}^t\|^2$$
$$+ F_i(\mathbf{x}_i^t, \mathbf{c}_i^t, \mathbf{w}^t) - F_i(\mathbf{x}_i^{t+1}, \mathbf{c}_i^{t+1}, \mathbf{w}^{t+1})\Big] + 2\lambda_p^2((1 + G_{Q_1})^2 + G_{Q_2}^2)\|\mathbf{w}_i^t - \mathbf{w}^t\|^2. \quad (62)$$

In (a) we use the bound from (56), and $L_{\min}$ is defined in (55).

Now we state a useful lemma that enables us to relate local version of the global model, $\mathbf{w}_i$, to global model itself, $\mathbf{w}$.

**Lemma 2.** *Let $\eta_3$ be chosen such that $\eta_3 \leq \sqrt{\frac{1}{48\tau^2\lambda_p^2}}$, then we have,*

$$\frac{1}{T}\sum_{t=0}^{T-1}\frac{1}{n}\sum_{i=1}^{n}\|\mathbf{w}^t - \mathbf{w}_i^t\|^2 \leq 6\tau^2\eta_3^2\kappa.$$

As a corollary:

**Corollary 2.** *Recall, $\mathbf{g}^t = \frac{1}{n}\sum_{i=1}^n \nabla_{\mathbf{w}^t} F_i(\mathbf{x}_i^{t+1}, \mathbf{c}_i^{t+1}, \mathbf{w}_i^t)$. Then, we have:*

$$\frac{1}{T}\sum_{t=0}^{T-1}\frac{1}{n}\sum_{i=1}^{n}\left\|\mathbf{g}^t - \nabla_{\mathbf{w}_i^t}F_i(\mathbf{x}_i^{t+1}, \mathbf{c}_i^{t+1}, \mathbf{w}_i^t)\right\|^2 \leq 144\lambda_p^2\tau^2\eta_3^2\kappa + 3\kappa.$$

See end of the section for the proofs. Using Lemma 2 and Corollary 2, summing the bound in (62) over time and clients, dividing by $T$ and $n$:

$$\frac{1}{T}\sum_{t=0}^{T-1}\frac{1}{n}\sum_{i=1}^{n}\|\mathbf{G}_i^t\|^2 \leq 36\frac{L_{\max}^2}{L_{\min}}\Big[(\eta_3 + 4\lambda_p\eta_3^2)(144\lambda_p^2\tau^2\eta_3^2\kappa + 3\kappa)$$

$$+ (1 + G_{Q_1}^2 + G_{Q_2}^2 + 4\eta_3\lambda_p + 16\lambda_p^2\eta_3^2)\lambda_p 6\tau^2\eta_3^2\kappa$$

$$+ \frac{\sum_{i=1}^n\left(F_i(\mathbf{x}_i^0, \mathbf{c}_i^0, \mathbf{w}_i^0) - F_i(\mathbf{x}_i^T, \mathbf{c}_i^T, \mathbf{w}_i^T)\right)}{nT}\Big]$$

$$+ 2\lambda_p^2((1 + G_{Q_1})^2 + G_{Q_2}^2)6\tau^2\eta_3^2\kappa$$

$$= 36\frac{L_{\max}^2}{L_{\min}}\Big[6\tau^2\eta_3^2\kappa(\lambda_p(1 + G_{Q_1}^2 + G_{Q_2}^2) + 28\eta_3\lambda_p^2 + 112\eta_3^2\lambda_p^3)$$

$$+ 3\eta_3\kappa + 12\lambda_p\eta_3^2\kappa + \frac{\Delta_F}{T}\Big] + 12\lambda_p^2((1 + G_{Q_1})^2 + G_{Q_2}^2)\tau^2\eta_3^2\kappa \quad (63)$$

where $\Delta_F = \frac{\sum_{i=1}^n\left(F_i(\mathbf{x}_i^0, \mathbf{c}_i^0, \mathbf{w}_i^0) - F_i(\mathbf{x}_i^T, \mathbf{c}_i^T, \mathbf{w}_i^T)\right)}{n}$.

**Choice of $\eta_3$.** Assuming $\tau \leq \sqrt{T}$ we can take $\eta_3 = \frac{1}{4\lambda_p\sqrt{T}}$, details of this choice is discussed in the end of this section, and after some algebra we have the end result:

$$\frac{1}{T}\sum_{t=0}^{T-1}\frac{1}{n}\sum_{i=1}^{n}\|\mathbf{G}_i^t\|^2 \leq \frac{54L_{\max}^2\tau\kappa(1 + G_{Q_1}^2 + G_{Q_2}^2) + 54L_{\max}^2 + 576L_{\max}^2\lambda_p\Delta_F}{\sqrt{T}}$$

$$+ \frac{1512L_{\max}^2\tau^2\kappa + \frac{3}{16}\tau^2\kappa^2((1 + G_{Q_1}^2) + G_{Q_2}^2)}{T}$$

$$+ \frac{6048L_{\max}^2\tau^2\kappa}{T^{\frac{3}{2}}} + 216L_{\max}^2\kappa$$

This concludes the proof.

### D.4 Proof of Lemma 2 and Corollary 2

Let us restate and prove Lemma 2,

**Lemma** (Restating Lemma 2)**.**

$$\frac{1}{T}\sum_{t=0}^{T-1}\frac{1}{n}\sum_{i=1}^{n}\|\mathbf{w}^t - \mathbf{w}_i^t\|^2 \leq 6\tau^2\eta_3^2\kappa \quad (64)$$

*Proof.* Let $t_c$ be the latest synchronization time before t. Define $\gamma_t = \frac{1}{n}\sum_{i=1}^n \|\mathbf{w}^t - \mathbf{w}_i^t\|^2$. Then:

$$\gamma_t = \frac{1}{n}\sum_{i=1}^n \left\| \mathbf{w}^{tc} - \frac{\eta_3}{n}\sum_{j=t_c}^t\sum_{k=1}^n \nabla_{\mathbf{w}_k^j} F_k(\mathbf{x}_k^{j+1}, \mathbf{c}_k^{j+1}, \mathbf{w}_k^j) - \left(\mathbf{w}^{tc} - \eta_3\sum_{j=t_c}^t \nabla_{\mathbf{w}_i^j} F_i(\mathbf{x}_i^{j+1}, \mathbf{c}_i^{j+1}, \mathbf{w}_i^j)\right) \right\|^2$$

$$\overset{(a)}{\leq} \tau\sum_{j=t_c}^t \frac{\eta_3^2}{n}\sum_{i=1}^n \left\| \frac{1}{n}\sum_{k=1}^n \nabla_{\mathbf{w}_k^j} F_k(\mathbf{x}_k^{j+1}, \mathbf{c}_k^{j+1}, \mathbf{w}_k^j) - \nabla_{\mathbf{w}_i^j} F_i(\mathbf{x}_i^{j+1}, \mathbf{c}_i^{j+1}, \mathbf{w}_i^j) \right\|^2 \qquad (\star 1)$$

$$= \tau\sum_{j=t_c}^t \frac{\eta_3^2}{n}\sum_{i=1}^n \Bigg[ \Bigg\| \frac{1}{n}\sum_{k=1}^n \left( \nabla_{\mathbf{w}_k^j} F_k(\mathbf{x}_k^{j+1}, \mathbf{c}_k^{j+1}, \mathbf{w}_k^j) - \nabla_{\mathbf{w}^j} F_k(\mathbf{x}_k^{j+1}, \mathbf{c}_k^{j+1}, \mathbf{w}^j) \right.$$

$$+ \nabla_{\mathbf{w}^j} F_k(\mathbf{x}_k^{j+1}, \mathbf{c}_k^{j+1}, \mathbf{w}^j)\Bigg) - \nabla_{\mathbf{w}^j} F_i(\mathbf{x}_k^{j+1}, \mathbf{c}_k^{j+1}, \mathbf{w}^j)$$

$$\left. + \nabla_{\mathbf{w}^j} F_i(\mathbf{x}_k^{j+1}, \mathbf{c}_k^{j+1}, \mathbf{w}^j) - \nabla_{\mathbf{w}_i^j} F_i(\mathbf{x}_k^{j+1}, \mathbf{c}_k^{j+1}, \mathbf{w}_i^j) \right\|^2 \Bigg]$$

$$\leq \tau\sum_{j=t_c}^{t_c+\tau} 3\frac{\eta_3^2}{n}\sum_{i=1}^n \Bigg[ \Bigg\| \frac{1}{n}\sum_{k=1}^n \left( \nabla_{\mathbf{w}_k^j} F_k(\mathbf{x}_k^{j+1}, \mathbf{c}_k^{j+1}, \mathbf{w}_k^j) - \nabla_{\mathbf{w}^j} F_k(\mathbf{x}_k^{j+1}, \mathbf{c}_k^{j+1}, \mathbf{w}^j) \right) \Bigg\|^2$$

$$+ \Bigg\| \frac{1}{n}\sum_{k=1}^n \nabla_{\mathbf{w}^j} F_k(\mathbf{x}_k^{j+1}, \mathbf{c}_k^{j+1}, \mathbf{w}^j) - \nabla_{\mathbf{w}^j} F_i(\mathbf{x}_i^{j+1}, \mathbf{c}_i^{j+1}, \mathbf{w}^j) \Bigg\|^2$$

$$+ \Bigg\| \nabla_{\mathbf{w}^j} F_i(\mathbf{x}_i^{j+1}, \mathbf{c}_i^{j+1}, \mathbf{w}^j) - \nabla_{\mathbf{w}_i^j} F_i(\mathbf{x}_i^{j+1}, \mathbf{c}_i^{j+1}, \mathbf{w}_i^j) \Bigg\|^2 \Bigg]$$

$$\leq \tau\sum_{j=t_c}^{t_c+\tau} 3\frac{\eta_3^2}{n}\sum_{i=1}^n \left[ \frac{4\lambda_p^2 n}{n^2}\sum_{k=1}^n \|\mathbf{w}_k^j - \mathbf{w}^j\|^2 + \kappa_i + 4\lambda_p^2 \|\mathbf{w}^j - \mathbf{w}_i^j\|^2 \right]$$

$$= \tau\sum_{j=t_c}^{t_c+\tau} 3\eta_3^2(4\lambda_p^2\gamma_j + \kappa + 4\lambda_p^2\gamma_j) \qquad (\star 2)$$

in (a) we use the facts that $\|\sum_{i=1}^K a_i\|^2 \leq K\sum_{i=1}^K \|a_i\|^2$, $t \leq \tau + t_c$ and that we are summing over non-negative terms. As a result, we have:

$$\gamma_t \leq \tau\sum_{j=t_c}^{t_c+\tau} 3\eta_3^2(8\lambda_p^2\gamma_j + \kappa)$$

$$\implies \sum_{t=t_c}^{t_c+\tau} \gamma_t \leq \sum_{t=t_c}^{t_c+\tau}\sum_{j=t_c}^{t_c+\tau} 3\tau\eta_3^2(8\lambda_p^2\gamma_j + \kappa) = 24\tau^2\eta_3^2\lambda_p^2\sum_{j=t_c}^{t_c+\tau}\gamma_j + 3\tau^3\eta_3^2\kappa$$

Let us choose $\eta_3$ such that $24\tau^2\eta_3^2\lambda_p^2 \leq \frac{1}{2} \Leftrightarrow \eta_3 \leq \sqrt{\frac{1}{48\tau^2\lambda_p^2}}$, sum over all syncronization times, and divide both sides by $T$:

$$\frac{1}{T}\sum_{t=0}^{T-1}\gamma_t \leq \frac{1}{2}\sum_{j=0}^{T-1}\gamma_j + 3\tau^2\eta_3^2\kappa$$

$$\implies \frac{1}{T}\sum_{t=0}^{T-1}\gamma_t \leq 6\tau^2\eta_3^2\kappa$$

$\square$

Let us restate and prove Corollary 2.

**Corollary** (Restating Corollary 2). *Recall,* $\mathbf{g}^t = \frac{1}{n}\sum_{i=1}^n \nabla_{\mathbf{w}^t} F_i(\mathbf{x}_i^{t+1}, \mathbf{c}_i^{t+1}, \mathbf{w}_i^t)$. *Then, we have:*

$$\frac{1}{T}\sum_{t=0}^{T-1}\frac{1}{n}\sum_{i=1}^n \left\| \mathbf{g}^t - \nabla_{\mathbf{w}_i^t} F_i(\mathbf{x}_i^{t+1}, \mathbf{c}_i^{t+1}, \mathbf{w}_i^t) \right\|^2 \leq 36\lambda_p^2\tau^2\eta_3^2\kappa + 3\kappa$$

*Proof.* From $(\star 1) \leq$ Lemma $\star 2$ in the proof of (2), we have:

$$\sum_{t=t_c}^{t_c+\tau} \frac{1}{n} \sum_{i=1}^{n} \left\| \mathbf{g}^t - \nabla_{\mathbf{w}_i^t} F_i(\mathbf{x}_i^{t+1}, \mathbf{c}_i^{t+1}, \mathbf{w}_i^t) \right\|^2 \leq \sum_{j=t_c}^{t_c+\tau} 3(8\lambda_p^2 \gamma_j + \kappa)$$

Summing over all $t_c$ and dividing by $T$:

$$\frac{1}{T} \sum_{t=0}^{T-1} \frac{1}{n} \sum_{i=1}^{n} \left\| \mathbf{g}^t - \nabla_{\mathbf{w}_i^t} F_i(\mathbf{x}_i^{t+1}, \mathbf{c}_i^{t+1}, \mathbf{w}_i^t) \right\|^2 \leq 24\lambda_p^2 \frac{1}{T} \sum_{t=0}^{T-1} \gamma_t + 3\kappa$$

$$\overset{(a)}{\leq} 144\lambda_p^2 \tau^2 \eta_3^2 \kappa + 3\kappa$$

where (a) is from (2).

### D.5 Choice of $\eta_3$

Note that, in Lemma 2 we chose $\eta_3$ such that $\eta_3 \leq \frac{1}{\sqrt{48}\lambda_p\tau}$. Now, we further introduce upper bounds on $\eta_3$.

- We can choose $\eta_3$ small enough so that $L_{\min} = \eta_3 - 4\lambda_p \eta_3^2$; see (55) for the definition of $L_{\min}$.

- We can choose $\eta_3$ small enough so that $\eta_3 - 4\lambda_p \eta_3^2 \geq \frac{\eta_3}{2}$. This is equivalent to choosing $\eta_3 \leq \frac{1}{8\lambda_p}$.

These two choices implies $L_{\min} \geq \frac{\eta_3}{2}$.

In the end, we have 2 critical constraints on $\eta_3$, $\{\eta_3 : \eta_3 \leq \frac{1}{\sqrt{48}\lambda_p\tau}, \eta_3 \leq \frac{1}{8\lambda_p}\}$. Then, let $\{\eta_3 : \eta_3 \leq \frac{1}{8\lambda_p\tau}\}$. Moreover, assuming $\tau \leq \sqrt{T}$ we can take $\eta_3 = \frac{1}{8\lambda_p\sqrt{T}}$ this choice clearly satisfies the above constraints.

From (63) we have,

$$\frac{1}{T} \sum_{t=0}^{T-1} \frac{1}{n} \sum_{i=1}^{n} \left\| \mathbf{G}_i^t \right\|^2 \leq 36 \frac{L_{\max}^2}{L_{\min}} \Big[ 6\tau^2 \eta_3^2 \kappa(\lambda_p(1 + G_{Q_1}^2 + G_{Q_2}^2) + 28\eta_3\lambda_p^2 + 112\eta_3^2\lambda_p^3)$$

$$+ 3\eta_3\kappa + 12\lambda_p\eta_3^2\kappa + \frac{\Delta_F}{T} \Big] + 12\lambda_p^2((1 + G_{Q_1})^2 + G_{Q_2}^2)\tau^2\eta_3^2\kappa$$

$$\leq 72 \frac{L_{\max}^2}{\eta_3} \Big[ 6\tau^2 \eta_3^2 \kappa(\lambda_p(1 + G_{Q_1}^2 + G_{Q_2}^2) + 28\eta_3\lambda_p^2 + 112\eta_3^2\lambda_p^3)$$

$$+ 3\eta_3\kappa + 12\lambda_p\eta_3^2\kappa + \frac{\Delta_F}{T} \Big] + 12\lambda_p^2((1 + G_{Q_1})^2 + G_{Q_2}^2)\tau^2\eta_3^2\kappa$$

$$= 72 L_{\max}^2 \Big[ 6\tau^2 \eta_3 \kappa(\lambda_p(1 + G_{Q_1}^2 + G_{Q_2}^2) + 28\eta_3\lambda_p^2 + 112\eta_3^2\lambda_p^3)$$

$$+ 3\kappa + 12\lambda_p\eta_3\kappa + \frac{\Delta_F}{\eta_3 T} \Big] + 12\lambda_p^2((1 + G_{Q_1})^2 + G_{Q_2}^2)\tau^2\eta_3^2\kappa$$

Now, we plug in $\eta_3 = \frac{1}{8\lambda_p\sqrt{T}}$:

$$= 72 L_{\max}^2 \Big[ \frac{3}{4}\tau^2\kappa \frac{(1 + G_{Q_1}^2 + G_{Q_2}^2)}{\sqrt{T}} + 21\tau^2\kappa\frac{1}{T} + 84\tau^2\kappa\frac{1}{T^{\frac{3}{2}}}$$

$$+ 3\kappa + \frac{3}{4}\frac{1}{\sqrt{T}} + \frac{8\lambda_p\Delta_F}{\sqrt{T}} \Big] + \frac{3}{16}((1 + G_{Q_1}^2) + G_{Q_2}^2)\tau^2\kappa^2\frac{1}{T}$$

$$= \frac{54L_{\max}^2\tau\kappa(1 + G_{Q_1}^2 + G_{Q_2}^2) + 54L_{\max}^2 + 576L_{\max}^2\lambda_p\Delta_F}{\sqrt{T}}$$
$$+ \frac{1512L_{\max}^2\tau^2\kappa + \frac{3}{16}\tau^2\kappa^2((1 + G_{Q_1}^2) + G_{Q_2}^2)}{T}$$
$$+ \frac{6048L_{\max}^2\tau^2\kappa}{T^{\frac{3}{2}}} + 216L_{\max}^2\kappa$$

where $L_{\max}$ is defined in (61).

This completes the proof of Theorem 3. $\qquad\square$

# E  Additional Details and Results for Experiments

In this section, we first discuss the implementation details for the prox steps for Algorithm 1 and Algorithm 2 in Section E.1. Section E.2 discusses implementation details for the algorithms along with hyperparameters which was omitted in Section 5 of the main paper due to space constraints. Section E.3 provides additional experiments for comparison of QuPeD (Algorithm 2) with other personalized learning schemes with additional types of heterogeneity and datasets. Section E.4 discusses the hardware resources on which the algorithms were implemented and the training times.

## E.1  Proximal Updates

For the implementation of Algorithm 1,2, we consider $\ell_1$-loss for the distance function $R(\mathbf{x}, \mathbf{c})$. In other words, $R(\mathbf{x}, \mathbf{c}) = \min\{\frac{1}{2}\|\mathbf{z} - \mathbf{x}\|_1 : z_i \in \{c_1, \cdots, c_m\}, \forall i\}$. For simplicity, we define $\mathcal{C} = \{\mathbf{z} : z_i \in \{c_1, \cdots, c_m\}, \forall i\}$. For the first type of update (update of $\mathbf{x}$) we have:

$$\begin{aligned}
\text{prox}_{\eta_1\lambda R(\cdot,\mathbf{c})}(\mathbf{y}) &= \underset{\mathbf{x}\in\mathbb{R}^d}{\arg\min}\left\{\frac{1}{2\eta_1}\|\mathbf{x} - \mathbf{y}\|_2^2 + \lambda R(\mathbf{x}, \mathbf{c})\right\} \\
&= \underset{\mathbf{x}\in\mathbb{R}^d}{\arg\min}\left\{\frac{1}{2\eta_1}\|\mathbf{x} - \mathbf{y}\|_2^2 + \frac{\lambda}{2}\underset{\mathbf{z}\in\mathcal{C}}{\min}\|\mathbf{z} - \mathbf{x}\|_1\right\} \\
&= \underset{\mathbf{x}\in\mathbb{R}^d}{\arg\min}\,\underset{\mathbf{z}\in\mathcal{C}}{\min}\left\{\frac{1}{2\eta_1}\|\mathbf{x} - \mathbf{y}\|_2^2 + \frac{\lambda}{2}\|\mathbf{z} - \mathbf{x}\|_1\right\}
\end{aligned}$$

(65)

This corresponds to solving:

$$\underset{\mathbf{z}\in\mathcal{C}}{\min}\,\underset{\mathbf{x}\in\mathbb{R}^d}{\min}\left\{\frac{1}{\eta_1}\|\mathbf{x} - \mathbf{y}\|_2^2 + \lambda\|\mathbf{z} - \mathbf{x}\|_1\right\}$$

Since both $\ell_1$ and squared $\ell_2$ norms are decomposable; if we fix $\mathbf{z}$, for the inner problem we have the following solution to soft thresholding:

$$x^\star(\mathbf{z})_i = \begin{cases} y_i - \frac{\lambda\eta_1}{2}, & \text{if } y_i - \frac{\lambda\eta_1}{2} > z_i \\ y_i + \frac{\lambda\eta_1}{2}, & \text{if } y_i + \frac{\lambda\eta_1}{2} < z_i \\ z_i, & \text{otherwise} \end{cases}$$

(66)

As a result we have:

$$\underset{\mathbf{z}\in\mathcal{C}}{\min}\left\{\frac{1}{\eta_1}\|x^\star(\mathbf{z}) - \mathbf{y}\|_2^2 + \lambda\|\mathbf{z} - x^\star(\mathbf{z})\|_1\right\}$$

This problem is separable, in other words we have:

$$\mathbf{z}_i^\star = \underset{z_i\in\{c_1,\cdots,c_m\}}{\arg\min}\left\{\frac{1}{\eta_1}(x^\star(\mathbf{z})_i - y_i)^2 + \lambda|z_i - x^\star(\mathbf{z})_i|\right\} \forall i$$

Substituting $x^\star(\mathbf{z})_i$ and solving for $z_i$ gives us:

$$\mathbf{z}_i^\star = \operatorname*{arg\,min}_{z_i \in \{c_1, \cdots, c_m\}} \{|z_i - y_i|\} \ \forall i$$

Or equivalently we have,

$$\mathbf{z}^\star = \operatorname*{arg\,min}_{\mathbf{z} \in \mathcal{C}} \|\mathbf{z} - \mathbf{y}\|_1 = Q_{\mathbf{c}}(\mathbf{y}) \tag{67}$$

As a result, $\mathrm{prox}_{\eta_1 \lambda R(\cdot, \mathbf{c})}(\cdot)$ becomes the soft thresholding operator:

$$\mathrm{prox}_{\eta_1 \lambda R(\cdot, \mathbf{c})}(\mathbf{y})_i = \begin{cases} y_i - \frac{\lambda \eta_1}{2}, & \text{if } y_i \geq Q_{\mathbf{c}}(\mathbf{y})_i + \frac{\lambda \eta_1}{2} \\ y_i + \frac{\lambda \eta_1}{2}, & \text{if } y_i \leq Q_{\mathbf{c}}(\mathbf{y})_i - \frac{\lambda \eta_1}{2} \\ Q_{\mathbf{c}}(\mathbf{y})_i, & \text{otherwise} \end{cases} \tag{68}$$

And for the second type of update we have $\mathrm{prox}_{\eta_2 \lambda R(\mathbf{x}, \cdot)}(\cdot)$ becomes:

$$\begin{aligned} \mathrm{prox}_{\eta_2 \lambda R(\mathbf{x}, \cdot)}(\boldsymbol{\mu}) &= \operatorname*{arg\,min}_{\mathbf{c} \in \mathbb{R}^m} \left\{ \frac{1}{2\eta_2} \|\mathbf{c} - \boldsymbol{\mu}\|_2^2 + \lambda R(\mathbf{x}, \mathbf{c}) \right\} \\ &= \operatorname*{arg\,min}_{\mathbf{c} \in \mathbb{R}^m} \left\{ \frac{1}{2\eta_2} \|\mathbf{c} - \boldsymbol{\mu}\|_2^2 + \frac{\lambda}{2} \min_{\mathbf{z} \in \mathcal{C}} \|\mathbf{z} - \mathbf{x}\|_1 \right\} \\ &= \operatorname*{arg\,min}_{\mathbf{c} \in \mathbb{R}^m} \left\{ \frac{1}{2\eta_2} \|\mathbf{c} - \boldsymbol{\mu}\|_2^2 + \frac{\lambda}{2} \|Q_{\mathbf{c}}(\mathbf{x}) - \mathbf{x}\|_1 \right\} \end{aligned} \tag{69}$$

Then,

$$\begin{aligned} \mathrm{prox}_{\eta_2 \lambda R(\mathbf{x}, \cdot)}(\boldsymbol{\mu})_j &= \operatorname*{arg\,min}_{\mathbf{c}_j \in \mathbb{R}^m} \left\{ \frac{1}{2\eta_2} (c_j - \mu_j)^2 + \frac{\lambda}{2} \sum_{i=1}^d |Q_{\mathbf{c}}(\mathbf{x})_i - x_i| \right\} \\ &= \operatorname*{arg\,min}_{\mathbf{c}_j \in \mathbb{R}^m} \left\{ \frac{1}{2\eta_2} (c_j - \mu_j)^2 + \frac{\lambda}{2} \sum_{i=1}^d \mathbb{1}(Q_{\mathbf{c}}(\mathbf{x})_i = c_j)|c_j - x_i| \right\} \end{aligned}$$

We remark that the second term of the optimization problem is hard to solve; in particular we need to know the assignments of $x_i$ to $c_j$. In the algorithm, at each time point $t$, we are given the previous epoch's assignments. We can utilize that and approximate the optimization problem by assuming $\mathbf{c}^{t+1}$ will be in a neighborhood of $\mathbf{c}^t$. We can take the gradient of $R(\mathbf{x}^{t+1}, \mathbf{c})$ at $\mathbf{c} = \mathbf{c}^t$ while finding the optimal point. This is also equivalent to optimizing the first order Taylor approximation around $\mathbf{c} = \mathbf{c}^t$. As a result we have the following optimization problem:

$$\begin{aligned} \mathrm{prox}_{\eta_2 \lambda R(\mathbf{x}^{t+1}, \cdot)}(\boldsymbol{\mu})_j \approx \operatorname*{arg\,min}_{\mathbf{c}_j \in \mathbb{R}^m} &\left\{ \frac{1}{2\eta_2} (c_j - \mu_j)^2 + \frac{\lambda}{2} \sum_{i=1}^d \mathbb{1}(Q_{\mathbf{c}^t}(\mathbf{x}^{t+1})_i = c_j^t)|c_j^t - x_i^{t+1}| \right. \\ &\left. + (c_j - c_j^t) \frac{\lambda}{2} \sum_{i=1}^d \mathbb{1}(Q_{\mathbf{c}^t}(\mathbf{x}^{t+1})_i = c_j^t) \frac{\partial |c_j^t - x_i^{t+1}|}{\partial c_j^t} \right\} \end{aligned}$$

In our implementation, we take $\frac{\partial |c_j^t - x_i^{t+1}|}{\partial c_j^t}$ as 1 if $c_j^t > x_i^{t+1}$, $-1$ if $c_j^t < x_i^{t+1}$ and 0 otherwise. Now taking the derivative with respect to $c_j$ and setting it to 0 gives us:

$$\begin{aligned} \mathrm{prox}_{\eta_2 \lambda R(\mathbf{x}^{t+1}, \cdot)}(\boldsymbol{\mu})_j \approx \mu_j - \frac{\lambda \eta_2}{2} \Big( &\sum_{i=1}^d \mathbb{1}(Q_{\mathbf{c}^t}(\mathbf{x}^{t+1})_i = c_j^t) \mathbb{1}(x_i^{t+1} > c_j^t) \\ &- \sum_{i=1}^d \mathbb{1}(Q_{\mathbf{c}^t}(\mathbf{x}^{t+1})_i = c_j^t) \mathbb{1}(x_i^{t+1} < c_j^t) \Big) \end{aligned}$$

Proximal map pulls the updated centers toward the median of the weights that are assigned to them.

**Using $P \to \infty$.** In the experiments we observed that using $P \to \infty$, i.e. using hard quantization function produces good results and also simplifies the implementation. The implications of $P \to \infty$ are as follows:

- We take $\nabla_{\mathbf{x}} f(\widetilde{Q}_{\mathbf{c}}(\mathbf{x})) = 0$ and $\nabla_{\mathbf{x}} f^{KD}(\widetilde{Q}_{\mathbf{c}}(\mathbf{x}), \mathbf{w}) = 0$.

- We take $\nabla_{\mathbf{c}} f(\widetilde{Q}_{\mathbf{c}}(\mathbf{x})) = \nabla_{\mathbf{c}} f(Q_{\mathbf{c}}(\mathbf{x})) = \begin{bmatrix} \sum_{i=1}^{d} \frac{\partial f(Q_{\mathbf{c}}(\mathbf{x}))}{\partial Q_{\mathbf{c}}(\mathbf{x})_i} \mathbb{1}(Q_{\mathbf{c}}(\mathbf{x})_i = c_1) \\ \vdots \\ \sum_{i=1}^{d} \frac{\partial f(Q_{\mathbf{c}}(\mathbf{x}))}{\partial Q_{\mathbf{c}}(\mathbf{x})_i} \mathbb{1}(Q_{\mathbf{c}}(\mathbf{x})_i = c_m) \end{bmatrix}$ and

$$\nabla_{\mathbf{c}} f^{KD}(\widetilde{Q}_{\mathbf{c}}(\mathbf{x}), \mathbf{w}) = \nabla_{\mathbf{c}} f^{KD}(Q_{\mathbf{c}}(\mathbf{x}), \mathbf{w}) = \begin{bmatrix} \sum_{i=1}^{d} \frac{\partial f^{KD}(Q_{\mathbf{c}}(\mathbf{x}), \mathbf{w})}{\partial Q_{\mathbf{c}}(\mathbf{x})_i} \mathbb{1}(Q_{\mathbf{c}}(\mathbf{x})_i = c_1) \\ \vdots \\ \sum_{i=1}^{d} \frac{\partial f^{KD}(Q_{\mathbf{c}}(\mathbf{x}), \mathbf{w})}{\partial Q_{\mathbf{c}}(\mathbf{x})_i} \mathbb{1}(Q_{\mathbf{c}}(\mathbf{x})_i = c_m) \end{bmatrix}.$$

### E.2  Implementation Details and Hyperparameters

In this section we discuss the implementation details and hyperparameters used for the algorithms considered in Section 5 of our main paper.

**Fine tuning.** In both centralized and federated settings we employ a fine tuning procedure similar to [1]. At the end of the regular training procedure, model weights are hard-quantized. After the hard-quantization, during the fine tuning epochs we let the unquantized parts of the network to continue training (e.g. batch normalization layers) and different from [1] we also continue to train quantization levels.

#### E.2.1  Centralized Setting

For centralized training, we use CIFAR-10 dataset and train a ResNet [13] model following [1] and [33]. We employ ADAM with learning rate 0.01 and no weight decay. We choose $\lambda(t) = 10^{-4}t$. For the implementation of ResNet models we used a toolbox[8]. In Table 1 we reported the results from [33] directly and implemented ProxQuant using their published code[9]. We use a learning schedule for $\eta_2$, particularly, we start with $\eta_2 = 10^{-4}$ and multiply it with 0.1 at epochs 80 and 140.

#### E.2.2  Federated Setting

For each of the methods we tuned the local step learning rate separately on the set $\{0.2, 0.15, 0.125, 0.1, 0.075, 0.05\}$. We observed that except for the two cases, for all other cases, 0.1 was the best choice for the learning rate in terms of accuracy: The two exceptions are the local training methods on FEMNIST and Per-FedAvg on CIFAR-10, for which, respectively, 0.075 and 0.125 were the best choices for the learning rate.

- QuPeD[10]: For CNN1 we choose $\lambda_p = 0.25$, $\lambda(t) = 10^{-6}t$ for 2Bits and $\lambda = 5 \times 10^{-7}t\frac{1}{0.99^t}$ for 1Bit training on CIFAR-10. On FEMNIST [11] and MNIST we choose $\lambda(t) = 5 \times 10^{-6}t$ for 2Bits and $\lambda = 10^{-6}t\frac{1}{0.99^t}$ for 1Bit training. For CNN2 we use $\lambda_p = 0.15$. Global model has the same learning schedule as the personalized models. Furthermore, we use $\eta_2 = 10^{-4}$.

  QuPeL: We used $\lambda_p = 0.2$, $\eta_3 = 0.5$ (same as pFedMe [7]) and took $\lambda$ values from QuPeD.

- Per-FedAvg [8] and pFedMe [7]:To implement Per-FedAvg, we used the same learning rate as mentioned in Section 5, schedule for main learning rate and $\alpha = 0.001$ for CNN1 and $\alpha = 2.5 \times 10^{-3}$ for CNN2 (we tuned in the interval $[8 \times 10^{-4}, 5 \times 10^{-3}]$), for the auxiliary learning rate. For pFedMe we used the same learning rate schedule for main learning rate, $K = 5$ for the number of local iterations; and we used $\lambda = 0.5$, $\eta = 0.2$ for CNN1 and $\lambda = 0.2$, $\eta = 0.15$ for CNN2 (we tuned in the interval $[0.1, 1]$ for both parameters).

- Federated Mutual Learning [29]: Since authors do not discuss the hyperparameters in the paper, we used $\alpha = \beta = 0.25$ for CNN1 and $\alpha = \beta = 0.15$ for CNN2, similar to our use of $\lambda_p$ in QuPeD. Global model has the same learning schedule as the personalized models.

---

[8]https://github.com/akamaster/pytorch_resnet_cifar10
[9]https://github.com/allenbai01/ProxQuant
[10]For federated experiments we have used Pytorch's Distributed package.
[11]We use https://github.com/tao-shen/FEMNIST_pytorch to import FEMNIST dataset.

For QuPeD and Federated ML we used CNN1 as the global model in all settings. For the other methods where global and personalized models cannot be different we used the same structure as personalized models.

### E.3  Additional Results for Federated Setting

In this section we provide additional experimental results for comparison of QuPeD with other pearsonalized learning schems from literature.

**Comparison on another CNN architecture (CNN2).** We first report experimental results on CIFAR-10 for CNN2 in Table 5 (with the same setting we have for Table 2). This is a deeper architecture than CNN1, as described in Section 5 in the main paper.

**Table 5** Test accuracy (in %) for CNN2 model at all clients, CIFAR-10.

| Method | Test Accuracy in % |
| --- | --- |
| FedAvg (FP) | $62.49 \pm 0.42$ |
| Local Training (FP) | $73.86 \pm 0.22$ |
| Local Training (2 Bits) | $73.24 \pm 0.14$ |
| Local Training (1 Bit) | $70.23 \pm 0.10$ |
| **QuPeD** (FP) | $\mathbf{76.39} \pm 0.36$ |
| **QuPeD** (2 Bits) | $75.32 \pm 0.18$ |
| **QuPeD** (1 Bit) | $72.01 \pm 0.31$ |
| PFedMe (FP) [7] | $74.70 \pm 0.10$ |
| Per-FedAvg(FP) [8] | $74.60 \pm 0.48$ |
| Federated Mutual Learning(FP) [29] | $75.74 \pm 0.56$ |

For the results in Table 5, it can be seen that the comments made for Table 2 in the main paper directly hold as QuPeD is able to outperform other schemes by a significant margin. This demonstrates that QuPeD also works for a deeper neural network (than CNN1 considered in the main paper).

**Table 6** Test accuracy (in %) for CNN1 model at all clients, with 3 classes accessed per client on CIFAR-10.

| Method | Test Accuracy (in %) |
| --- | --- |
| FedAvg (FP) | $59.23 \pm 0.25$ |
| Local Training (FP) | $78.03 \pm 0.59$ |
| Local Training (2 Bits) | $77.47 \pm 0.64$ |
| Local Training (1 Bit) | $75.89 \pm 0.66$ |
| **QuPeD** (FP) | $\mathbf{80.30} \pm 0.60$ |
| **QuPeD** (2 Bits) | $79.31 \pm 0.74$ |
| **QuPeD** (1 Bit) | $77.23 \pm 0.58$ |
| QuPeL (2 Bits) | $77.87 \pm 0.53$ |
| QuPeL (1 Bits) | $74.46 \pm 0.73$ |
| pFedMe (FP) [7] | $78.22 \pm 0.91$ |
| Per-FedAvg (FP) [8] | $75.08 \pm 0.39$ |
| Federated ML (FP) [29] | $79.44 \pm 0.82$ |

**Another Type of Data Heterogeneity.** We report results for another data heterogeneity setting where each client has access to data samples from random 3 classes on CIFAR-10. Sampling data from 3 random classes per client is a more challenging setting compared 4 classes per client considered in Section 5. In Table 6 we see that FedAvg's performance further decreased due to increased heterogeneity. Moreover, most of the other personalized FL methods are outperformed by local training whereas QuPeD still performs better than local training, and other personalized FL methods. We observe that QuPeD with 2 Bits aggressive quantization outperforms all the other competing methods except Federated ML [29] (for which it shows a similar accuracy). Moreover, QuPeD (1Bit) is able to outperform Per-FedAvg.

**Table 7** Test accuracy (in %) comparison between the cases with and without center updates for CNN1 model at all clients, 4 classes accessed per client on FEMNIST.

| Method | Test Accuracy (in %) |
|---|---|
| **QuPeD** (FP) | **97.31** $\pm$ 0.12 |
| **QuPeD** (2 Bits) | 96.73 $\pm$ 0.27 |
| **QuPeD** (1 Bit) | 95.15 $\pm$ 0.21 |
| **QuPeD** (2 Bits) no center updates | 96.48 $\pm$ 0.10 |
| **QuPeD** (1 Bit) no center updates | 91.17 $\pm$ 0.58 |

**Importance of updating the centers.** In our proposed schemes: Algorithm 2, we optimize over both the quantization levels and the model parameters. We compare performance of our proposed scheme with the case when we only optimize over model parameters and not quantization levels in Table 7. As seen from the results in the table, having the center updates in the optimization problem is critical, particularly, for the 1Bit quantization case for which we observe an increase in the performance by 4%.

**Results on MNIST.** We now provide additional results on MNIST dataset to compared QuPeD with other competing schemes. We consider 50 clients in total, where each client samples data from 3 or 4 random classes and uses CNN1. We train for a total of 50 epochs, for quantized training we allocate the last 7 epochs for finetuning.

**Table 8** Test accuracy (in %) for CNN1 model at all clients, on MNIST.

| Method | 3 classes per client | 4 classes per client |
|---|---|---|
| FedAvg (FP) | 98.64 $\pm$ 0.10 | 98.65 $\pm$ 0.09 |
| Local Training (FP) | 98.79 $\pm$ 0.03 | 98.66 $\pm$ 0.15 |
| Local Training (2 Bits) | 98.53 $\pm$ 0.07 | 98.37 $\pm$ 0.11 |
| Local Training (1 Bit) | 98.41 $\pm$ 0.02 | 97.95 $\pm$ 0.20 |
| **QuPeD** (FP) | **99.05** $\pm$ 0.10 | 98.89 $\pm$ 0.11 |
| **QuPeD** (2 Bits) | 98.96 $\pm$ 0.13 | 98.67 $\pm$ 0.18 |
| **QuPeD** (1 Bit) | 98.57 $\pm$ 0.08 | 98.25 $\pm$ 0.16 |
| QuPeL (2 Bits) | 98.95 $\pm$ 0.12 | 98.61 $\pm$ 0.19 |
| QuPeL (1 Bits) | 98.33 $\pm$ 0.14 | 98.11 $\pm$ 0.26 |
| pFedMe (FP) [7] | 98.98 $\pm$ 0.05 | 98.82 $\pm$ 0.15 |
| Per-FedAvg (FP) [8] | 98.82 $\pm$ 0.05 | **98.93** $\pm$ 0.09 |
| Federated ML (FP) [29] | 99.00 $\pm$ 0.06 | 98.84 $\pm$ 0.13 |

QuPeD (FP) outperforms all methods except Per-FedAvg on MNIST when clients sample data from 4 random classes. The difference is almost negligible (0.04%). As we can observe in Table 8 with the increased heterogeneity QuPeD starts to outperform Per-FedAvg by a 0.20% margin. Moreover, we observe QuPeD with 2Bit quantization also outperforms Per-FedAvg.

**Text classification task on AG News Dataset.** To show that our method can also be applied for tasks different than vision tasks. text classification problem using the AG News dataset (available at https://pytorch.org/text/stable/datasets.html). We used half of the dataset to make the training procedure more challenging. We used EmbeddingBag structure available at https://pytorch.org/tutorials/beginner/text_sentiment_ngrams_tutorial.html and distributed the data such that each of the 42 clients has access to samples from 3 out of 4 classes. The results we obtained are provided in Table 9

These results demonstrate the effectiveness of QuPeD on text data in comparison with local training.

### E.4 Computational Resources

For our experiments, we used a server which has 6 Nvidia RTX2080Ti GPU's and Intel Xeon Gold 6230 CPU @ 2.10GHz CPU's. The longest epoch time is 125 seconds with QuPeD (2Bit) training on CIFAR-10. Our code uses a maximum of 9GB memory per GPU.

**Table 9** Test accuracy (in %) for Embedding Bag model at all clients, on AG News.

| Method | |
|---|---|
| FedAvg (FP) | $83.04 \pm 0.60$ |
| Local training (FP) | $84.20 \pm 1.53$ |
| Local training (2 Bits) | $82.68 \pm 0.34$ |
| Local training (1 Bit) | $82.12 \pm 2.17$ |
| QuPeD (FP) | $\mathbf{85.06} \pm 1.07$ |
| QuPeD (2 Bits) | $83.62 \pm 0.50$ |
| QuPeD (1 Bit) | $82.72 \pm 1.20$ |