# OpenReview forum: "QuPeD: Quantized Personalization via Distillation with Applications to Federated Learning"
_NeurIPS.cc/2021/Conference — NeurIPS 2021 Poster_

### Official Review · Reviewer_hK95 · 2021-07-06

**Rating:** 6
**Confidence:** 4

**Summary:**

This paper proposes a novel optimization objective to address the heterogeneity issue in the setting of federated learning. The heterogeneity comes from the fact that different local clients may have different data distribution and resource constraints. The proposed method allows every client to learn a quantized model with possibly different sizes. Locally, each client optimizes for a loss function containing two major terms: 1) loss of the local quantized model; 2) the distance between the quantized model and the global model.

The authors propose an alternating proximal algorithm, and provide convergence analysis under standard assumptions. The authors then empirically show that the proposed algorithm outperforms related works on CIFAR-10 and FEMNIST.

**Limitations And Societal Impact:**

Yes, the authors pointed out the potential societal impact in Section 6.

**Main Review:**

Strength:
- The paper is logically well structured. The design idea behind the proposed loss function is clearly stated in Section 2. All the theoretical results are followed by remarks that explain the implications, which greatly improves the readability of this paper.
- The idea of allowing every client to learn a personalized quantized model of different size seems new in the federated learning literature.
- The paper provides convergence analysis of the proposed algorithm.

Weakness:
- This paper is motivated by the cross-device federated learning (as stated Section 1), but the proposed algorithm may be difficult to use in the cross-device setting. Federated learning has two settings (see Table 1 in [1]):
   - Cross-device: in this setting, a client can be a mobile device (e.g., smartphones), and there can be millions of clients in total, only a few of them (say hundred) participate in each training round, and each client usually participates *at most once* in the entire training session.
    - Cross-silo: in this setting, a client can be an organization (e.g., banks or hospitals), and there can be hundreds of clients in total, and it is ok to assume that all clients are available and participate in every round.

The proposed algorithm (Algorithm 2 in the paper) requires that all clients participate in every training round, and that every client maintains a local model and is responsible to update and carry it from this round to the next round. Furthermore, the proposed algorithm does not support inference on new clients (i.e., clients that do not appear in the training). These constraints make it difficult to use the proposed algorithm in the real-world cross-device federated learning applications.

- The paper does not compare with a natural baseline algorithm FedAvg + fine-tuning [2], i.e., each client fine-tunes the global model (learned by FedAvg) locally to create a personalized model. This simple algorithm can be used in both cross-device and cross-silo settings. If needed, one can add an extra quantization step after local fine-tuning to create a quantized personalized model.

- The hyperparameters are not properly tuned for the baseline algorithms compared in Section 5. For example, in Section 5, the authors said that “For all methods, if applicable, we set τ = 10 local iterations, initialize learning rate of 0.1...”. It is unclear why the authors used the same learning rate for all baseline algorithms, and it would be more convincing if the authors can tune the learning rates (and also epsilons if adaptive optimizer like Adam is used) for different algorithms.

[1] Kairouz et al., “Advances and Open Problems in Federated Learning”, arXiv: 1912.04977, 2019.

[2] Jiang et al., “Improving Federated Learning Personalization via Model Agnostic Meta Learning”, arXiv: 1909.12488, 2019.

**Time Spent Reviewing:**

5

---

> ### Author Response · Authors · 2021-08-10
> **Client sampling, comparison to FedAvg + fine tuning, tuning hyperparameters**
>
>  We thank the reviewer for their time and comments. We will address the concerns one by one:
>
>  * **Partial client participation.** We did not consider partial client participation in the main paper because we wanted to focus on our novel problem formulation combining model compression and personalization along-with its convergence analysis. Incorporating partial client participation and analyzing the resulting algorithm is fairly simple. We have extended our analysis incorporating this, which essentially only changes Lemma 1 and Corollary 1, as everything before that is for local updates only. Now we give a summary of what changes:
>
> 	Let $\mathcal{K}_t$ denote the set of clients that participates at time $t$, where $|\mathcal{K}_t|=K$, i.e., $K$ clients participate in the training process at any time. In this case, we define the average parameter $w^t$ and the gradient $g^t$ as the average over the respective parameters of only the active clients at time $t$; we also define $\gamma_t$ similarly (see line 760).
>
> 	**Change in the proof of Lemma 1:** In the proof of Lemma 1, the second term on the RHS of the second inequality, with the above modification will be equal to $\||\frac{1}{K} \sum_\{k \in \mathcal{K_t} \} \nabla_\{\mathbf{w}^j}F_k(\mathbf{x_k}^\{j+1\},\mathbf{c_k}^\{j+1\},\mathbf{w}^\{j\}) - \nabla_\{\mathbf{w}^j}F_i(\mathbf{x_i}^\{j+1\},\mathbf{c_i}^\{j+1\},\mathbf{w}^{j})\||^2$. Earlier, the average was over all clients from $1$ to $n$ and this term was bounded by $\kappa_i$ using Assumption A.6. Now, we can use the Jensen's inequality (iteratively) and Assumption A.6 and bound this by $2\kappa_i+\frac{2}{K}\sum_{j\in\mathcal{K}_t}\kappa_j$. This change will propagate over til the end. That's it.
>
> 	**Change in the proof of Corollary 1:** Since this is a corollary to Lemma 1, this will also see a similar change.
>
> 	**Remaining convergence proof:** Now, continuing the exact same convergence proof in lines 743-756 and using the modified bounds of Lemma 1 and Corollary 1 will give the bound of our algorithm with partial client participation. We have essentially given the modification in the entire proof here.
>
> 	**New experiments with partial client participation:** We also ran experiments on the FEMNIST dataset with partial client participation. We fixed the ratio of participating clients to be $1/3$'rd in each round and sampled $n/3$ clients accordingly uniformly at random. With the same setup in the main paper we obtained the following results:
>
> 	FedAvg (FP):  $91.30\\% \pm 0.43$
>
> 	QuPeD (FP):  $94.93 \\%\pm 0.25$
>
> 	QuPeD (2 Bits): $ 94.56 \\% \pm 0.18$
>
> 	QuPeD (1 Bit):  $ 92.52 \\% \pm 0.64$
>
> 	Per-FedAvg (FP):  $92.10 \\% \pm 0.22$
>
> 	pFedMe (FP):  $ 93.70 \\% \pm 0.39$
>
> 	It is clear that even with partial client participation, QuPeD outperforms the competing methods even when aggressive compression is employed. We will include these results regarding client sampling in the next revision.
>
> 	**Supporting new clients:** Our algorithm can be extended to support new clients; one way would be to use the global model as an initialization for the new clients, then those clients could do local updates as in FedAvg + fine tuning algorithm. As shown in the centralized experiments our quantization method can handle a pre-trained initial model.
> 	With these answers we hope we demonstrate that our method is suitable for a cross-device setting. If there are unclear points we can do further clarifications.
>
> * **FedAve+fine-tuning:** Following your suggestion we implemented FedAvg + fine tuning algorithm. On FEMNIST dataset with the setting in Table 3 in main paper FedAvg + finetuning obtains $96.72\\% \pm 0.10$. Here, after training on FedAvg we let the clients do 10 local iterations on their local data, we also tried letting them do a full epoch but that resulted in a worse performance.
>
> * **Hyperparameter tuning:**  We followed the standard practice for hyperparameter tuning and chose local learning rate schedule the same across clients; for example, see [1,2,3], etc. In fact, we use the same model and learning rate that is used in [1] where the authors use the same local learning rate schedule for all the competing methods; we did not tune the learning rate based on our own method. Consequently, we believe the experimental results we provide are fair.
>
> We hope to have answered your questions and concerns adequately; if not, we are happy to clarify further.
>
> **References**
>
> [1] Zhang, Michael, et al. "Personalized Federated Learning with First Order Model Optimization." International Conference on Learning Representations. 2020.
>
> [2] Fallah, Alireza, Aryan Mokhtari, and Asuman Ozdaglar. "Personalized federated learning: A meta-learning approach." Advances in Neural Information Processing Systems 33. 2020.
>
> [3] Liang, Paul Pu, et al. "Think locally, act globally: Federated learning with local and global representations." arXiv preprint arXiv:2001.01523 (2020).

---

> > ### Comment · Reviewer_hK95 · 2021-08-13
> > **Small-scale and large-scale settings, and hyperparameter tuning**
> >
> > Thanks for the additional analysis and experiments on FEMNIST with partial client participation. However, this does not fully convince me that the provided algorithm would perform well in the real cross-device federated learning setting. The additional experiments on FEMNIST assumes that 1/3 of clients can participate in each round, but this ratio is way too high compared to what happens in a real cross-device application (only a few thousands participate in each round while the whole population may be hundreds of millions [1]). The proposed algorithm requires the clients to maintain a local state (i.e., a personalized model) and keeps updating it from round to round. In the cross-device setting, this local state will soon become stale because a client usually only participates a very small number of times during the entire training process. As a result, stateful algorithms often result in performance degradation in the cross-device setting (see the SCAFFOLD discussion in [2] Section 5.1). Overall the proposed algorithm is novel and seems to work well for small-scale cross-silo settings, although the motivation on learning a quantized model may be weakened in that setting. I highly recommend the authors to add a discussion on the large-scale and small-scale settings and potential limitations in future versions of this paper.
> >
> > My concern on hyperparameters tuning also remains. Directly using (i.e., without tuning) the same hyperparameters like the learning rates when comparing all algorithms does not seem to be a fair comparison to me. Performance of the same algorithm can vary a lot for different optimizers or learning rates (see, e.g., Section 5.2 in [2]). In the rebuttal, the authors mentioned [3], but [3] does not conduct experiments on FEMNIST, and does not seem to tune the learning rates as well.
> >
> > I appreciate the overall algorithmic and theoretical contributions of this paper, but I still decide to keep my current score given the reasons above.
> >
> > [1] Kairouz et al., “Advances and Open Problems in Federated Learning”, arXiv: 1912.04977, 2019.
> >
> > [2] Reddi et al., “Adaptive Federated Optimization”, arxiv: 2003.00295, 2020.
> >
> > [3] Zhang, Michael, et al. "Personalized Federated Learning with First Order Model Optimization." International Conference on Learning Representations. 2020.

---

> > > ### Author Response · Authors · 2021-08-19
> > > **Results with hyperparameter tuning, results with lower client sampling ratio and discussion on experimental setting**
> > >
> > >  We appreciate your comment acknowledging novelty of our work in both theory and algorithms.
> > >
> > > **On hyperparameter tuning:** Thank you for your feedback. Following your suggestion, we tuned the learning rates independently for each method that we compared our algorithm against, both on CIFAR-10 and FEMNIST (in the same setting of Table 3 in the paper). In particular, we tuned the learning rate of each method in the set $\\{0.2, 0.15, 0.125, 0.1, 0.075, 0.05\\}$. We observed that except for the two cases, for all other cases, 0.1 was the best choice for the learning rate in terms of accuracy: The two exceptions are the local training methods on FEMNIST and Per-FedAvg on CIFAR-10, for which, respectively, 0.075 and 0.125 were the best choices for the learning rate. These updates do not change our conclusions in the main paper. As can be seen in the results below, our method still outperforms all the methods with the modified hyperparameters as well. Accordingly, this will only change (compared to the reported numbers in Table 3 in the paper) the performance of Local Training on FEMNIST and Per-FedAvg on CIFAR-10. For convenience, we report the accuracy obtained with the fine-tuned learning rates in the following table, which only differs from Table 3 (in the paper) in the three numbers on the FEMNIST column corresponding to Local Training (FP/2Bits/1Bit) and one number on CIFAR-10 column corresponding to Per-FedAvg (FP).
> > >
> > > | Method | FEMNIST | CIFAR-10 |
> > > | --- | ----------- |----------- |
> > > | FedAvg (FP) | $94.92 \pm 0.04 $ | $61.40 \pm 0.29$ |
> > > | Local Training (FP)  | $94.86 \pm 0.55 $| $71.57 \pm 0.28 $ |
> > > | Local Training (2 Bits)   | $93.95 \pm 0.16 $| $70.87 \pm 0.15 $ |
> > > | Local Training (1 Bit)   |$93.00 \pm 0.29 $| $69.05 \pm 0.13 $ |
> > > | QuPeD (FP)  |$\mathbf{97.31} \pm 0.12 $| $\mathbf{75.06} \pm 0.40 $ |
> > > |QuPeD (2 Bits)  |$96.73 \pm 0.27 $| $74.58 \pm 0.44$ |
> > > | QuPeD (1 Bit) |$95.15 \pm 0.21 $  | $71.20 \pm 0.33$ |
> > > | QuPeL (2 Bits)  | $96.10 \pm 0.14 $| $73.52 \pm 0.51$|
> > > | QuPeL (1 Bits)  | $94.06 \pm 0.28 $| $71.01 \pm 0.32$ |
> > > | pFedMe (FP)  | $96.60 \pm 0.37 $| $73.66 \pm 0.65$ |
> > > | Per-FedAvg (FP)  |$97.16 \pm 0.21 $| $74.15 \pm 0.62$ |
> > > | Federated ML (FP) | $96.32 \pm 0.32 $ | $74.34 \pm 0.30$ |
> > >
> > >
> > > We hope that we have demonstrated that our method outperforms other methods in a fair way. We believe that our numerical evaluations are both reasonably extensive and fair, and demonstrate the advantages of our approach. Note that if one compares full precision performance, our approach outperforms the reported methods; and even with compression, in many cases, it performs better than the full precision performance of reported methods. We therefore believe that ours is a promising approach to compressed personalized learning with different model sizes.
> > >
> > > **Performance in cross device setting:** Following your feedback, we did additional experiments for training CNN1 on MNIST with 50 clients, where each client has data from 3 random classes. This time we chose the client sampling ratio to be 0.1. For these experiments also we fine-tuned the learning rates independently for individual methods and obtained the following results (which are averaged over 2 runs):
> > >
> > >
> > > FedAvg (FP): $92.87\\% \pm 0.05 $
> > >
> > > FedAvg + tuning (FP): $95.02\\% \pm 0.07$
> > >
> > > QuPeD (FP): $98.17\\% \pm 0.32$
> > >
> > > QuPeD (2 Bits): $98.01\\% \pm 0.15$
> > >
> > > QuPeD (1 Bit): $97.58\\% \pm 0.23$
> > >
> > > Per-FedAvg (FP):$95.80\\% \pm 0.29$
> > >
> > > pFedMe (FP): $97.79\\% \pm 0.03$
> > >
> > > We observe that under 0.1 client sampling ratio our method still outperforms others. Note that even QuPeD (1 Bit) is quite competitive compared to other methods. This indicates robustness of our compression method to client sampling.
> > >
> > > In our experimental setting, we tried to follow the previous personalized FL works. For instance, [1] has 50 clients with 0.2 sampling ratio, [2] has 20 clients with 0.25 sampling ratio, and [3] has two settings: 15 clients with full sampling ratio and 100 clients with 0.1 sampling ratio. We can try to do the experiments with up to 500/1000 clients and a smaller client sampling ratio, if needed. We can certainly add a discussion on the large-scale and small-scale settings and potential limitations of our work in future versions. However, a production level system with millions of nodes are out of scope of this submission, where our focus is on a new personalized learning scheme with model compression, with new theory and algorithms.
> > >
> > > For your comment *The additional experiments on FEMNIST assumes that 1/3 of clients can participate in each round, but this ratio is way too high compared to what happens in a real cross-device application (only a few thousands participate in each round while the whole population may be hundreds of millions [1]).* As mentioned earlier we believe simulating the full-fledged FL production setup which involves 100s of millions of clients (where only thousands are sampled in each round) is perhaps infeasible to check the efficacy of our method (or of any method for that matter that people propose) due to obvious reasons.
> > >
> > > We would like to emphasize that FL has a diverse set of challenges and a significant amount of effort is going on to address them. Among these challenges, data and resource heterogeneities are among the major ones, and we address them simultaneously (with personalization) by first formulating a non-trivial optimization problem, proposing a method (that we call QuPeD) to solve that, and analyzing its convergence guarantees, along with reasonable numerical experiments demonstrating the superiority of our methods against all the existing methods that do not adequately address those challenges. Note that the resource heterogeneity (where clients learn models with different dimension and different precision) that we consider has not been studied before in literature, as you have also observed; for quantization, unlike before, clients also optimize over the quantization values for compression. This is a challenging problem and, in our opinion, we did a significant amount of technical work in formulating it as an optimization problem and then thoroughly analyzing the convergence of our proposed algorithm. With all that, we even recovered the convergence rate of existing works (as special cases). We strongly believe that the technical content of this paper from the problem formulation to the analysis is highly non-trivial, and should not be diminished. We also believe that we have addressed the client sampling issue raised through additional experiments, within the scale of most other reported works, and therefore show its advantages even in these scenarios.
> > >
> > > As in any research,  our goal is to demonstrate new ideas, algorithms, theory and evaluate its performance at reasonable scale; as mentioned earlier, FL production level scale is infeasible in such works where the focus is more on new ideas/theory/algorithms. However, we believe that it can serve as a basis for a new direction of research in this area, as our formulation and the analysis are new and non-trivial, and, most of all, we believe that it addresses very challenging and perhaps some of the central problems of federated learning.
> > >
> > > We sincerely hope you consider our response and efforts in addressing your concerns and would greatly appreciate rethinking your score based on our responses. Please let us know if you have any more concerns.
> > >
> > > **References**
> > >
> > > [1] Fallah et al. "Personalized federated learning: A meta-learning approach." Advances in Neural Information Processing Systems 33. 2020.
> > >
> > > [2] Dinh et al. "Personalized Federated Learning with Moreau Envelopes." Advances in Neural Information Processing Systems 33. 2020.
> > >
> > > [3] Zhang, Michael, et al. "Personalized Federated Learning with First Order Model Optimization." International Conference on Learning Representations. 2020.

---

> > > > ### Comment · Reviewer_hK95 · 2021-08-21
> > > > **Adding discussions on limitations is highly recommended**
> > > >
> > > > Thanks again for the additional experimental results with learning rate tuning. Adding those results (including the FedAvg + fine-tuning results in the previous response) to the paper will definitely make the performance evaluation part more convincing.
> > > >
> > > > Thanks for the new results at a lower sampling ratio 0.1 when the total number of clients are 50, I want to point out again that this is still far from what’s happening in a practical cross-device FL setting (and as mentioned in my previous comment, I am worried that stateful algorithms may not perform well in this setting). And by that, I didn't mean to ask authors to simulate “the full-fledged FL production setup which involves 100s of millions of clients”, instead, I was trying to recommend authors to run simulations at a larger-scale setting, e.g., a few thousands of clients in total, and tens of clients in each round (see, e.g., datasets used in “Adaptive Federated Optimization” paper).
> > > >
> > > > While I appreciate all the comments centered around the algorithmic and theoretical contributions of this paper, I would like to emphasize that bridging the gap between FL research and practice is also important, especially given that this paper aims at addressing a real practical problem. Hence, I still recommend authors to add a discussion on the large-scale vs small-scale settings and potential limitations in the paper.
> > > >
> > > > Given all the considerations above, I decided to increase my score to 6.

---

> > > > > ### Author Response · Authors · 2021-08-21
> > > > > **Thank you for increasing the rating**
> > > > >
> > > > > We appreciate that the reviewer has carefully gone over our rebuttals and decided to increase their rating. We will include the additional new results in the final version of the paper (if accepted). Specifically, we will make the following changes:
> > > > > * Create a subsection in Appendix E regarding the new experimental results with client sampling.
> > > > > * Create a subsection in Appendix C where we extend the proof of Theorem 2 to include client sampling.
> > > > > *Add a discussion on the large-scale versus small-scale settings for the experiments along with potential limitations of our paper.
> > > > > * Update the results with individually tuned hyperparameters.

---

### Official Review · Reviewer_g4tj · 2021-07-12

**Rating:** 6
**Confidence:** 4

**Summary:**

This paper studies a quantized and personalized algorithm for faster inference and adapting to data/system heterogeneity in federated learning.
Personalization is facilitated by knowledge distillation. Model compression is achieved by end-to-end training to optimize for quantized models and quantization parameters.


**Limitations And Societal Impact:**

The authors have discussed the limitations of their work, and I don't think there is potential negative societal impact of this work.

**Main Review:**


Pros:

* The problem of jointly consider personalization and compression is interesting.

* In terms of quantization alone, the formulation (6) is novel to me (to the best of my knowledge), and demonstrates improvements compared with (4) or (5), although it is a direct combination of existing works.

* The observation from Theorem 2 that clients with more representative data can tolerate more aggressive compression is interesting.

* Experiments clearly show the effectiveness of both the quantization method, and the main contribution QuPeD. QuPeD also outperforms existing personalization methods, while allowing for heterogeneous model architecture and precision.

Concerns:


* (a) One bottleneck of federated learning is the computation and memory constraints of on-device training. The proposed method only reduces inference time, but adds much training overhead, which is very expensive. Other directions of compression methods (beyond teacher-student knowledge distillation) that can reduce training cost (computation or memory) while producing a smaller model could be more desirable for federated learning, for example, training factorized layers. (b) If only considering inference, there are other natural and simpler baselines, such as first performing regular personalization training (via KD or other techniques) and then pruning the models as a post-processing step. How does this work empirically?

* The convergence analyses do not support partial client participation.

* I think the experiments are lacking results on text data. As the gradient distribution of image data and text/language data are very usually different., it not clear if the proposed method could still yield improvements on the new type of data, and how much.

* Both the quantization method and the personalization method (KD) have been studied by previous literature, and the proposed optimization objective (2) combines the two ideas. I think discussing how potentially these two parts interact with each other (i.e., how to design better personalization methods that are inherently suitable for quantization) would be more interesting.

My major concerns are the first two points.


## update
I have updated my score (see detailed comments in my response below).

**Time Spent Reviewing:**

3 hours

---

> ### Author Response · Authors · 2021-08-10
> **Explanation on training overhead, introducing partial participation, results on text data, coupling of quantizaiton and personalization.**
>
> We thank the reviewer for their comments and time. We will address the concerns one by one:
> * (a) Our method does not add much overhead in training: compared to training full precision personalized FL methods, we only need to add a quantization codebook which has negligible effect on memory. Note that we do not employ a classical teacher-student knowledge distillation for compression via distillation. In our formulation, the global model acts like a teacher network and personalized models act like student networks for personalization but not quantization. These two types of networks (personalized and global) are of similar structure (global model is not a very deep teacher network), and exist in the previous literature for personalized FL. Assuming the availability of more resources for training and not for inference time is in fact motivated by FL with edge devices (*e.g.*, smartphones), where model training is typically done when the phone is charging and is idle (so more power/memory is available), whereas, inference is more frequent and is done when phones are actively used and have other functions. Since relatively less power/memory is available during inference time, the deployed model should respect stricter resource constraints. This is actually how the Google assistant improves itself; see the following reference: https://support.google.com/assistant/answer/10176224?hl=en#zippy=%2Chow-google-assistant-improves-with-federated-learning.
> Of course, one can alternatively employ other compression methods, and we hope that our paper inspires future works with different compression and personalization approaches. However, as far as we are aware, there is not a paper that formalizes an optimization problem and provides theoretical convergence guarantees for methods that have compressed training; these all are integral parts of our method.
>
> 	(b) Compression (pruning, quantization, etc.) as a post-processing step performs poorly empirically compared to quantization aware training [1]. For example, as far as we are aware there are not post-processing quantization schemes that allow 1 or 2 bit quantization without incurring significant accuracy loss (see [2] for a 4Bit post-processing quantization work and see [3] for the empirical performance of a recent post-processing quantization work). Thus, it is critical to embed the compression process into training and optimize over the quantization parameters also, as we do in our paper.
>
>  * **Partial client participation.** We did not consider partial client participation in the main paper because we wanted to focus on our novel problem formulation combining model compression and personalization along-with its convergence analysis. Incorporating partial client participation and analyzing the resulting algorithm is fairly simple. We have extended our analysis incorporating this, which essentially only changes Lemma 1 and Corollary 1, as everything before that is for local updates only. Now we give a summary of what changes:
>
> 	Let $\mathcal{K}_t$ denote the set of clients that participates at time $t$, where $|\mathcal{K}_t|=K$, i.e., $K$ clients participate in the training process at any time. In this case, we define the average parameter $w^t$ and the gradient $g^t$ as the average over the respective parameters of only the active clients at time $t$; we also define $\gamma_t$ similarly (see line 760).
>
> 	**Change in the proof of Lemma 1:** In the proof of Lemma 1, the second term on the RHS of the second inequality, with the above modification will be equal to $\||\frac{1}{K} \sum_\{k \in \mathcal{K_t} \} \nabla_\{\mathbf{w}^j}F_k(\mathbf{x_k}^\{j+1\},\mathbf{c_k}^\{j+1\},\mathbf{w}^\{j\}) - \nabla_\{\mathbf{w}^j}F_i(\mathbf{x_i}^\{j+1\},\mathbf{c_i}^\{j+1\},\mathbf{w}^{j})\||^2$. Earlier, the average was over all clients from $1$ to $n$ and this term was bounded by $\kappa_i$ using Assumption A.6. Now, we can use the Jensen's inequality (iteratively) and Assumption A.6 and bound this by $2\kappa_i+\frac{2}{K}\sum_{j\in\mathcal{K}_t}\kappa_j$. This change will propagate over til the end. That's it.
>
> 	**Change in the proof of Corollary 1:** Since this is a corollary to Lemma 1, this will also see a similar change.
>
> 	**Remaining convergence proof:** Now, continuing the exact same convergence proof in lines 743-756 and using the modified bounds of Lemma 1 and Corollary 1 will give the bound of our algorithm with partial client participation. We have essentially given the modification in the entire proof here.
>
> 	**New experiments with partial client participation:** We also ran experiments on the FEMNIST dataset with partial client participation. We fixed the ratio of participating clients to be $1/3$'rd in each round and sampled $n/3$ clients accordingly uniformly at random. With the same setup in the main paper we obtained the following results:
>
> 	FedAvg (FP):  $91.30\\% \pm 0.43$
>
> 	QuPeD (FP):  $94.93 \\%\pm 0.25$
>
> 	QuPeD (2 Bits): $ 94.56 \\% \pm 0.18$
>
> 	QuPeD (1 Bit):  $ 92.52 \\% \pm 0.64$
>
> 	Per-FedAvg (FP):  $92.10 \\% \pm 0.22$
>
> 	pFedMe (FP):  $ 93.70 \\% \pm 0.39$
>
> 	It is clear that even with partial client participation, QuPeD outperforms the competing methods even when aggressive compression is employed. We will include these results regarding client sampling in the next revision.
>
> * **New experiments with text data:** Thank you for your suggestion. We tested our algorithm on a text classification problem using AG News dataset (available at https://pytorch.org/text/stable/datasets.html). We used half of the dataset to make the training procedure more challenging. We used EmbeddingBag structure available at https://pytorch.org/tutorials/beginner/text_sentiment_ngrams_tutorial.html and distributed the data such that each of the 42 clients has access to samples from 3 out of 4 classes. We obtained the following results:
>
> 	FedAvg (FP):  $83.04\\% \pm 0.60$
>
> 	Local training (FP):  $84.20\\% \pm 1.53$
>
> 	Local training (2 Bits):  $82.68\\% \pm 0.34$
>
> 	Local training (1 Bit):  $82.12\\%\pm 2.17$
>
> 	QuPeD (FP):  $85.06\\% \pm 1.07$
>
> 	QuPeD (2 Bits):  $83.62\\% \pm 0.50$
>
> 	QuPeD (1 Bit):  $82.72\\%\pm 1.20$
>
> 	These results (which we will put in the revision) demonstrate the effectiveness of QuPeD on text data.
>
> * **Quantization and personalization:** Our quantization and personalization schemes are novel. One idea that is new in our quantization scheme is that we formally optimize over quantization levels, which has not been done before in previous literature (except for heuristic updates) and has proven to be crucial for improved empirical performance. Moreover, although KD is used in personalized FL, we formalize the optimzation problem and analyze it; and our method, for the first time, uses reverse KL divergence updates to update the global model.
>
> 	**Interaction of compression and personalization:**  QuPeD couples quantization and personalization together. Quantization of personalized models depends on the data that clients have; even with same number of bits, quantization levels will have different values and optimized with respect to the local data. As a result, quantization in each client is tailored according to their own data and resources.
>
> 	One of the key points of our work is the flexibility that it provides; our method allows different architecture, level of precision, and quantization levels for each individual clients. We envisage subsequent works to adhere with these freedom, particularly because clients may be vastly heterogeneous (in terms of data and resources) in FL. Thinking about the interaction of clients with potentially different structures naturally yields KD as a personalization method, we cannot think of any alternatives but this might be an interesting future direction.
>
> We hope to have answered your questions and concerns adequately; if not, we are happy to clarify further.
>
> **References**
>
> [1] Gholami et al. "A survey of quantization methods for efficient neural network inference." arXiv preprint arXiv:2103.13630 (2021).
>
> [2] Banner  et al. "Post training 4-bit quantization of convolutional networks for rapid-deployment." Proceedings of the 33rd International Conference on Neural Information Processing Systems. 2019.
>
> [3] Li et al. "BRECQ: Pushing the Limit of Post-Training Quantization by Block Reconstruction." International Conference on Learning Representations. 2020.

---

> > ### Comment · Reviewer_g4tj · 2021-08-12
> > **Updates**
> >
> > Thanks for the response.
> >
> > My original concern of the additional overhead of the algorithm still remains. I understand that compression is not achieved by KD (but rather, by a different formulation that involves jointly optimizing for quantization levels). However, in terms of the training cost, the proposed compression + personalization technique is more expensive than only performing personalization, which limits the benefits of compression to inference only.
> >
> > Having said this, I appreciate the overall contributions of this work, as well as the new results regarding the theorems and experiments. Therefore, I will raise my score to 6.

---

> > > ### Author Response · Authors · 2021-08-21
> > > **Thank you for increasing the rating**
> > >
> > > We thank the reviewer for their reconsideration and for increasing their rating. In the final version of our paper (if accepted), we will discuss the effect of quantization on the training cost and include new results on client sampling as we mentioned in our previous response.

---

### Official Review · Reviewer_JdrY · 2021-07-16

**Rating:** 6
**Confidence:** 3

**Summary:**

This paper proposes a QuPeD framework personalized FL via knowledge distillation (KD) to alleviate both data and resource heterogeneity. The authors first present a relaxed formulation and an algorithm under the centralized setting. Next, they extend the previous formulation to FL setting to enable different model dimensions and precision requirements for clients. The authors further provide detailed convergence arguments and empirically evaluate the proposed methods on practical datasets.

**Limitations And Societal Impact:**

Yes, the authors have explicitly addressed the limitations (appendix) and potential negative societal impact (last section in the paper).

**Main Review:**

Merits:
-The paper considers both data and resource heterogeneity, which is a fundamental issue in FL. The paper is well written and organized.  Algorithms and their corresponding convergence results are clearly interpreted. Empirical evidence is sufficient and the results seem promising.

-The use of KD is natural (but not new) as each client may hold models with different dimensions and precision. To formalize the use of KD as an optimization problem is original and important as it facilities the theoretical characterizations.

-The implications of the coupled effect of data and resource heterogeneity on the convergence rate from Theorem.2 are interesting.

Weakness/potential improvements
-The client's loss function (i.e., Eq.2) seems contrived since it simply combines a compressed objective and an integrated KD loss. Is there a principled way for each client to choose $\lambda$ and $\lambda_p$?

-Further evaluations on more challenging tasks e.g. Stack Overflow are needed.


**Time Spent Reviewing:**

6

---

> ### Author Response · Authors · 2021-08-10
> **General principles for choosing hyperparamters and results for text classification.**
>
> We thank the reviewer for their comments and time.
>
> * Yes, there are general principles that can be followed. For $\lambda_p$, the choice heavily depends on data distribution; for instance, when data is distributed extremely heterogeneously (such as 2 classes per client), it might be better to choose a small $\lambda_p$ and rely on local updates rather than global model (which is expected to perform poorly). For $\lambda$, the choice mostly depends on the model architecture: if you have a model with layers consisting of vast number of parameters, then it might be better to choose a small $\lambda$, so the weight updates are not only affected by the position of weights but are also affected by the gradients. It might seem contrived but we created this optimization problem after empirical observations and theoretical considerations. For instance, we kept the global model as a teacher model and used reverse KL divergence updates to train it. This resulted in good empirical results and compatibility with our alternating update scheme.
>
> * We did not have sufficient time to train on the Stack Overflow dataset, however, we tested our algorithm on another text classification problem using the AG News dataset (available at https://pytorch.org/text/stable/datasets.html). We used half of the dataset to make the training procedure more challenging. We used EmbeddingBag structure available at https://pytorch.org/tutorials/beginner/text_sentiment_ngrams_tutorial.html and distributed the data such that each of the 42 clients has access to samples from 3 out of 4 classes. We obtained the following results:
>
>
> 	FedAvg (FP):  $83.04\\% \pm 0.60$
>
> 	Local training (FP):  $84.20\\% \pm 1.53$
>
> 	Local training (2 Bits):  $82.68\\% \pm 0.34$
>
> 	Local training (1 Bit):  $82.12\\%\pm 2.17$
>
> 	QuPeD (FP):  $85.06\\% \pm 1.07$
>
> 	QuPeD (2 Bits):  $83.62\\% \pm 0.50$
>
> 	QuPeD (1 Bit):  $82.72\\%\pm 1.20$
>
> 	These results (which we will put in the revision) clearly demonstrate the effectiveness of QuPeD on text data. Though we expect to see a similar trend on the Stack Overflow dataset, if required, we can also do experiments and include those results in the final version if the paper is accepted.
>
> We hope to have answered your questions and concerns adequately; if not, we are happy to clarify further.

---

> > ### Comment · Reviewer_JdrY · 2021-08-24
> > **Addressing my concerns; additional experiments**
> >
> > I appreciate the authors' explanations of my concerns and additional experiments. After reading other reviewers' comments, I decide to keep my positive score.
> >
> >
> > .

---

### Official Review · Reviewer_UdzD · 2021-07-16

**Rating:** 7
**Confidence:** 4

**Summary:**

The paper addresses the important problem of quantized personalization in Federated Learning. The paper introduces knowledge distillation across heterogeneous clients requiring different precision/architectures. To that effect a loss function is formulated, and the algorithm QuPeD is proposed. Theoretical converge of QuPed is provided, along with experimental validation.

**Ethical Concerns:**

No ethical concerns.

**Limitations And Societal Impact:**

Mentioned briefly. For further limitations, see above for detailed comments.

**Main Review:**

The difference between model x_i and quantization center c_i should be emphasized. I would like to see an example of such quantization with particular values of c_i.

Motivate the choice of R(x,c). l_1 norm is used, with is suited for proximal algorithms used in QuPeD, but such choice should be motivated well. what not l_2. Is there any sparsity concerns?

As mentioned by the authors themselves, the bounded gradient assumption is too strong.

Further explanation needed on Assumptions A.4 and A.5. Appendix A does not provide any values of the lipschitz/smoothness parameters. Since, the authors are using sigmoid/tanh, I would like to see the parameters derived for such functions. Only A.6 is justified in the appendix, by connecting it to Fallah, et.al. Moreover, A.7 seems very strong. Overall I believe that the set of assumptions are too strong, and more justifications are required. Either show standard instances where these thing hold, or argue that these are standard by citing papers using the same assumption.

Theorem 2 is very hard to parse. There are several parameters, and scaling of those are not clear, so the result right now is non-intuitive. Also, after the theorem, I would request the authors to add implications/remarks about the theorem, maybe get back to some standard results.

I have a question about the 1/\sqrt{T} rate on gradient-norm-squared. In standard distributed learning, with full gradient (which is same as this paper), a 1/T rate on gradient norm squared can be obtained (see https://arxiv.org/pdf/1911.09721.pdf, https://arxiv.org/pdf/1803.01498.pdf with no Byzantine setup). I would request the authors to comment on this. Why is the rate slower? Is this because of the proximal operator? Is it optimal? Why do we need a proximal operator if it hurts the rate of conergence?


**Time Spent Reviewing:**

3

---

> ### Author Response · Authors · 2021-08-10
> **Explanation for questions, Assumptions A.4,5,7; Remarks on Theorem 2 and convergence rate.**
>
> We thank the reviewer for their comments and time.
> * **Difference between $\mathbf{x}_i$ and $\mathbf{c}_i$.** $\mathbf{x}_i$ denotes client $i$’s model parameters and $\mathbf{c}_i$ denotes the set of quantization values to which $\mathbf{x}_i$ will be mapped via hard quantization; both of these parameters are being optimized over in our relaxed optimization problem to find good quantization values and mappings. On Cifar-10 with 1Bit CNN2, one example set of centers is as follows: [(-0.0593,0.681), (-0.0794, 0.0845), (-0.0340,0.0335), (-0.0667, 0.0542)] where each tuple denotes the codebook entries for a layer.
>
> * Our motivation for choosing $\ell_1$ norm (over the $\ell_2$ norm) for $R(\mathbf{x},\mathbf{c})$ is due to empirical performance; we tried both and observed that $\ell_1$ performs better than $\ell_2$.
>
> * Bounded gradient, while a strong assumption, is used frequently for non-convex optimization. See [1] for a case in personalized FL setting. We hope that our novel analysis can inspire future works where it is relaxed.
> * **Assumptions A.4, A.5 are implied by the choice of $\widetilde{Q}_\mathbf{c}(\mathbf{x})$.** Firstly, apologies for the missing part in Appendix A. We had prepared a section illustrating an example for soft quantization function and showing its smoothness properties, but commented that out by mistake. In particular, we can define the following soft quantization function: $\widetilde{Q}_\mathbf{c}(\mathbf{x}):\mathbb{R}^{d+m}\rightarrow\mathbb{R}^d$ with $\widetilde{Q}_\mathbf{c}(\mathbf{x})_i:=\sum_\{j=2\}^m (c_j-c_\{j-1\})\sigma(P(x_i-\frac{c_j+c_\{j-1\}}{2}))+c_1$ for any $i\in[d]$, where $\sigma$ denotes the sigmoid function and $P$ is the so called temperature parameter controlling how closely $\widetilde{Q}_\mathbf{c}(\mathbf{x})$ approximates $Q_\mathbf{c}(\mathbf{x})$. This function can be seen as a simplification of the function that was used in [2]. Using several standard facts (including that $\sigma(P\mathbf{x})$ is a $\frac{P}{4}$-Lipschitz function)
> gives **A.4, A.5** for this function.
> * **Assumption A.7 is corollary of previous assumptions.** Assumption **A.7** is not a strong one; it is, in fact, a corollary of our other assumptions. This follows from the facts that (i) the loss functions $f(\mathbf{x})$ and $f_\mathbf{w}(\mathbf{w})$ (this is a neural network loss function defined for the global model) are smooth and (ii) $f^{KD}(\mathbf{x}, \mathbf{w})$ is (almost) a version of these loss functions with soft labels. With these facts in place, it can be shown that Assumptions **A.1-A.5** imply Assumption **A.7**. We will in fact add the sections about soft quantization function and full proof of how Assumptions **A.1-A.5** imply Assumption **A.7** to Appendix A in the revision.
> * **Missing implications of Theorem 2.** We actually have a remark (**Remark 4**) right after **Theorem 2** giving more insights into the result along-with discussing its implications. There we make an important first observation: "Suppose data distributions are fixed across clients (i.e., $\kappa_i$'s are fixed) and we need to choose model architectures/precision for each client in the federated ecosystem. Then, for a fast convergence, for the clients that have local data that is not a representative of the general distribution (large $\kappa_i$), it is critical to choose models with small smoothness parameter (e.g., choosing a less aggressive quantization); whereas, clients with data that is representative of the overall data distribution (small $\kappa_i$) can tolerate having a less smooth model.". Our convergence rate matches the convergence rate of [3] which is $\mathcal{O}(\frac{1}{\sqrt{T}})$ with an constant error term depending on data heterogeneity $\overline{\kappa}$.  We are sorry that Theorem 2 is hard to parse, we had to introduce new constants due to space limitation; we hope the remark makes the theorem more interpretable for you. If you have further questions regarding Theorem 2 we would be happy to answer.
> * **$\mathcal{O}(\frac{1}{\sqrt{T}})$ convergence is slow.** Thank you for asking this question. It is a subtle point. This is not because of the proximal updates. In fact, it all depends on what error we want to tolerate; for example, we could also get $\mathcal{O}(\frac{1}{T})$ rate, but with a slightly increased error, as explained below. Since the referenced Byzantine papers do not perform local iterations, to make a fair comparison, let us take $\tau=1$ in our bound in Theorem 2. Note that with our current convergence rate of $\mathcal{O}(\frac{1}{\sqrt{T}})$ in Theorem 2, we get an error that scales purely with data heterogeneity and nothing else (see the last term on line 756 in the proof of Theorem 2). To get a better rate, consider the bound on the gradients in line 755. Note that the learning rate $\eta_3$ has to satisfy $\eta_3\leq\frac{1}{4\lambda_pL_w\sqrt{C_L}}$. If we simply put this in the bound on the RHS of line 755, we would also get a rate of $\mathcal{O}(\frac{1}{T})$ but with an error that now depends not only on the heterogeneity but also on the Lipschitz constants and other parameters, which may be slightly bigger than the error that we get with the $\mathcal{O}(\frac{1}{\sqrt{T}})$ convergence rate, as in Theorem 2. In order to get an error as small as possible, we choose $\eta_3=\frac{1}{4\lambda_pL_w\sqrt{C_L}\sqrt{T}}$, which gives $\mathcal{O}(\frac{1}{\sqrt{T}})$ rate with an error that only depends on the data heterogeneity. We believe that the error in our convergence is unavoidable as clients learn different personalized models (tailored to their local data) and so it is likely that the convergence error will depend on the data heterogeneity. We also would like to point out that there are in fact other papers on personalized federated learning (albeit in a much simpler setting than ours) on heterogeneous data that get convergence rate and error similar to ours; see, for example, [3] below.
>
> We hope to have answered your questions and concerns adequately; if not, we are happy to clarify further.
>
> **References**
>
> [1] Fallah et al. "Personalized federated learning: A meta-learning approach." Advances in Neural Information Processing Systems 33. 2020.
>
> [2] Yang et al. "Quantization networks." Proceedings of the IEEE/CVF Conference on Computer Vision and Pattern Recognition. 2019.
>
> [3] Dinh et al. "Personalized Federated Learning with Moreau Envelopes." Advances in Neural Information Processing Systems 33. 2020.

---

> > ### Comment · Reviewer_UdzD · 2021-08-24
> > **Thanks the reviewers for answering my questions**
> >
> > I thank the reviewer for answering all my concerns. Apart from the set of assumptions, all my queries are answered. I am increasing my score to 7.

---

### Decision · Program_Chairs · 2021-09-28

**Decision:**

Accept (Poster)

**Comment:**

The paper discusses how to properly design personalized models in Federated Learning. A novel aspect of the paper is to  introduce knowledge distillation across heterogeneous clients, and require that different precision/architectures can be used across different network users.  Theoretical converge of the proposed QuPed is provided, along with extensive experimental validation. Overall the paper is well-written, and the authors have made some interesting observations, such as  how data and resource heterogeneity can affect the convergence rate, and that clients with more representative data can tolerate more aggressive compression are interesting to the community. On the downside, the KD for (personalized) federated learning has already been studied, and the optimization formulation for the proposed algorithm seems to be a straightforward combination of different aspects of the modeling (quantization and KD).

**Consistency Experiment:**

NeurIPS has a long history of experimentation. In 2014, NeurIPS ran an experiment in which 10% of submissions were reviewed by two independent committees to quantify the randomness in the review process. This year, we repeated a variant of this experiment to see how the quality of the review process has changed over time.  This paper was part of the experiment and was therefore assigned to two committees (consisting of reviewers, an Area Chair, and a Senior Area Chair) that reached independent decisions.  If both committees made the same recommendation, this recommendation was followed. If a single committee recommended acceptance, the paper was accepted (with the exception of a few cases in which the other committee identified what we considered a fatal flaw, e.g., an error in a key result).

Both committees reached the same decision: **Accept (Poster)**

The other committee assigned to the paper recommended **Accept (Poster)**.  You can find the other set of reviews, along with any follow up discussion with the authors here:
https://openreview.net/forum?id=Yowoe1scJOD